# Constraints on beta functions in field theories

Han Ma$^{\triangle}$, Sung-Sik Lee$^{\triangle,\dagger}$

$^{\triangle}$ *Perimeter Institute for Theoretical Physics,*
*Waterloo, Ontario N2L 2Y5, Canada,*
*and*
$^{\dagger}$ *Department of Physics & Astronomy,*
*McMaster University,*
*1280 Main St. W., Hamilton,*
*Ontario L8S 4M1, Canada*

## Abstract

The $\beta$-functions describe how couplings run under the renormalization group flow in field theories. In general, all couplings that respect the symmetry and locality are generated under the renormalization group flow, and the exact renormalization group flow is characterized by the $\beta$-functions defined in the infinite dimensional space of couplings. In this paper, we show that the renormalization group flow is highly constrained so that the $\beta$-functions defined in a measure zero subspace of couplings completely determine the $\beta$-functions in the entire space of couplings. We provide a quantum renormalization group-based algorithm for reconstructing the full $\beta$-functions from the $\beta$-functions defined in the subspace. As examples, we derive the full $\beta$-functions for the $O(N)$ vector model and the $O_L(N) \times O_R(N)$ matrix model entirely from the $\beta$-functions defined in the subspace of single-trace couplings.

**CONTENTS**

## I. INTRODUCTION

One of the greatest advances in modern theoretical physics is the invention of the renormalization group (RG) [1–12]. The idea is to organize a complicated many-body system in terms of length scales of constituent degrees of freedom. Thanks to locality that greatly limits the way short-distance modes influence long-distance modes, one can understand coarse grained properties of the system in terms of effective field theories without considering all degrees of freedom in the system. This opens the door for systematic understandings of many physical phenomena that are otherwise too difficult to study theoretically.

The central object in RG is the $\beta$-function. It describes how an effective theory gradually changes as the length scale is increased. The renormalization of the couplings for long-wavelength modes is driven by fluctuations of short-wavelength modes, which creates the RG flow in the space of theories. While the $\beta$-function contains the full information on the fate of a system in the infrared limit, it is in practice hard to keep track of the exact RG flow. Even if one starts with a relatively simple theory with a small number of couplings at a short distance scale, all couplings that respect symmetry and locality are eventually

generated at larger distances. In general, one has to deal with the RG flow in the infinite dimensional space of couplings.

Therefore, it is desirable to take advantage of constraints that $\beta$-functions satisfy if there is any. It is easy to see that not all $\beta$-functions are independent around free field theory fixed points. For example, the scaling dimension of $\phi^{2n}$ is $n$ times that of $\phi^2$ at the Gaussian fixed point. Therefore, the $\beta$-function of the former is fixed by that of the latter to the linear order in the sources for the operators. It is then natural to ask whether such constraints exist for interacting theories and, if so, what the general rules are. There are proposals under special circumstances [13, 14]. In this paper, we show that $\beta$-functions in all field theories are highly constrained : *the $\beta$-functions defined in a measure zero subspace of couplings completely fix the $\beta$-functions in the entire space of couplings.*

Our result is beyond the well known constraint for beta functions present in continuum field theories. Consider a field theory defined non-perturbatively with a finite UV cutoff. Examples include field theories regularized on lattice. While infinitely many couplings can be turned on in the UV theory, at energy scales much smaller than the UV cutoff all couplings are fixed by a finite number of marginal and relevant couplings. As a result, the flow of most couplings is controlled by the marginal and relevant couplings in the low-energy limit. However, the constraints discussed in our paper applies to $\beta$-functions at all energy scales including the scales that are comparable to the UV cutoff. At high energy scales close to the UV cutoff, irrelevant couplings can be tuned independently, and they are not fixed by the marginal and relevant couplings through the usual constraint that emerges only in the low-energy limit. In this paper, we are concerned about the general kinematic constraints that $\beta$-functions obey at all energy scales.

To uncover the constraints that $\beta$-functions satisfy in general field theories, we use the quantum RG[15, 16]. Quantum RG reformulates the Wilsonian RG by projecting the full RG flow onto a subspace of couplings. The subspace is spanned by couplings for the so-called single-trace operators. Single-trace operators are basic building blocks of general operators in that all operators that respect the symmetry can be written as composites of single-trace operators. In large $N$ theories, the set of single-trace operators consists of the operators that involve one trace of flavor or color indices[17]. However, the notion of single-trace operators can be defined in any field theory once the fundamental degrees of freedom and the symmetry of the theory are fixed[16]. Although quantum RG does not include composites of the single-trace operators (called multi-trace operators) directly, it exactly takes into account their effects by promoting the single-trace couplings to dynamical variables. The pattern of fluctuations of the single-trace couplings precisely captures the multi-trace couplings. As a result, the classical Wilsonian RG flow defined in the full space of couplings is replaced with a sum over RG paths defined in the subspace of single-trace couplings in the quantum RG. The $\beta$-functions of the Wilsonian RG is then replaced with

an action for dynamical single-trace couplings that determines the weight of fluctuating RG paths. The bulk theory constructed from quantum RG is free of UV divergence as far as the original theory is regularized.

For a $D$-dimensional field theory, the theory for the dynamical single-trace couplings takes the form of a $(D + 1)$-dimensional theory, where the dynamical couplings depend on the $D$-dimensional space and the RG scale. The theory includes dynamical gravity because the coupling functions for the single-trace energy-momentum tensor is nothing but a metric that is promoted to a dynamical variable in quantum RG[16]. For this reason, quantum RG provides a natural framework for the AdS/CFT correspondence [18–20] in which the extra dimension in the bulk is interpreted as the RG scale [21–28][1].

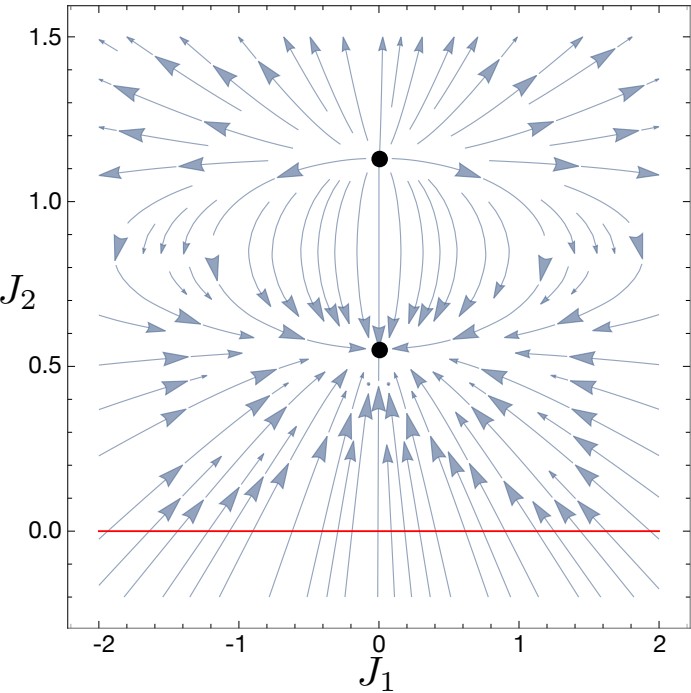

FIG. 1. RG flow of a toy model considered in Sec. V A. $J_1(J_2)$ is the single-trace (double-trace) coupling. The $\beta$-functions in the $J_2 = 0$ subspace, which is denoted as the red horizontal line, fix the full $\beta$-functions in the space of $J_1$ and $J_2$. The full $\beta$-functions exhibit rich structures that include one stable and one unstable fixed points away from the single-trace subspace, which are fully encoded in the $\beta$-functions within the subspace of $J_2 = 0$.

In the Wilsonian RG, a field theory is represented as a point in the space of all couplings.

---

[1] In order to construct a background independent gravitational theory for fluctuating couplings in quantum RG, one needs to use a coarse graining scheme[29] that does not introduce a fixed background [15, 30] and satisfy a consistency condition[31, 32]. For our purpose of demonstrating the existence of constraints of $\beta$-functions, however, the issue is not crucial. The quantum RG is an exact reformulation of the Wilsonian RG in any coarse graining scheme irrespective of whether the scheme is background independent or not.

In quantum RG, a field theory is represented as a wavefunction defined in the subspace of the single-trace couplings. The peak position of the wavefunction indicates the value of the single-trace coupling. Around the peak, the second and higher moments of the fluctuating single-trace coupling contain the information on the double-trace and higher-trace couplings. The classical flow of couplings in the Wilsonian RG is replaced with a quantum evolution of the wavefunction in quantum RG. The bulk theory that governs the quantum RG flow is entirely fixed by the $\beta$-functions defined in the subspace of single-trace couplings[15, 16]. Since the wavefunction at an RG scale encodes the full information on all couplings at the scale, the bulk theory fully determines the $\beta$-functions of all multi-trace couplings. This implies that the full $\beta$-functions is fixed by $\beta$-functions defined in the subspace of single-trace couplings. A simple example that illustrates the main result of this paper is shown in Fig. 1.

In this work, we provide a general algorithm for extracting the full $\beta$-functions from the $\beta$-functions defined in the subspace of single-trace couplings. The algorithm consists of the following steps. First, we construct the bulk theory for quantum RG from the $\beta$-functions defined in the space of single-trace couplings. Second, we solve the (functional) Schrodinger equation that evolves an initial state fixed by the UV theory to IR. Finally, we identify the ground state of the quantum RG Hamiltonian as the IR fixed point of the theory, and states with local excitations as the IR fixed point deformed with local operators. This allows us to extract the full $\beta$-functions from the spectrum of the quantum RG Hamiltonian. This dictionary in the algorithm is summarized in Tab. I.

TABLE I. Dictionary of correspondence between boundary field theory and bulk theory.

| boundary field theory | bulk theory for quantum RG |
| --- | --- |
| RG time (logarithmic length scale) | extra bulk dimension |
| single-trace coupling (operator) | dynamic field (conjugate momentum) |
| Boltzmann weight in the partition function | bulk state |
| RG transformation | radial quantum evolution |
| stable fixed point | ground state of the RG Hamiltonian |
| local scaling operators | local excitations |
| scaling dimensions | spectrum of the RG Hamiltonian |

The rest of the paper is organized as follows. In Sec. II, we present the main result of our paper using two concrete models : the $O(N)$ vector model and the $O_L(N) \times O_R(N)$ matrix model regularized on a lattice. Through quantum RG, the exact RG flow is mapped into quantum evolution of a wavefunction defined in the space of single-trace couplings. We show that the resulting quantum theories from quantum RG flow of the regularized field theories are finite and well defined. We explicitly compute the full $\beta$-functions for these models from the $\beta$-functions defined in the subspace of single-trace couplings. Eq. (37)

and Eq. (60) are the main results. In Secs. III and IV, we generalize the results obtained through the concrete examples. In Sec. V, we consider toy models in which the bulk theory is non-interacting, and the exact RG flow equation can be exactly solved through quantum RG.

## II. CONSTRAINTS ON $\beta$ FUNCTIONS

In this section, we illustrate the main result of the paper using two examples. The first example is the $O(N)$ vector model, and the second one is the $O_L(N) \times O_R(N)$ matrix model. To be concrete, we consider those theories regularized on a $D$-dimensional Euclidean lattice. We first review how the exact Wilsonian RG defined in the space of all couplings can be reformulated as a quantum evolution of wavefunction defined in the subspace of single-trace couplings[15, 16]. From this, we constructively show that the full beta functions are determined from the beta functions defined in the subspace of single-trace couplings.

### A. The O(N) vector model

In describing RG flow of a theory, it is convenient to choose a reference theory as the origin of the space of theories. A general theory is then viewed as a theory obtained by adding a deformation to the reference theory. The RG flow then describes the change of the deformation as a function of length scale. We write the $O(N)$ vector model as

$$S[\phi] = S_0[\phi] + S_1[\phi], \tag{1}$$

where $S_0$ is the reference action,

$$S_0[\phi] = \frac{1}{2}m^2 \sum_i (\phi_i \phi_i), \tag{2}$$

and $S_1$ is a deformation,

$$S_1 = \sum_{ij} J_{ij}^{(1)} (\phi_i \phi_j) + \frac{J^{(2)}}{N} \sum_i (\phi_i^2)^2. \tag{3}$$

$\phi_i^a$ is a real field with flavour $a = 1, 2, .., N$ defined at site $i$, and $(\phi_i \phi_j) \equiv \sum_a \phi_i^a \phi_j^a$. $S_0$ represents the trivial gapped fixed point. $J_{ij}^{(1)}$ is the hopping amplitude between site $i$ and $j$, and $J^{(2)}$ is the on-site quartic interaction. Depending on the magnitudes of the deformations, the theory may stay in the insulating phase, or flows to a different fixed point associated with the critical point or the symmetry broken state. Our goal is to understand the exact RG flow of the theory. Since we choose the ultra-local gapped fixed point action

as the reference action, we use the real space RG scheme in which $S_0$ is invariant under the coarse graining transformation. However, different RG schemes can be used as is discussed in Appendix A. The choice of RG scheme does not affect the physics.

### 1. Classical RG

We first review the exact Wilsonian RG[2]. The exact Wilsonian RG flow is generated from the following steps[6].

- Separation of $\phi$ into low-energy modes and high-energy modes :

  This is needed before we integrate out the high-energy modes to obtain an effective action for the low-energy modes. In the real space, we usually consider a scheme where a block of sites is merged to generate a coarse-grained lattice[33]. However, this forces the RG steps to be discrete. To avoid this, we employ the scheme in which the field $\phi$ is partially integrated out without changing the number of sites. For this, we introduce an auxiliary field $\Phi$ with mass $\mu$. The total action is written as

  $$S[\phi, \Phi] = S_0[\phi] + \tilde{S}_0[\Phi] + S_1[\phi], \tag{4}$$

  where

  $$\tilde{S}_0[\Phi] = \frac{1}{2}\mu^2 \sum_i \Phi_i^2. \tag{5}$$

  Now we rotate $\phi$ and $\Phi$ into a new pair of fields,

  $$\phi_i = \phi_i' + \Phi_i', \quad \Phi_i = A\phi_i' + B\Phi_i', \tag{6}$$

  where $A = \frac{m^2}{\tilde{\mu}\mu}$ and $B = -\frac{\tilde{\mu}}{\mu}$ with $\tilde{\mu} = \frac{m}{\sqrt{e^{2dz}-1}} \approx \frac{m}{\sqrt{2dz}}$. $dz$ is an infinitesimal parameter that labels the continuous coarse graining steps. The coefficients are chosen such that the original field $\phi$ is given by the sum of the low-energy field ($\phi'$) and the high-energy field ($\Phi'$), and the low-energy field has mass $me^{dz}$. The action for $\phi'$ and $\Phi'$ becomes

  $$S[\phi', \Phi'] = \frac{1}{2}m^2 e^{2dz} \sum_i (\phi_i')^2 + \frac{1}{2}\frac{m^2}{2dz}(\Phi_i')^2 + S_1[\phi' + \Phi']. \tag{7}$$

  The field $\phi'$ acquires the larger mass indicating that it has less fluctuation than $\phi$. The missing fluctuation is transferred to the higher energy field $\Phi'$.

---

[2] In this paper, we use the terms Wilsonian RG and classical RG interchangeably.

- Coarse graining :

  The high energy field $\Phi'$ is integrated out. This has the effect of partially including fluctuations of physical degrees of freedom without reducing the number of sites. This gives rise to corrections that renormalize $S_1$ to $S_1 + \delta S_1$, where

  $$\delta S_1 = \frac{dz}{m^2}\left[\frac{\partial^2}{\partial(\phi_i')^2}S_1 - \left(\frac{\partial S_1}{\partial\phi_i'}\right)^2\right] \tag{8}$$

  up to the leading order in $dz$.

- Rescaling of field;

  To be able to perform this coarse graining procedure iteratively, we need to bring the mass from $me^{dz}$ back to $m$. This can be done by rescaling the field as

  $$\phi_i' = e^{-dz}\phi_i''. \tag{9}$$

  This restores the original reference action and generates an additional correction to $S_1$,

  $$\delta S_1' = -dz\phi_i''\frac{\partial}{\partial\phi_i''}. \tag{10}$$

This completes one cycle of the coarse graining. After this exact RG transformation, the effective action becomes $S(dz) = S + \delta S$, where

$$\delta S = -Ndz\Big[\sum_i \beta_i^{(0)} + \frac{1}{N}\sum_{\{i_1,j_1\}}\beta_{i_1,j_1}^{(1)}(\phi_{i_1}\phi_{j_1}) + \sum_{\{i_1,j_1,i_2,j_2\}}\beta_{i_1,j_1,i_2,j_2}^{(2)}\frac{1}{N^2}(\phi_{i_1}\phi_{j_1})(\phi_{i_2}\phi_{j_2})$$
$$+ \sum_{\{i_1,j_1,i_2,j_2i_3j_3\}}\beta_{i_1,j_1,i_2,j_2,i_3j_3}^{(3)}\frac{1}{N^3}(\phi_{i_1}\phi_{j_1})(\phi_{i_2}\phi_{j_2})(\phi_{i_3}\phi_{j_3})\Big]. \tag{11}$$

Here the beta functions are $\beta_i^{(0)}[J^{(1)},J^{(2)}] = -\frac{2}{m^2}J_{ii}^{(1)}$, $\beta_{ij}^{(1)}[J^{(1)},J^{(2)}] = 2J_{ij}^{(1)} + \frac{4}{m^2}\sum_k J_{ki}^{(1)}J_{kj}^{(1)} - \frac{4J^{(2)}}{m^2}(1+\frac{2}{N})\delta_{ij}$, $\beta_{i_1j_1i_2j_2}^{(2)}[J^{(1)},J^{(2)}] = 4J^{(2)}\delta_{i_1j_1}\delta_{j_1j_2}\delta_{i_1i_2} + \frac{16}{m^2}J^{(2)}J_{i_1j_1}^{(1)}\delta_{i_1i_2}\delta_{i_2j_2}$, $\beta_{i_1j_1i_2j_2i_3j_3}^{(3)}[J^{(1)},J^{(2)}]$ $= \frac{16(J^{(2)})^2}{m^2}\delta_{i_1j_1}\delta_{i_2j_2}\delta_{i_3j_3}\delta_{i_1i_2}\delta_{i_1i_3}$. The exact RG transformation not only renormalizes the terms that are already present in the action but also generates new operators that are order of $\phi^6$. In the subsequent RG steps, infinitely many other operators are generated. The general effective action takes the form of

$$S = S_0 + N\sum_{K=1}^{\infty}\sum_{i_1,j_1,\ldots,i_K,j_K}J_{i_1,j_1,\ldots,i_K,j_K}^{(K)}\mathbf{O}_{i_1,j_1,\ldots,i_K,j_K}, \tag{12}$$

where

$$\mathbf{O}_{i_1,j_1,\dots,i_K,j_K} = \frac{1}{N^K} \prod_{n=1}^{K} (\phi_{i_n}\phi_{j_n}) \tag{13}$$

is the set of most general $O(N)$ invariant operators. $\mathbf{O}_{i_1j_1;i_2j_2;..;i_Kj_K}$ with $K = 0$ is the identity operator. $\mathbf{O}_{i_1j_1;i_2j_2;..;i_Kj_K}$'s with $K = 1$ are referred to as single-trace operators because they involve one summation of flavour indices. Those with $K > 1$ are multi-trace operators. $J^{(K)}_{i_1,j_1,\dots,i_K,j_K}$ is the source for $\mathbf{O}_{i_1,j_1,\dots,i_K,j_K}$. In Eq. (12) and Eq. (13), factors of $N$ are chosen so that $J^{(K)}_{i_1,j_1,\dots,i_K,j_K}$ and $\mathbf{O}_{i_1,j_1,\dots,i_K,j_K}$ are $\mathcal{O}(1)$. Even in the large $N$ limit, multi-trace couplings are not negligible. The exact RG flow is encoded in the beta functions,

$$\frac{dJ^{(K)}_{i_1,j_1,\dots,i_K,j_K}}{dz} = -\beta^{(K)}_{i_1,j_1,\dots,i_K,j_K}\left(J^{(1)}, J^{(2)}, \dots\right) \tag{14}$$

that is defined in the infinite dimensional space of couplings, $\left\{J^{(K)}_{i_1,j_1,\dots,i_K,j_K}\right\}$. Since each coupling in $\left\{J^{(K)}_{i_1,j_1,\dots,i_K,j_K}\right\}$ can be added to the UV theory in Eq. (1) and tuned independently [3], there is in general no particular relation among the couplings at high energy scales. A universal relation among couplings emerge only in the long distance limit as all couplings are determined in terms of a few relevant and marginal couplings in the continuum limit. Here our goal is to describe the entire RG flow that covers from the lattice scale to the long distance limit. At short length scales, couplings are not related to each other, and the full beta functions at general values of couplings are needed in Eq. (15). While the full $\beta$-functions define the vector field in the infinite dimensional space of all couplings, we will show that the information of all $\beta$-functions is entirely encoded in the $\beta$-functions defined in a subspace of single-trace couplings. We emphasize that this constraint among beta functions holds at all length scales even close to the lattice scale, and is not a consequence of the relations that emerge in the long distance limit.

### 2. Quantum RG

The exact RG flow of the effective action can be written as[6]

$$\frac{\partial S_1(z)}{\partial z} = \frac{1}{m^2}\left[\frac{\partial}{\partial\phi_i}\frac{\partial}{\partial\phi_i}S_1(z) - \left(\frac{\partial S_1(z)}{\partial\phi_i}\right)^2\right] - \phi_i\frac{\partial S_1(z)}{\partial\phi_i}, \tag{15}$$

where the total effective action at scale $z$ is given by $S(z) = S_0 + S_1(z)$. This, in turn, can be written as a differential operator acting on $e^{-S}$ as

$$e^{-S(z)} = e^{-\hat{H}z}e^{-S}, \tag{16}$$

_______

[3] For local theories, the multi-local couplings should decay exponentially in space.

where

$$\hat{H} = e^{-\hat{S}_0} \sum_{i,a} \left[ i(\hat{\phi}_i \hat{\pi}_i) + \frac{1}{m^2}(\hat{\pi}_i \hat{\pi}_i) \right] e^{\hat{S}_0}, \tag{17}$$

and $\hat{\pi}_i^a = -i\frac{\delta}{\delta \phi_i^a}$ is the conjugate momentum of $\hat{\phi}_i^a$. In Eq. (16), $e^{-S}$ plays the role of a wavefunction, and $\hat{H}$ acts as a quantum Hamiltonian for an imaginary time evolution. Here, the imaginary time corresponds to the logarithmic length scale in RG. For this reason, we call $\hat{H}$ the *RG Hamiltonian*.

The observation that the RG flow can be generated from the quantum RG Hamiltonian suggests that the space of theories can be viewed as a vector space. In this picture, $e^{-S}$ is viewed as a wave function, and the partition function becomes an overlap between two wavefunctions[30],

$$Z = \int \mathcal{D}\phi \; e^{-S} = \langle \mathbb{1} | S \rangle, \tag{18}$$

where

$$|S\rangle = \int \mathcal{D}\phi \; e^{-S[\phi]} |\phi\rangle \tag{19}$$

is the state associated with the action $S$[30], and

$$|\mathbb{1}\rangle = \int \mathcal{D}\phi \; |\phi\rangle \tag{20}$$

is the state whose wavefunction is 1. $|\phi\rangle$ denotes the basis state whose inner product is given by $\langle \phi' | \phi \rangle = \prod_{i,a} \delta\left( \phi_i'^a - \phi_i^a \right)$. Although $|\mathbb{1}\rangle$ is not normalizable, the overlap in Eq. (18) is well defined. One can check that the RG Hamiltonian leaves Eq. (20) invariant when applied from the right : $\langle \mathbb{1} | H = 0$ or equivalently $H^\dagger | \mathbb{1} \rangle = 0$. Therefore, the partition function is invariant under the insertion of the RG evolution operator in the overlap,

$$Z = \langle \mathbb{1} | e^{-z\hat{H}} | S \rangle, \tag{21}$$

where $dz$ is an infinitesimal change of the logarithmic length scale. This reflects the fact that the partition function is unchanged under the RG transformation. Only the form of the effective action changes as a function of the length scale. The flow of the effective action is encoded in the state evolution,

$$|S(z)\rangle = e^{-z\hat{H}} |S\rangle, \tag{22}$$

and the effective action at scale $z$ is given by $S(z) = -\ln\langle \phi | S(z) \rangle$. Even if $S$ has only simple interaction terms such as the ultra-local quartic interaction, all local operators that

are invariant under the $O(N)$ symmetry are generated in $S(z)$ for $z > 0$. This makes it difficult to follow the exact RG flow in the space of actions.

The complication can be alleviated in quantum RG that takes advantage of the facts that the space of theories can be viewed as a Hilbert space, and that there exists a set of basis states that span the full Hilbert space. The full Hilbert space associated with the $O(N)$ invariant action is spanned by a set of basis state whose wavefunctions include only the single-trace operators,

$$\mathcal{O}_{ij} = \frac{\phi_i \phi_j}{N}. \tag{23}$$

The basis states are written as

$$|t\rangle = \int \mathcal{D}\phi \, e^{-S_0 + iN \sum_{ij} t_{ij} \mathcal{O}_{ij}} |\phi\rangle. \tag{24}$$

Here $S_0$ is the fixed reference action, and the basis state is labeled by the bi-local field, $t_{ij}$. Because general $O(N)$ invariant operators in Eq. (13) can be written as polynomials of Eq. (23), Eq. (24) forms a complete basis. Suppose we start with the general $O(N)$ invariant action shown in Eq. (12). The state associated with Eq. (12) can be written as a linear superposition of $|t\rangle$ as

$$|S\rangle = \int \mathcal{D}t \, \Psi_J[t] |t\rangle, \tag{25}$$

where

$$\Psi_J[t] = \int \mathcal{D}p \, e^{-iN \sum_{ij} t_{ij} p_{ij} - N \sum_{i_1, j_1, \ldots, i_K, j_K} J^{(K)}_{i_1, j_1, \ldots, i_K, j_K} p_{i_1 j_1} \cdots p_{i_K j_K}}. \tag{26}$$

Here the integrations over the dynamical sources $t_{ij}$ and its conjugate field $p_{ij}$ are defined along the real axis as $\int \mathcal{D}t \equiv \prod_{ij} \int_{-\infty}^{\infty} dt_{ij}$ and $\int \mathcal{D}p \equiv \prod_{ij} \int_{-\infty}^{\infty} dp_{ij}$. $\Psi_J[t]$ is the wavefunction defined in the space of the single-trace couplings. Due to the linear superposition principle, the RG evolution of the general action can be carried out solely in terms of how each basis state is evolved under the RG Hamiltonian. After an infinitesimal RG evolution, we obtain $|S(dz)\rangle = e^{-dz\hat{H}} |S\rangle$, where

$$|S(dz)\rangle = \int \mathcal{D}\phi \left[ \int \mathcal{D}t \Psi_J[t] e^{-S_0 + iN \sum_{ij} t_{ij} \mathcal{O}_{ij} + N \left( \beta^{(0)}[-it, 0, ..] + \sum_{ij} \beta^{(1)}_{ij}[-it, 0, ..] \mathcal{O}_{ij} \right) dz} \right] |\phi\rangle. \tag{27}$$

The resulting state can be written as a linear superposition of the basis states as $|S(dz)\rangle = \int \mathcal{D}t' \, \Psi_J^{dz}[t'] |t'\rangle$, where the new wavefunction is given by

$$\Psi_J^{dz}[t'] = \int \mathcal{D}p' \mathcal{D}t \, e^{dzN \left( -i \sum_{ij} p'_{ij} \frac{t'_{ij} - t_{ij}}{dz} + \beta^{(0)}[-it, 0] + \sum_{ij} \beta^{(1)}_{ij}[-it, 0] p'_{ij} \right)} \Psi_J[t]. \tag{28}$$

Now, Eq. (28) can be viewed as an evolution of the wavefunction defined in the space of single-trace coupling $t$,

$$\Psi_J^{dz}[t] = e^{-Ndz\hat{H}_{bulk}} \ \Psi_J[t], \tag{29}$$

where $\hat{H}_{bulk}$ is the bulk Hamiltonian given by

$$\hat{H}_{bulk} = -\beta^{(0)}[-i\hat{t}, 0] - \sum_{ij} \hat{p}_{ij} \beta_{ij}^{(1)}[-i\hat{t}, 0]$$

$$= \frac{2i}{m^2} \sum_k \hat{t}_{kk} - 2i \sum_{kl} \hat{p}_{kl} \hat{t}_{kl} + \frac{4}{m^2} \sum_{kji} \hat{p}_{ij} \hat{t}_{ik} \hat{t}_{kj}. \tag{30}$$

Here, $\hat{t}$ and $\hat{p}$ are conjugate operators $[\hat{t}_{ij}, \hat{p}_{kl}] = -\frac{i}{2N} (\delta_{ik}\delta_{jl} + \delta_{il}\delta_{kj})$. The state at scale $z$ becomes

$$\Psi_J^z[t] = e^{-Nz\hat{H}_{bulk}} \ \Psi_J[t], \tag{31}$$

and the exact renormalized effective action at scale $z$ is given by $S(z) = -\log \int Dt \ \Psi_J^z[t]\langle\phi|t\rangle$. $\Psi_J^z[t]$ is not Gaussian, and the non-Gaussianity of the wavefunction encodes the higher order operators that are generated in the effective action. Through the standard mapping, Eq. (31) can be written as a path integration as

$$\Psi_J^z[t^z] = \int \mathbf{D}t\mathbf{D}p \ e^{-NS_{bulk}} \ \Psi_J^0[t^0], \tag{32}$$

where $\int \mathbf{D}t\mathbf{D}p = \int \prod_{0 \le z' < z} \mathcal{D}t^{z'} \prod_{0 < z' \le z} \mathcal{D}p^{z'}$ sums over RG paths for $t_{ij}(z)$ in the subspace of the bi-local single-trace couplings (and the conjugate variables), and $S_{bulk}$ is the bulk action that determines the weight for each RG path,

$$S_{bulk} = \int_0^z dz' \left[ i \sum_{ij} p_{ij}^{z'} \partial_{z'} t_{ij}^{z'} + H_{bulk}[p^{z'}, t^{z'}] \right]. \tag{33}$$

The bulk theory is fully regularized and well defined because the bi-local variables are on the lattice on which the original field theory is defined. For a system made of $L$ sites, there exist $L(L+1)/2$ independent bi-local fields, and Eq. (31) has no UV divergence. The exact RG evolution originally defined in the space of all couplings is now replaced with a path integration of fluctuating RG paths in the subspace of single-trace couplings [15]. This mapping is exact at any $N$. In the large $N$ limit, one can use the semi-classical approximation to replace the bulk path integration with a saddle-point approximation[4]. For alternative approaches to the $O(N)$ vector model, see Refs. [35–37].

---

[4] For an explicit computation of the effective action in the large $N$ limit, see Ref. [34].

### 3. Full $\beta$-functions

$\hat{H}_{bulk}$ in Eq. (30) is entirely fixed by the $\beta$-functions defined in the space of single-trace couplings with $J^{(K)} = 0$ for $K > 1$. For the $O(N)$ vector model, only $\beta^{(0)}$ and $\beta^{(1)}_{ij}$ are non-zero in the subspace, and $\hat{H}_{bulk}$ depends only on $\beta^{(0)}[-it, 0, ..]$ and $\beta^{(1)}[-it, 0, ..]$. Since the full RG flow is controlled by $\hat{H}_{bulk}$, the full beta functions can be recovered from these $\beta$-functions . To see this, let us write $\delta S = S(dz) - S$ in terms of the beta functions as

$$\delta S = -dzN \sum_K \sum_{i_1, j_1, ..., i_K, j_K} \beta^{(K)}_{i_1, j_1, ..., i_K, j_K} \prod_{n=1}^{K} \mathcal{O}_{i_n j_n}. \tag{34}$$

In quantum RG, $\delta S$ can be written as

$$\delta S = -\ln \frac{\int \mathcal{D}t \left[ e^{-Ndz\hat{H}_{bulk}} \Psi_J[t] \right] e^{iN \sum_{ij} t_{ij} \mathcal{O}_{ij}}}{\int \mathcal{D}t \ \Psi_J[t] e^{iN \sum_{ij} t_{ij} \mathcal{O}_{ij}}}. \tag{35}$$

Equating Eq. (34) and Eq. (35), we have

$$\sum_K \sum_{i_1, j_1, ..., i_K, j_K} \beta^{(K)}_{i_1, j_1, ..., i_K, j_K} \prod_{n=1}^{K} \mathcal{O}_{i_n j_n} = -\frac{\int \mathcal{D}t \left[ \hat{H}_{bulk} \Psi_J[t] \right] e^{iN \sum_{ij} t_{ij} \mathcal{O}_{ij}}}{\int \mathcal{D}t \ \Psi_J[t] e^{iN \sum_{ij} t_{ij} \mathcal{O}_{ij}}}. \tag{36}$$

In the large $N$ limit, all single-trace operators are independent, and the general beta functions are obtained by equating the coefficient of monomials of single-trace operators in Eq. (36),

$$\beta^{(K)}_{i_1, j_1, ..., i_K, j_K} \left( J^{(1)}, J^{(2)}, ... \right)$$

$$= -\frac{1}{K!} \left( \frac{\partial}{\partial \mathcal{O}_{i_1 j_1}} \right) \cdots \left( \frac{\partial}{\partial \mathcal{O}_{i_K j_K}} \right) \frac{\int \mathcal{D}t \left[ \hat{H}_{bulk} \Psi_J[t] \right] e^{iN \sum_{ij} t_{ij} \mathcal{O}_{ij}}}{\int \mathcal{D}t \ \Psi_J[t] e^{iN \sum_{ij} t_{ij} \mathcal{O}_{ij}}} \Bigg|_{\mathcal{O}=0}. \tag{37}$$

For a finite $N$, not all single-trace operators are independent. For example, $O_{ij}O_{kl} = O_{ik}O_{jl}$ for $N = 1$. This leads to multiple ways to express general operators in terms of the single-trace operators. This is analogous to a gauge freedom in which one physical state can be represented in multiple ways. Namely, $\delta S$ is gauge invariant, but there are multiple ways to express it as a polynomial of $\mathcal{O}_{ij}$. In this case, one has to fix the gauge freedom to determine the $\beta$-functions unambiguously. The natural choice is to treat $\mathcal{O}_{ij}$ as independent variables in Eq. (36). This is possible because both Eq. (34) and Eq. (35) are functions of $\mathcal{O}_{ij}$ only. In this prescription, Eq. (37) holds for any $N$. This is not the only prescription, but Eq. (37) certainly gives the exact RG flow of the effective action for any $N$. The right-hand side of Eq. (37) depends only on $J^{(K)}_{i_1, j_1, ..., i_K, j_K}$, $\beta^{(0)}[-it, 0]$ and $\beta^{(1)}[-it, 0]$ through

Eq. (26) and Eq. (30). This shows that all beta functions in the presence of general couplings $J_{i_1,j_1,\dots,i_K,j_K}^{(K)}$ are completely fixed by the beta functions defined in the subspace of the single-trace couplings.

From Eq. (37), we can find the general expression for $\beta^{(K)}[J^{(1)}, J^{(2)}, \dots]$ from $\beta^{(0),(1)}[-it, 0]$. Starting from the general action in Eq. (12) associated with the wave function in Eq. (26), we obtain

$$\hat{H}_{bulk}\Psi_J[t] = \left(-\beta^{(0)}[-i\hat{t}, 0] - \sum_{ij}\hat{p}_{ij}\beta^{(1)}[-i\hat{t}, 0]\right)\Psi_J[t] \tag{38}$$

$$= \int \mathcal{D}p\, e^{-iNt_{ij}p_{ij}}\left(-\beta^{(0)}[-\frac{1}{N}\frac{\partial}{\partial p}, 0] - \sum_{ij}p_{ij}\beta^{(1)}[-\frac{1}{N}\frac{\partial}{\partial p}, 0]\right)e^{-NJ_{i_1,j_1,\dots,i_K,j_K}^{(K)}p_{i_1 j_1}\dots p_{i_K j_K}}.$$

This gives

$$\int \mathcal{D}t\left[\hat{H}_{bulk}\Psi_J[t]\right]e^{iN\sum_{ij}t_{ij}\mathcal{O}_{ij}} = \left(-\beta^{(0)}[-\frac{1}{N}\frac{\partial}{\partial\mathcal{O}}, 0] - \sum_{ij}\mathcal{O}_{ij}\beta^{(1)}[-\frac{1}{N}\frac{\partial}{\partial\mathcal{O}}, 0]\right)$$

$$\times \exp\left\{-NJ_{i_1,j_1,\dots,i_K,j_K}^{(K)}\mathcal{O}_{i_1 j_1}\dots\mathcal{O}_{i_K j_K}\right\}. \tag{39}$$

In the O(N) vector model, $\beta^{(0)}$ and $\beta^{(1)}$ take the form of

$$\beta^{(0)}[J^{(1)}, 0] = \sum_{kl}\beta_{kl}^{(0),1}J_{kl}^{(1)}, \quad \beta_{ij}^{(1)}[J^{(1)}, 0] = \beta_{ij}^{(1),1}J_{ij}^{(1)} + \sum_{kl}\beta_{ijkl}^{(1),2}J_{ik}^{(1)}J_{jl}^{(1)}, \tag{40}$$

where

$$\beta_{kl}^{(0),1} = -\frac{2}{m^2}\delta_{kl}, \quad \beta_{ij}^{(1),1} = 2, \quad \beta_{ijkl}^{(1),2} = \frac{4}{m^2}\delta_{kl}. \tag{41}$$

Through Eq. (39) the full $\beta$-functions can be written solely in terms of $\beta_{kl}^{(0),1}$ $\beta_{ij}^{(1),1}$ and $\beta_{ijkl}^{(1),2}$ as

$$\beta_{i_1,j_1,\dots,i_K,j_K}^{(K)} = (K+1)\sum_{kl}\beta_{kl}^{(0),1}J_{k,l,i_1,j_1,\dots,i_K,j_K}^{(K+1)} + K\beta_{i_1,j_1}^{(1),1}J_{i_1,j_1,\dots,i_K,j_K}^{(K)}$$

$$- \frac{(K+1)K}{N}\sum_{kl}\beta_{i_1,j_1,k,l}^{(1),2}J_{i_1,k,j_1,l,i_2,j_2\dots,i_K,j_K}^{(K+1)}$$

$$+ \sum_{M=1}^{K}M(K+1-M)\sum_{kl}\beta_{i_1,j_1,k,l}^{(1),2}J_{i_1,k,i_2,j_2,\dots,i_M,j_M}^{(M)}J_{j_1,l,i_{M+1},j_{M+1},\dots,i_K,j_K}^{(K+1-M)} \tag{42}$$

It is noted that the full $\beta$-functions at general couplings are completely characterized by the data in Eq. (41) that defines the $\beta$-functions in the subspace of single-trace coupling[5].

---

[5] As a special case, one can check that the $\beta$-functions in the presence of $J_{ij}^{(1)}$ and $J_{iiii}^{(2)}$ are reproduced.

This shows that $\beta$-functions away from the subspace of single-trace couplings is fixed by $\beta$-functions in the subspace.

The $O(N)$ model is rather special in that $\hat{H}_{bulk}$ is linear in $p_{ij}$[38], and the single-trace coupling $t_{ij}$ is non-dynamical in the bulk. However, the constraints among $\beta$-functions hold for general theories in which $\hat{H}_{bulk}$ is not linear in the conjugate momenta. To see this, we consider a matrix model as our next example[15].

## B.   A matrix model

As a next example, we consider a matrix model defined on a D-dimensional Euclidean lattice. The fundamental field is a real $N \times N$ matrix field, $\phi_i^{aa'}$, where $i$ is the lattice site and $a, a' = 1, 2, .., N$ are the flavour indices. Under the global $O_L(N) \times O_R(N)$ symmetry, the matrix field transforms as $\phi_i \to A\phi_i B$, where $A \in O_L(N)$ and $B \in O_R(N)$. Single-trace operators that are invariant under $O_L(N) \times O_R(N)$ symmetry are denoted as

$$\mathcal{O}_I = \frac{1}{N}\text{tr}(\phi_{i_1}\phi_{i_2}^T\phi_{i_3}\phi_{i_4}^T \ldots \phi_{i_{2m-1}}\phi_{i_{2m}}^T), \tag{43}$$

where the trace sums over the flavour indices, and $I = (i_1, i_2, \ldots, i_{2m})$ is a short-hand notation for a series of sites that form a loop through a trace over flavour indices. Because the trace is invariant under cyclic permutation and transpose, $(i_1, i_2, i_3, i_4, \ldots, i_{2m-1}, i_{2m}) = (i_3, i_4, \ldots, i_{2m-1}, i_{2m}, i_1, i_2)$ and $(i_1, i_2, \ldots, i_{2m}) = (i_{2m}, i_{2m-1}, \ldots, i_2, i_1)$. Henceforth, we refer to $I$ as a loop. The set of single-trace operators plays the special role because general $O(N)$ invariant operators can be written as polynomials of the single-trace operators as

$$\mathbf{O}_{I_1, I_2, \ldots, I_K} = \prod_{n=1}^{K} \mathcal{O}_{I_n}. \tag{44}$$

### 1.   Classical RG

The general action that is invariant under the symmetry can be written as

$$S(z^*) = S_0 + N^2 \sum_{K=1}^{\infty} \sum_{\{I_1, \ldots, I_K\}} J^{(K)}_{I_1, \ldots, I_K} \mathcal{O}_{I_1} \ldots \mathcal{O}_{I_K}, \tag{45}$$

where $S_0 = \frac{N}{2}m^2\text{tr}(\phi_i\phi_i^T)$ is the reference action, and $J^{(K)}_{I_1, I_2, \ldots, I_K}$ is the coupling for the $K$-trace operators. Under the exact renormalization group flow, the scale dependent couplings obey

$$\frac{dJ^{(K)}_{I_1, I_2, \ldots, I_K}}{dz} = -\beta^{(K)}_{I_1, I_2, \ldots, I_K}\left(J^{(1)}, J^{(2)}, \ldots\right), \tag{46}$$

where $\beta^{(K)}_{I_1, I_2, \ldots, I_K}$ is the beta functions. Even if one starts with a simple UV action, all multi-trace operators are generated under the RG flow. Each coupling can be tuned independently at UV, and the exact RG flow is encoded in the beta functions of all couplings defined in the presence of general couplings. Below, we show that the general $\beta$-functions are completely fixed by the $\beta$-functions defined in the subspace with $J^{(K)} = 0$ for $K > 1$ as is the case for the $O(N)$ vector model.

### 2. Quantum RG

As is discussed in the previous section, the space of theories is identified as a Hilbert space. The Hilbert space can be spanned by a set of basis states whose wavefunctions include only single-trace operators. For the matrix model, the basis states are chosen to be

$$|t\rangle = \int \mathcal{D}\phi \; e^{-S_0 + iN^2 \sum_I t_I \mathcal{O}_I} \; |\phi\rangle, \tag{47}$$

where $t_I$ is the source for the single-trace operator $\mathcal{O}_I$. The quantum state associated with the general $O(N)$ invariant action in Eq. (45) can be written as a linear superposition of the basis states as

$$|S\rangle = \int \mathcal{D}t \; \Psi^0_J[t]|t\rangle, \tag{48}$$

where $\Psi^0_J[t]$ is the wave function defined in the space of $t_I$,

$$\Psi^0_J[t] = \int \mathcal{D}p \; \exp\left\{ -iN^2 \sum_I t_I p_I - N^2 \sum_{K=1}^{\infty} \sum_{\{I_1,\ldots,I_K\}} J^{(K)}_{I_1,\ldots,I_K} p_{I_1} \ldots p_{I_K} \right\}. \tag{49}$$

Here $\mathcal{D}t = \prod_I \int_{-\infty}^{\infty} dt_I$ represents the integration over each single-trace coupling along the real axis. $p_I$ represents the field that is conjugate to $t_I$, and $\mathcal{D}p = \prod_I \int_{-\infty}^{\infty} dp_I$. It is straightforward to check that the integration over $t_I$ and $p_I$ reproduce the original action : $|S\rangle = \int \mathcal{D}\phi \; e^{-S} \; |\phi\rangle$.

As is shown in the previous section, the exact real space RG flow is generated by the RG Hamiltonian,

$$\hat{H} = e^{-\hat{S}_0} \sum_i \left[ i \mathrm{tr}(\hat{\phi}_i \hat{\pi}_i) + \frac{1}{Nm^2} \mathrm{tr}(\hat{\pi}_i \hat{\pi}_i^T) \right] e^{\hat{S}_0}, \tag{50}$$

where $\hat{\pi}_i^{ba}$ is the conjugate momentum of $\hat{\phi}_i^{ab}$. In the $|\phi\rangle$ basis, $\hat{\pi}_i^{ba} = -i\frac{\partial}{\partial \phi_i^{ab}}$. The renormalized action at scale $z$ is obtained by $S(z) = -\log\langle\phi|S(z)\rangle$, where $|S(z)\rangle = e^{-z\hat{H}}|S\rangle$. Because the set of basis states $\{|t\rangle\}$ is complete, it is enough to know how each basis state is evolved under $e^{-z\hat{H}}$. Furthermore, $|S(z)\rangle$ can be also written as $|S(z)\rangle = \int \mathcal{D}t \; \Psi^z_J[t]|t\rangle$,

where $\Psi^z_J[t]$ is the wave function at scale $z$. Following the steps used in the previous section, it is straightforward to show that $\Psi^z_J[t]$ is related to $\Psi^0_J[t]$ through the evolution given by

$$\Psi^z_J[t] = e^{-N^2 z \hat{H}_{bulk}} \Psi^0_J[t] \tag{51}$$

where $\hat{H}_{bulk}$ is the induced RG Hamiltonian for the dynamical single-trace couplings $t_I$ and their conjugate momenta $p_I$,

$$\hat{H}_{bulk} = -\beta^{(0)}[-i\hat{t}, 0] - \sum_I \hat{p}_I \beta^{(1)}_I[-i\hat{t}, 0] - \sum_{I_1 I_2} \hat{p}_{I_1} \hat{p}_{I_2} \beta^{(2)}_{I_1 I_2}[-i\hat{t}, 0]. \tag{52}$$

$\beta^{(0)}[J^{(1)}, 0]$, $\beta^{(1)}[J^{(1)}, 0]$, and $\beta^{(2)}[J^{(1)}, 0]$ are the $\beta$-functions for the identity operator, the single-trace operators and the double-trace operators, respectively, in the presence of single-trace couplings $J^{(1)}$ only,

$$\beta^{(0)}[J^{(1)}, 0] = -\sum_i \frac{2}{m^2} J^{(1)}_{(ii)},$$

$$\beta^{(1)}_I[J^{(1)}, 0] = n_I J^{(1)}_I - \frac{2}{m^2} \sum_{I' \in L_{I+2}} J^{(1)}_{I'} - \frac{2}{Nm^2} \sum_{I' \in L'_{I+2}} J^{(1)}_{I'} + \frac{1}{m^2} \sum_{(I', I'') \in L^+_I} J^{(1)}_{I'} J^{(1)}_{I''},$$

$$\beta^{(2)}_{I_1 I_2}[J^{(1)}, 0] = -\frac{2}{m^2} \sum_{I' \in L^-_{I_1, I_2}} J^{(1)}_{I'}. \tag{53}$$

$\hat{t}_I$ and $\hat{p}_{I'}$ obeys the commutation relation, $[\hat{t}_I, \hat{p}_{I'}] = -\frac{i}{N^2} \delta_{I,I'}$, where $\delta_{I,I'}$ denotes the Kronecker delta function defined in the space of loops. $J^{(1)}_{(ii)}$ denotes the source of $\text{tr}(\phi_i \phi_i^T)$. $n_I$ denotes the number of $\phi$ fields in $\mathcal{O}_I$. $L_{I+2}$ denotes the set of loops that can be made by adding two identical sites to $I$ consecutively. $L'_{I+2}$ denotes the set of loops that can be made by adding two identical sites at two even positions or at two odd positions of loop $I$. For $I = (i_1, i_2, i_3, i_4)$,

$$L_{I+2} = \left\{ (j, j, i_1, i_2, i_3, i_4), (i_1, j, j, i_2, i_3, i_4), (i_1, i_2, j, j, i_3, i_4), (i_1, i_2, i_3, j, j, i_4) \,\Big|\, j \in \mathbf{R} \right\},$$

$$L'_{I+2} = \left\{ (j, i_1, j, i_2, i_3, i_4), (i_1, j, i_2, j, i_3, i_4), (i_1, i_2, j, i_3, j, i_4), (i_1, i_2, i_3, j, i_4, j) \,\Big|\, j \in \mathbf{R} \right\},$$

$$\tag{54}$$

where $\mathbf{R}$ represents all possible sites. $L^+_I$ is the set of pairs of loops that can be merged into loop $I$ by removing one common site from each of the two loops. For example, for $I = (i_1, i_2, i_3, i_4)$,

$$L^+_I = \left\{ \Big((i_1, i_2, i_3, j), (j, i_4)\Big), \Big((i_4, i_1, i_2, j), (j, i_3)\Big), \Big((i_3, i_4, i_1, j), (j, i_2)\Big), \Big((i_2, i_3, i_4, j), (j, i_1)\Big), \right.$$

$$\left. \Big((j, i_4), (i_1, i_2, i_3, j)\Big), \Big((j, i_3), (i_4, i_1, i_2, j)\Big), \Big((j, i_2), (i_3, i_4, i_1, j)\Big), \Big((j, i_1), (i_2, i_3, i_4, j)\Big) \Big| j \in \mathbf{R} \right\}. \tag{55}$$

Finally, $L^-_{I_1,I_2}$ denotes the set of loops that can be split into two loops $I_1$ and $I_2$ by removing two identical sites, one at an even position and the other at an odd position. For $I_1 = (i_1, i_2)$ and $I_2 = (i_3, i_4)$, $L^-_{I_1,I_2} = \left\{ (i_1, i_2, j, i_3, i_4, j) | j \in \mathbf{R} \right\}$. In the path integral representation, Eq. (51) can be written as[15, 39]

$$\Psi^z_J[t^z] = \int \mathbf{D}t\mathbf{D}p \ e^{-N^2 S_{bulk}} \ \Psi^0_J[t^0], \tag{56}$$

where $\int \mathbf{D}t\mathbf{D}p = \int \prod_{0 \le z' < z} \mathcal{D}t^{z'} \prod_{0 < z' \le z} \mathcal{D}p^{z'}$ represents the sum over RG paths in the space of single-trace couplings and

$$S_{bulk} = \int_0^z dz' \left[ i \sum_I p^{z'}_I \partial_{z'} t^{z'}_I + H_{bulk}[p^{z'}_I, t^{z'}_I] \right] \tag{57}$$

is the bulk action. The bulk theory, which is fully regularized and well defined, describes dynamics of loop variables in the bulk. Unlike the vector model, the bulk action is quadratic in $p_I$ and the loop variables are genuinely dynamical in the bulk[6]. While the mapping itself is exact at any $N$, the bulk theory becomes weakly interacting only in the large $N$ limit. For general $N$, one has to solve the quantum theory of strongly interacting loops, which is a hard problem. However, one can extract the general constraints that $\beta$-functions obey without solving the full quantum problem for general $N$.

### 3. Full $\beta$-functions

The bulk Hamiltonian is given by the beta functions in the presence of the single-trace couplings only[15, 16]. Because the evolution generated by Eq. (52) contains the full information of the exact RG flow, all beta functions can be extracted from $\beta^{(0)}[-i\hat{t}, 0]$, $\beta^{(1)}[-i\hat{t}, 0]$ and $\beta^{(2)}[-i\hat{t}, 0]$. The change of the effective action under an infinitesimal RG transformation,

$$\delta S = -N^2 dz \sum_K \sum_{\{I_1,...,I_K\}} \beta^{(K)}_{I_1,...,I_K}[J^{(1)}, J^{(2)}, \ldots] \mathcal{O}_{I_1} \mathcal{O}_{I_2}..\mathcal{O}_{I_K}, \tag{58}$$

can be also written as

$$\delta S = -\ln \frac{\int \mathcal{D}t \ \left[ e^{-N^2 dz \hat{H}_{bulk}} \Psi_J[t] \right] e^{iN^2 \sum_I t_I \mathcal{O}_I}}{\int \mathcal{D}t \ \Psi_J[t] e^{iN^2 \sum_I t_I \mathcal{O}_I}} \tag{59}$$

---

[6] There are no higher order terms in $p_I$ because double-trace operators are generated, but triple or higher trace operators are not generated in the subspace of single-trace couplings.

in quantum RG. Equating Eq. (58) and Eq. (59), the general beta functions can be extracted from $\hat{H}_{bulk}$ as

$$
\begin{aligned}
&\beta^{(K)}_{I_1,I_2,\ldots,I_K}\left(J^{(1)}, J^{(2)}, \ldots\right) \\
&= -\frac{1}{K!}\left(\frac{\partial}{\partial \mathcal{O}_{I_1}}\right)\cdots\left(\frac{\partial}{\partial \mathcal{O}_{I_K}}\right)\frac{\int \mathcal{D}t\,\left[\hat{H}_{bulk}\Psi_J[t]\right]e^{iN^2\sum_I t_I \mathcal{O}_I}}{\int \mathcal{D}t\,\Psi_J[t]e^{iN^2\sum_I t_I \mathcal{O}_I}}\Bigg|_{\mathcal{O}=0}.
\end{aligned}
\tag{60}
$$

From

$$
\begin{aligned}
\int \mathcal{D}t\,\left[\hat{H}_{bulk}\Psi_J[t]\right]e^{iN^2\sum_I t_I \mathcal{O}_I} &= \Bigg(-\beta^{(0)}\left[-\frac{1}{N^2}\frac{\partial}{\partial\mathcal{O}},0\right] - \sum_I \mathcal{O}_I\beta^{(1)}_I\left[-\frac{1}{N^2}\frac{\partial}{\partial\mathcal{O}},0\right] \\
&\quad - \sum_{I_1,I_2}\mathcal{O}_{I_1}\mathcal{O}_{I_2}\beta^{(2)}_{I_1,I_2}\left[-\frac{1}{N^2}\frac{\partial}{\partial\mathcal{O}},0\right]\Bigg)e^{-N^2\sum_K\sum_{I_1,\ldots,I_K}J^{(K)}_{I_1,\ldots,I_K}\mathcal{O}_{I_1}\cdots\mathcal{O}_{I_K}},
\end{aligned}
\tag{61}
$$

the general $\beta$-functions are obtained to be

$$
\beta^{(0)} = -\frac{2}{m^2}\sum_i J^{(1)}_{(ii)},
$$

$$
\begin{aligned}
\beta^{(K>0)}_{I_1,\ldots,I_K} &= -(K+1)\frac{2}{m^2}\sum_i J^{(K+1)}_{(ii),I_1,\ldots,I_K} + \left(\sum_{t=1}^{K}n_{I_t}\right)J^{(K)}_{I_1,I_2,\ldots,I_K} - K\frac{2}{m^2}\sum_{I'\in L_{I_1+2}}J^{(K)}_{I',I_2,\ldots,I_K} \\
&\quad - K\frac{2}{Nm^2}\sum_{I'\in L'_{I_1+2}}J^{(K)}_{I',I_2,\ldots,I_K} - K(K+1)\frac{1}{m^2N^2}\sum_{I',I''\in L^+_{I_K}}J^{(K+1)}_{I',I'',I_1,\ldots,I_{K-1}} \\
&\quad + \frac{1}{m^2}\sum_{I',I''\in L^+_{I_K}}\sum_{M=1}^{K}(K-M+1)MJ^{(K-M+1)}_{I'',I_1,\ldots,I_{K-M}}J^{(M)}_{I',I_{K-M+1},\ldots,I_{K-1}} \\
&\quad - (K-1)\frac{2}{m^2}\sum_{I'\in L^-_{I_1 I_2}}J^{(K-1)}_{I',I_3,\ldots,I_K}.
\end{aligned}
\tag{62}
$$

It is noted that the full $\beta$-functions in Eq. (62) are entirely determined from $\beta^{(0)}[-i\hat{t},0]$, $\beta^{(1)}[-i\hat{t},0]$ and $\beta^{(2)}[-i\hat{t},0]$.

We close this section with a few remarks. First, the $\beta$-functions in Eq. (42) and Eq. (62), which have been derived from the $\beta$-functions defined on the subspace of single-trace couplings, are exact for any $N$. The validity of the mapping from the exact Wilsonian RG to quantum RG and the constraints that are derived from it do not require that the bulk theory is in the semi-classical limit. Second, the constraints can be extended to general theories because the notion of single-trace operators can be defined in any theory. This follows from the fact that the space of theories can be in general viewed as a Hilbert space,

where an action $S[\phi]$ for fundamental field $\phi$ defines a wavefunction $e^{-S[\phi]}$ in the Hilbert space. Furthermore, there exists a set of basis states that span the Hilbert space. In general, there exist multiple choices of basis states. The basis states do not need to be orthogonal, and an over-complete set is an acceptable choice. All we need for quantum RG is one choice of complete basis states [7]. For the vector model and the matrix model, the complete set of basis states are given in Eq. (24) and Eq. (47), respectively. Once a complete set of basis is chosen, the wavefunctions of the basis states define a set of actions. The operators that are needed to construct the wavefunctions of the basis states define the set of single-trace operators in general theories. They are given in Eq. (23) and Eq. (43) for the vector model and the matrix model, respectively. Because the Hilbert space structure and the basis states can be defined in any theory, there exist constraints among $\beta$-functions in general theories. The generalization is discussed in the next section. Finally, the full $\beta$-functions could have been computed directly from the exact Polchinski RG equation in Eq. (15). The salient point of our paper is not the exact $\beta$-functions itself but the fact that the entire $\beta$-functions are fully characterized by a small set of data defined in the subspace of single-trace couplings. As a result, it is impossible to change a theory or RG scheme such that the $\beta$-functions away from the subspace of single-trace couplings are modified without modifying $\beta$-functions in the subspace. Because multi-trace operators are composites of the single-trace operators, the RG flow in the presence of general multi-trace operators are completely fixed by the $\beta$-functions defined in the subspace of single-trace couplings. This constraint holds even when multi-trace operators have large anomalous dimensions.

## III. GENERALIZATION

In this section, we generalize the results obtained for the two concrete models in the previous section. Let us consider a field theory in the $D$-dimensional Euclidean space with the partition function,

$$Z = \int \mathcal{D}\phi \; e^{-S[\phi]}, \tag{63}$$

where $\phi(x)$ represents a set of fundamental fields and $S$ is an action that is invariant under a symmetry group $G$. The RG flow is defined in the space of local theories in a given symmetry sector. To describe the RG flow, one first needs to coordinatize the space of theories. For this, we divide $S$ into a reference action $S_0$ and a deformation,

$$S = S_0[\phi] + \sum_M \int d^D x \; J_M(x)\mathbf{O}_M(x). \tag{64}$$

---

[7] It may well be the case that one choice of basis states gives a simpler bulk theory than others.

Here the reference action $S_0$ sets the origin in the space of theories. $\{\mathbf{O}_M(x)\}$ represents the complete set of local operators that are invariant under the symmetry $G$. We call these operators 'symmetry-allowed operators'. $J_M(x)$ is the coupling function that deforms the reference theory. One infinitesimal step of coarse graining consists of integrating out fast modes of the fundamental fields, and rescaling the space and fields. The one cycle of coarse graining puts the theory into the same form as before except for renormalized sources,

$$Z = \int \mathcal{D}\phi \; e^{-S_0[\phi] - \sum_M \int d^D x \left( J_M(x) - \beta_M[J;x] dz \right) \mathbf{O}_M(x)}, \tag{65}$$

where $dz$ is an infinitesimal parameter and $\beta_M[J;x]$ is the beta function for the local operator $\mathbf{O}_M(x)$. $\beta_M[J;x]$ is a function of $x$ and a functional of coupling functions $J_N(x)$. In weakly coupled field theories, one may ignore operators whose couplings remain small in the perturbative series. In general, one has to keep all couplings. Successive applications of the coarse graining give rise to the exact Wilsonian renormalization group (RG) flow in the infinite dimensional space of couplings[6].

Alternatively, the RG flow can be projected to a subspace of couplings at the price of promoting the deterministic RG flow to a path integration over RG paths (quantum RG) within the subspace[15, 16]. To see this, we define a quantum state from the action by promoting the Boltzmann weight to a wave function as in Eq. (19) [30]. This correspondence between a $D$-dimensional action and a $D$-dimensional quantum state is not the same as the correspondence between a $D$-dimensional action and the ground state defined on a $(D-1)$-dimensional slice with a fixed imaginary time. As we pointed out in Eq. (18), the partition function in Eq. (63) can be written as an overlap between two states, $Z = \langle \mathbb{1}|S \rangle$, where $|\mathbb{1}\rangle$ is defined in Eq. (20), represents the trivial fixed point with zero correlation length. In this picture, one infinitesimal step of coarse graining is generated by a quantum operator inserted between the overlap of $\langle \mathbb{1}|$ and $|S\rangle$. Here, we rewrite the Eq. (21) for convenience,

$$Z = \langle \mathbb{1}|e^{-dzH}|S\rangle, \tag{66}$$

where $H$ is the RG Hamiltonian that generates the coarse graining transformation that satisfies $\langle \mathbb{1}|H = 0$[30]. A concrete example of RG Hamiltonian is discussed in Sec. II. Since $|\mathbb{1}\rangle$ is invariant under the evolution generated by $H$, the partition function remains the same under the insertion of the operator. Nonetheless, $H$ generates a non-trivial evolution of $|S\rangle$ once it is applied to the right in Eq. (66) [8].

The one-to-one correspondence between states and actions implies that the resulting state corresponds to a renormalized action,

$$e^{-dzH}|S\rangle = |S + \delta S\rangle, \tag{67}$$

---

[8] In more generality, one may choose $|S_*\rangle = \int \mathcal{D}\phi \; e^{-S_*}|\phi\rangle$ with a different fixed point action $S_*$ instead of $|\mathbb{1}\rangle$[30]. In this case, the partition function is written as $Z = \langle S_*|S_1\rangle$, where $S_1 = S - S_*$ is the deformation measured with respect to $S_*$. In this case, the coarse graining Hamiltonian that satisfies $\langle S_*|H' = 0$ is related to $H$ through a similarity transformation, $H' = e^{S_*} H e^{-S_*}$.

where

$$\delta S = -dz \sum_M \int dx \beta_M[J;x]\mathbf{O}_M(x). \tag{68}$$

Successive applications of the coarse graining transformations give rise to a scale dependent quantum state which corresponds to the scale dependent Wilsonian action,

$$e^{-zH}|S\rangle = \int \mathcal{D}\phi \ e^{-S_0[\phi] - \sum_M \int d^D x J_M(x,z)\mathbf{O}_M(x)}|\phi\rangle, \tag{69}$$

where $J_M(x,z)$ is the renormalized coupling function that satisfies

$$\frac{\partial J_M(x,z)}{\partial z} = -\beta_M[J;x] \tag{70}$$

with the initial condition $J_M(x,0) = J_M(x)$. In Eq. (69), $J_M(x,z)$'s are classical parameters that keep track of the exact Wilsonian RG flow. However, Eq. (70) is a rather inefficient way of keeping track of the evolution of quantum state in that the number of classical parameters one needs to keep is far greater than the number of linearly independent quantum states.

Once we realize that the space of theories can be viewed as a vector space, a more natural description of the RG flow is to take advantage of the linear superposition principle. Instead of labeling a quantum state in terms of classical parameters, a quantum state is expressed as a linear superposition of basis states, which as a set is much smaller than $\{J_M(x)\}$. A complete set of basis states can be chosen to be

$$|t\rangle = \int \mathcal{D}\phi \ e^{-S_0[\phi] + i \sum_m \int d^D x \ t_m(x)\mathcal{O}_m(x)}|\phi\rangle, \tag{71}$$

where $\{\mathcal{O}_m(x)\}$ is a subset of symmetry-allowed local operators from which all symmetry-allowed local operators can be written as composites,

$$\mathbf{O}_M(x) = \mathcal{O}_{m_1}(x)\left[\partial_{\mu_1}..\partial_{\mu_{l_1}}\mathcal{O}_{m_2}(x)\right]\left[\partial_{\mu_1}..\partial_{\mu_{l_2}}\mathcal{O}_{m_3}(x)\right]..\left[\partial_{\mu_1}..\partial_{\mu_{l_{k-1}}}\mathcal{O}_{m_k}(x)\right]. \tag{72}$$

We call this subset of operators single-trace operators because they are the set of operators that involve one trace in large $N$ matrix models.

In the example of the scalar field theory with the $Z_2$ symmetry, the single-trace operators are given by the set of quadratic operators, $\mathcal{O}_{\mu_1,\mu_2,..,\mu_n}(x) = \phi(x)\partial_{\mu_1}\partial_{\mu_2}..\partial_{\mu_n}\phi(x)$. Other $Z_2$ invariant operators that are quartic or higher order in $\phi$ can be written as composites of the single-trace operators, and they are called multi-trace operators. It is noted that the distinction between single-trace and multi-trace operators depends not only on the field content of the theory but also on the symmetry. In the absence of the $Z_2$ symmetry, the fundamental field $\phi$ is the only single-trace operator, and everything else is regarded as multi-trace operator. We emphasize that operators can be divided into single-trace operators and multi-trace operators in any theory.

It is straightforward to see that Eq. (71) forms a complete basis, and Eq. (19) with Eq. (64) can be written as [15, 16]

$$|S\rangle = \int \mathcal{D}t \ \Psi_J[t]|t\rangle, \tag{73}$$

where $\Psi_J[t]$ is a wavefunction defined in the space of the single-trace sources. In the next section, we will provide a prescription to find $\Psi_J[t]$ from a general action. The measure and the integration path of the sources in Eq. (73) will be carefully defined when we discuss examples in the following sections. The state for a general theory with multi-trace operators can be written as a linear superposition of those whose wavefunctions only include single-trace operators. The multi-trace operators that are not explicitly included in the description endows the single-trace couplings with quantum fluctuations.

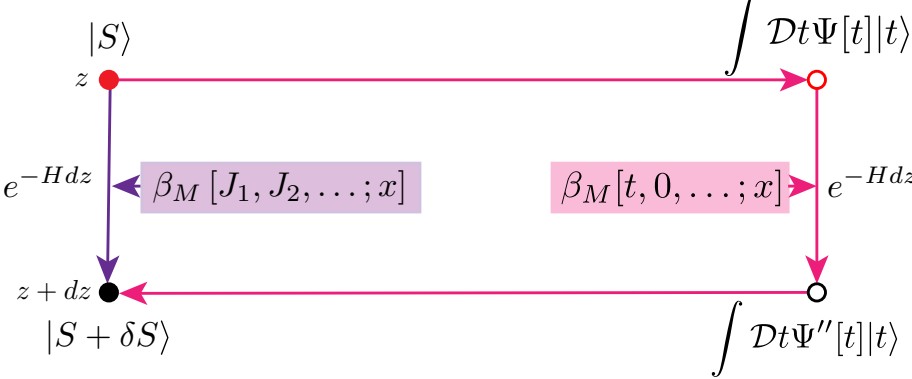

FIG. 2. A field theory $S$ at scale $z$ is represented by a quantum state $|S\rangle$ on the upper left corner. The vertical arrows represent an infinitesimal step of coarse graining generated by $H$. The coarse graining directly applied to $|S\rangle$, denoted as the vertical arrow on the left, maps $|S\rangle$ to $|S+\delta S\rangle$. $S+\delta S$ is the renormalized action at scale $z + dz$, and $\delta S$ encodes the information on the full $\beta$-functions. Alternatively, $|S\rangle$ is first written as a linear superposition of basis states whose action only includes single-trace deformations through the horizontal arrow that points to the right. The result of the coarse graining applied to that state, denoted as the vertical arrow on the right, only depends on the $\beta$ functions defined in the space of single-trace couplings. The resulting coarse grained state on the lower right corner is finally mapped back to a renormalized action on the lower left corner. The commutativity of the diagram implies the full $\beta$-functions are fixed by the $\beta$-functions defined in the subspace of single-trace couplings.

Because $H$ is a linear operator, the RG flow in Eq. (67) is entirely fixed by how the coarse graining operator acts on the basis states,

$$e^{-dzH}|S\rangle = \int \mathcal{D}t \ \Psi_J[t] \, e^{-dzH}|t\rangle. \tag{74}$$

To figure out the resulting state, we simply use the expression in Eq. (69) by turning off all multi-trace sources. From Eq. (71), we obtain

$$e^{-dzH}|t\rangle = \int \mathcal{D}\phi \; e^{-S_0[\phi]+i\sum_n \int d^D x \; t_n(x)\mathcal{O}_m(x) - dz \sum_M \int d^D x \beta_M[-it,0,..;x]\mathbf{O}_M(x)}|\phi\rangle. \quad (75)$$

Here the beta functions are expressed as $\beta_M[J;x] = \beta_M[-it, -it^{[2]}, ..; x]$, where $-it$ is the single-trace coupling and $-it^{[k]}$'s with $k \geq 2$ are the couplings for $k$-trace operators which are composites of $k$ single-trace operators. $\beta_M[-it, 0, ..; x]$ is the beta function defined in the subspace of single-trace sources. Because $\{|t\rangle\}$ is complete, Eq. (75) can in turn be expressed as a linear superposition of $|t\rangle$ as

$$e^{-dzH}|t\rangle = \int \mathcal{D}t' \; \Phi_{dz}\left[t; t'\right]|t'\rangle, \quad (76)$$

where $\Phi_{dz}\left[t; t'\right]$ is a propagator of the RG transformation that is determined from $\beta_M[-it, 0, ..; x]$. This allows us to write the resulting state after a coarse graining transformation as

$$e^{-dzH}|S\rangle = \int \mathcal{D}t' \Psi''\left[t'\right]|t'\rangle, \quad (77)$$

where

$$\Psi''\left[t'\right] = \int \mathcal{D}t \; \Psi_J\left[t\right]\Phi_{dz}\left[t; t'\right]. \quad (78)$$

In the end, the RG transformation leads to the evolution of wave function, $\Psi'' = e^{-\mathcal{H}dz}\Psi$, where $\mathcal{H}$ is an induced coarse graining operator defined by $\mathcal{H}\Psi[t'] = -\frac{1}{dz}\left(\int \mathcal{D}t\Psi\left[t\right]\Phi_{dz}\left[t; t'\right] - \Psi[t']\right)$. By equating Eq. (74) with Eq. (67), we obtain

$$e^{-dzH}|S\rangle = \int \mathcal{D}\phi \; e^{-S_0}\left[\int \mathcal{D}t'(x)\Psi''\left[t'\right]e^{i\sum_n \int d^D x \; t'_n(x)\mathcal{O}_m(x)}\right]|\phi\rangle$$

$$= \int \mathcal{D}\phi e^{-S_0 - \sum_M \int dx \left(J_M(x) - dz\beta_M[J;x]\right)\mathbf{O}_M(x)}|\phi\rangle. \quad (79)$$

This shows that

$$\int \mathcal{D}t'\mathcal{D}t\Psi_J\left[t\right]\Phi_{dz}\left[t; t'\right]e^{i\sum_n \int d^D x \; t'_n(x)\mathcal{O}_m(x)} = e^{-\sum_M \int dx \left(J_M(x) - dz\beta_M[J;x]\right)\mathbf{O}_M(x)}. \quad (80)$$

Eq. (80) is the main result of the paper. On the left hand side of Eq. (80), $\Psi_J[t]$ is fixed by the theory at scale $z$ through Eq. (73), and $\Phi_{dz}\left[t; t'\right]$ is entirely determined from $\beta_M[-it, 0, ..; x]$ through Eqs. (75) and (76). This, in turn, fixes the full beta functions $\beta_M[J;x]$ through Eq. (80). *Therefore, the beta functions defined in the subspace of the single-trace operators completely fix the full beta functions away from the subspace.* This is illustrated in Fig. 2.

## IV.   QUANTUM RENORMALIZATION GROUP

In this section, we lay out an algorithm for extracting general scaling operators, scaling dimensions and operator product expansion coefficients from $\beta$-functions defined in the subspace of single-trace operators.

### A.   Action-state correspondence

To be concrete, we consider a partition function given by

$$Z\left[J_1, J_2, \ldots\right] = \langle \mathbb{1} | S_{J_1, J_2, \ldots} \rangle, \tag{81}$$

where

$$|S_{J_1, J_2, \ldots}\rangle = \int \mathcal{D}\phi \; e^{-S_0 - \int d^D x \left( \sum_n J_n(x) \mathcal{O}^n[\phi(x)] + \ldots \right)} |\phi\rangle. \tag{82}$$

Here $S_0$ is the reference action. $\mathcal{O}(x)$ is a real and local single-trace operator. $\mathcal{O}^n(x)$ with $n > 1$ represents local multi-trace operators. The ellipsis includes multi-trace operators with derivatives as is shown Eq. (72). We assume that the deformation is bounded from below, and the highest multi-trace operator is an even power of $\mathcal{O}$ with a positive coupling. We can remove the multi-trace operator in the action by using an identity,

$$
\begin{aligned}
f(\mathcal{O}(y)) &= \int_R \mathcal{D}t' \mathcal{D}p' \; e^{-i \int d^D x \; t'(x)[p'(x) - \mathcal{O}(x)]} f(p'(y)) \\
&= \int_I \mathcal{D}t \mathcal{D}p \; e^{-i \int d^D x \; t(x)[p(x) - i\mathcal{O}(x)]} f(-ip(y)),
\end{aligned}
\tag{83}
$$

for any $f(x)$. The integration of $t'(x)$ and $p'(x)$ are defined along the real axes. In the second line, we define $t(x) = -it'(x)$ and $p(x) = ip'(x)$ so that the integration for $t(x)$ and $p(x)$ are defined along the imaginary axes[9]. The path of the integration variables is denoted by the subscripts $R$ (real) and $I$ (imaginary). Then, Eq. (82) can be rewritten as

$$
\begin{aligned}
|S_{J_1, J_2, \ldots}\rangle &= \int \mathcal{D}\phi \; e^{-S_0} \int_I \mathcal{D}t \mathcal{D}p \; e^{-i \int d^D x t(x)[p(x) - i\mathcal{O}(x)]} e^{-\int d^D x \left( \sum_n J_n(x)[-ip(x)]^n + \ldots \right)} |\phi\rangle \\
&= \int_I \mathcal{D}t \mathcal{D}p \; e^{-\int d^D x \left( it(x)p(x) + \sum_n (-i)^n J_n(x) p^n(x) + \ldots \right)} \left[ \int \mathcal{D}\phi \; e^{-S_0 - \int d^D x \; t(x)\mathcal{O}(x)} |\phi\rangle \right].
\end{aligned}
\tag{84}
$$

---

[9] This Wick's rotation has the advantage that the source for $\mathcal{O}$ is simply represented by $t(x)$.

This implies that the state can be written as a linear superposition of the basis states as

$$|S_{J_1,J_2,\dots}\rangle = \int_I \mathcal{D}t \; \Psi_{J_1,J_2,\dots}[t] \, |t\rangle, \tag{85}$$

where

$$|t\rangle = \int \mathcal{D}\phi \; |\phi\rangle T_{\phi,t} \tag{86}$$

is the complete basis state whose wavefunction is made of the reference action and the single-trace deformation only,

$$T_{\phi,t} = e^{-S_0 - \int d^D x \; t(x)\mathcal{O}(x)}, \tag{87}$$

and

$$\Psi_{J_1,J_2,\dots}[t] = \int_I \mathcal{D}p \; e^{-\int d^D x \left( it(x)p(x) + \sum_n (-i)^n J_n(x)p^n(x) + \dots \right)} \tag{88}$$

is the wavefunction of the dynamical single-trace source. The integration over $p$ in Eq. (88) is convergent for deformations that are bounded from below. This allows us to write the original partition function in terms of partition functions that involve only single-trace deformations, $Z[J_1, J_2, \dots] = \int_I \mathcal{D}t \; \Psi_{J_1,J_2,\dots}[t] \, Z[t, 0, \dots]$ with $\langle \mathbb{1}|t\rangle = Z[t, 0, \dots]$.

Let us first consider a simple example with an ultra-local deformation (no derivative terms) with $J_{n>2} = 0$. In this case, the wave function is Gaussian,

$$\Psi_{J_1,J_2}[t] = \int_I \mathcal{D}p \; e^{-\int d^D x \left( itp - iJ_1 p - J_2 p^2 \right)} = \left[ \prod_x \sqrt{\frac{\pi}{J_2(x)}} \right] e^{\int d^D x \frac{[t(x) - J_1(x)]^2}{4 J_2(x)}}. \tag{89}$$

We note that $t$ in Eq. (89) is to be integrated along the imaginary axis, and the wavefunction is normalizable if $J_2 > 0$. This shows how both single-trace and multi-trace couplings are encoded in the wavefunction :

$$\langle \Psi_{J_1,J_2}|t(x)|\Psi_{J_1,J_2}\rangle = J_1(x),$$
$$\langle \Psi_{J_1,J_2}|t^2(x)|\Psi_{J_1,J_2}\rangle - \langle \Psi_{J_1,J_2}|t(x)|\Psi_{J_1,J_2}\rangle^2 = -2J_2(x). \tag{90}$$

The expectation value of $t(x)$ gives the single-trace coupling, and the second cumulant gives the double-trace coupling. The second cumulant is negative because $t(x)$ fluctuates along the imaginary axis.

If double-trace operators have a support over a finite region, the action becomes

$$S' = S_0 + \int d^D x \Big( J_1(x)\mathcal{O}(x) + \int d^D x' J_2(x - x')\mathcal{O}(x)\mathcal{O}(x') \Big), \tag{91}$$

where $J_2(x - x')$ is the source for the bi-local double-trace operator. The corresponding wave function is written as

$$\Psi_{J_1,J_2}[t] = \int_I \mathcal{D}p(x) e^{-\int d^D x \left( it(x)p(x) - iJ_1(x)p(x) - \int d^D x' J_2(x-x')p(x)p(x') \right)}$$

$$= \sqrt{\det\left[\pi J_2^{-1}(x - x')\right]} e^{\frac{1}{4}\int d^D x \int d^D x'[t(x)-J_1(x)]J_2^{-1}(x-x')[t(x')-J_1(x')]}. \tag{92}$$

The non-zero correlation length for $j$ gives

$$\langle t(x)t(x')\rangle - \langle t(x)\rangle\langle t(x')\rangle = -2J_2(x - x'). \tag{93}$$

This shows that the correlation between fluctuating single-trace sources encodes the information on how the source for the bi-local double-trace operator decays in space. In general, all multi-trace couplings can be extracted from higher order cumulants. It is noted that the single-trace coupling has non-trivial quantum fluctuations only in the presence of multi-trace couplings[10] Moreover, Eq. (82) can be written as

$$|S_{J_1,J_2,\dots}\rangle = \int \mathcal{D}\phi\, \mathcal{W}_\phi[J_1, J_2, \dots]\,|\phi\rangle, \tag{94}$$

where

$$\mathcal{W}_\phi[J_1, J_2, \dots] = \int_I \mathcal{D}t\, T_{\phi,t}\Psi_{J_1,J_2,\dots}[t]. \tag{95}$$

We denote the vector spaces formed by $\{\mathcal{W}_\phi\}$ and $\{\Psi_J[t]\}$ as $W$ and $V$, respectively. $W$ is the space of Boltzmann weights within a given symmetry sector with inner product, $(\mathcal{W}, \mathcal{W}') = \int \mathcal{D}\phi\, \mathcal{W}_\phi^* \mathcal{W}'_\phi$. $V$ is the space of wavefunctions of the single-trace sources with inner product, $(\Psi, \Psi') = \int \mathcal{D}t\mathcal{D}t'\, \Psi^*[t]\langle t|t'\rangle\Psi'[t']$. Eq. (95) provides a bijective map, $\mathcal{T}: V \to W$. Accordingly, for every linear operator $\hat{\mathcal{A}}_\phi$ that acts on $W$, there exists a linear operator $\hat{\mathcal{A}}_j$ that acts on $V$ such that

$$(\hat{\mathcal{A}}_\phi \mathcal{W})_\phi[J_1, J_2, \dots] = \int_I \mathcal{D}t\, \hat{T}_{\phi,t}(\hat{\mathcal{A}}_t \Psi)_{J_1,J_2,\dots}[t]. \tag{96}$$

In order to find the correspondence between $\hat{\mathcal{A}}_\phi$ and $\hat{\mathcal{A}}_t$, we consider the detailed form of $\Psi$ in Eq. (88). $\hat{\mathcal{A}}_t$, generated by $\frac{\delta}{\delta t}$ and $t$, gives the relations

$$\frac{\delta}{\delta t(x)}\Psi_{J_1,\dots}[t] = \int_I \mathcal{D}p\, (-i)p(x)e^{-\int d^D x \left( it(x)p(x) + \sum_n (-i)^n J_n(x)p^n(x) \right)},$$

$$t(x)\Psi_{J_1,\dots}[t] = \int_I \mathcal{D}p\left( -i\frac{\delta}{\delta p(x)} e^{-\int d^D x \sum_n (-i)^n J_n(x)p^n(x)} \right) e^{-i\int d^D x\, t(x)p(x)}. \tag{97}$$

_______

[10] In the context of the holographic renormalization group, this amounts to the fact that integrating out bulk degrees of freedom generates multi-trace operators at a new UV boundary [26, 40].

Since $p(x)$ is already identified as $i\mathcal{O}(x)$ in Eq. (84), $t(x)$ and $\frac{\delta}{\delta t(x)}$ acting on $V$ correspond to following operator acting on $W$ [11].

$$\frac{\delta}{\delta t(x)} \Leftrightarrow \mathcal{O}(x),$$

$$t(x) \Leftrightarrow -e^{-S_0} \frac{\delta}{\delta \mathcal{O}(x)} e^{S_0}. \tag{98}$$

Therefore, we obtain $\hat{\mathcal{A}}_t \left[ t, \frac{\delta}{\delta t} \right] \Leftrightarrow \hat{\mathcal{A}}_\phi \left[ -e^{-S_0} \frac{\delta}{\delta \mathcal{O}} e^{S_0}, \mathcal{O} \right]$.

## B. RG flow as quantum evolution

Identifying the space of theories as a vector space naturally leads to the quantum RG[16]. We first consider the field theory in Eq. (81) defined in a finite box with the linear system size $1/\lambda$, where $\lambda$ corresponds to an IR cutoff energy scale. From Eq. (84), the partition function is equivalent to

$$Z_\lambda \left[ J_1, J_2, \dots \right] = \int_I \mathcal{D}t^{(0)} \ \Psi_{J_1, J_2, \dots} \left[ t^{(0)} \right] \ Z_\lambda \left[ t^{(0)}, 0, \dots \right], \tag{99}$$

where $\Psi_{J_1, J_2, \dots}[t^{(0)}]$ is the wavefunction defined in Eq. (88), and $Z_\lambda \left[ t^{(0)}, 0, \dots \right] = \langle \mathbb{1} | t^{(0)} \rangle = \int \mathcal{D}\phi \ e^{-S_0 - \int^{1/\lambda} d^D x \ t^{(0)}(x) \mathcal{O}(x)}$. The subscript $\lambda$ of $Z$ keeps track of the IR cutoff. Next we perform a coarse graining transformation on $Z_\lambda \left[ t^{(0)}, 0, \dots \right]$ as is discussed in the previous section : integrating out high-energy modes of $\phi$ which reduces the UV cutoff from $\Lambda$ to $\Lambda' = b\Lambda$ with $b = e^{-dz} < 1$. In general, not only the single-trace source is renormalized but also multi-trace operators are generated. Under rescaling of the field and the space, $\phi'(x') = b^{-\Delta_\phi}\phi(x)$ with $x' = bx$, $\Lambda'$ is brought back to $\Lambda$, but the system size decreases as $1/\lambda' = b/\lambda$. The resulting partition function becomes[12]

$$Z_\lambda \left[ t^{(0)}, 0, \dots \right] = \int \mathcal{D}\phi \ e^{-S_0[\phi]} e^{-\int^{b/\lambda} d^D x' \left\{ t^{(0)}(x') \mathcal{O}(x') - dz \sum_{n \geq 0} \beta_n \left[ t^{(0)}, 0, \dots \right] \mathcal{O}^n(x') \right\}}. \tag{100}$$

The change of the coupling for $\mathcal{O}^n(x)$ with increasing length scale is given by $-\beta_n dz$. $-\beta_0 dz$ corresponds to the free energy contributed from the high-energy modes that are integrated out in the infinitesimal coarse graining step. In general, the $\beta$-functions depend on all the

---

[11] We can explicitly derive this correspondence as

$$\int_I \mathcal{D}t \left( \frac{\delta}{\delta t} \Psi_{J_1, J_2, \dots} [t] \right) |t\rangle = -\int_I \mathcal{D}t \ \Psi_{J_1, J_2, \dots} [t] \frac{\delta}{\delta t} |t\rangle$$

$$= -\int_I \mathcal{D}t \ \Psi_{J_1, J_2, \dots} [t] \int \mathcal{D}\phi \frac{\delta}{\delta t} e^{-S_0 - \int d^D x \ t(x) \mathcal{O}(x)} |\phi\rangle = \int \mathcal{D}\phi(x) \left( \mathcal{O} \mathcal{W}_\phi \right) |\phi\rangle,$$

$$\int_I \mathcal{D}t \left( t(x) \Psi_{J_1, J_2, \dots} [t] \right) |t\rangle = \int_I \mathcal{D}t \int \mathcal{D}\phi \ \Psi_{J_1, J_2, \dots} [t] \ t(x) e^{-S_0 - \int d^D x \ t(x) \mathcal{O}(x)} |\phi\rangle$$

$$= \int \mathcal{D}\phi \left( -e^{-S_0} \frac{\delta}{\delta \mathcal{O}} e^{S_0} \mathcal{W}_\phi \right) [J_1, J_2, \dots] |\phi\rangle.$$

[12] Here we omit an additional subscript $\lambda/b$ in $S_0$ to avoid clutter in notation.

couplings $J_n$. However, what enters in Eq. (100) are the beta functions in the subspace of single-trace couplings only. From Eq. (84), the partition function in Eq. (100) can be written as

$$
Z_\lambda \left[ t^{(0)}, 0, ... \right]
$$
$$
= \int_I \mathcal{D} \, t^{(1)}(x) \mathcal{D} p^{(1)}(x) e^{- \int^{b/\lambda} d^D x \left[ ip^{(1)} \left( t^{(1)} - t^{(0)} \right) - dz \sum_n \beta_n dz (-ip^{(1)})^n \right]} Z_{b^{-1}\lambda} \left[ t^{(1)}, 0, \dots \right].
$$
(101)

After $M$ steps of coarse graining, we take the $dz \to 0$ limit, keeping $M dz = z^*$ fixed. This results in

$$
Z_\lambda [J_1, J_2, \dots] = \int_I \mathcal{D}t(x,z) \mathcal{D}p(x,z) \Psi_{J_1, J_2, \dots} [t(0)] e^{- \int_0^{z^*} dz L[t,p,z]} Z_{e^{z^*}\lambda} [t(z^*), 0, \dots],
$$
(102)

where $z$ is the extra direction in the bulk that labels the logarithmic length scale. The bulk Lagrangian is given by

$$
L[t,p,z] = \int^{e^{-z}/\lambda} d^D x \left( ip(x,z) \partial_z t(x,z) - \sum_{n \geq 0} \beta_n [t(z), 0, ..; x] \, (-ip(x,z))^n \right). \quad (103)
$$

In Eq. (102), the RG flow of the $D$-dimensional field theory is replaced with a $(D+1)$-dimensional path integration of the dynamical single-trace sources[15, 16]. The fluctuations of the RG paths encode the information of the multi-trace operators. The sum over all possible RG paths within the subspace of the single-trace sources is weighted with the $(D+1)$-dimensional bulk action. Equivalently, the RG flow is described by the quantum evolution of the wavefunction of the single-trace source. In the Hamiltonian picture, $t(x)$ and $p(x)$ are canonical conjugate operators that satisfy the commutation relation $[t(x), p(x')] = -i\delta(x-x')$. The RG evolution can be viewed as an imaginary time evolution, $\mathcal{R}(z+dz, z) = e^{-\mathcal{H}^\lambda[t,p,z]dz}$ generated by the bulk RG Hamiltonian,

$$
\mathcal{H}^\lambda [t,p,z] = - \sum_{n \geq 0} \int^{e^{-z}/\lambda} d^D x \, \beta_n [t, 0, ..; x] \, (-ip(x))^n, \quad (104)
$$

where the superscript $\lambda$ represents the IR cutoff scale associated with the finite system size. $\mathcal{H}^\lambda$ depends on $z$ explicitly through $\lambda$ as the system size decreases with increasing $z$. In the thermodynamic limit ($\lambda = 0$), $\mathcal{H}^0$ is independent of $z$. Being an operator that acts on wavefunctions defined in the space of single-trace sources, $\mathcal{H}^\lambda$ generates the quantum evolution of the state associated with the RG flow. The RG Hamiltonian is fixed by the $\beta$-functions within the subspace of the single-trace sources only.

## C. Reconstruction of the Wilsonian RG from the quantum RG

In this section, we explain how the full $\beta$-functions of a field theory can be reconstructed from the quantum evolution with the RG Hamiltonian, although the latter is fixed by the $\beta$-functions within the subspace of the single-trace couplings. Suppose that there exists a unique IR fixed point in the thermodynamic limit. The fixed point action is invariant under the RG transformation, and the corresponding quantum state should be an eigenstate of $\mathcal{H}^\lambda$ at $\lambda = 0$. Furthermore, the stable IR fixed point must correspond to the ground state of $\mathcal{H}^0$ because generic initial state is always projected onto it in the large $z$ limit. More generally, one can consider excited states of $\mathcal{H}^0$. States of particular interest are eigenstates that support excitations local in space. Those states correspond to the IR fixed point perturbed with local operators with definite scaling dimensions. These scaling dimensions are given by the energy differences between the excited states and the ground state. In the rest of this section, we establish the correspondence between the ground state (excited states) and the stable fixed point (the stable fixed point with operator insertions).

### 1. Stable IR fixed point as the bulk ground state

We begin with the discussion of the ground state. The ground state of $\mathcal{H}^0$ satisfies

$$\mathcal{H}^0 \psi_0 [t] = \mathcal{E}_0 \psi_0 [t] \tag{105}$$

with the lowest eigenvalue. The partition function for the the IR fixed point is given by

$$Z^* [J_1^*, J_2^*, \dots] = \int_I \mathcal{D}t \; \psi_0 [t] \int \mathcal{D}\phi \; e^{-S_0 - \int d^D x \; t(x)\mathcal{O}(x)}, \tag{106}$$

where $\mathcal{O}$ is the single-trace operator. To extract the multi-trace couplings at the fixed point, we use the cumulant expansion $\langle e^{-\Omega} \rangle = e^{-\langle\Omega\rangle + \frac{1}{2}(\langle\Omega^2\rangle - \langle\Omega\rangle^2) + \cdots}$ to rewrite Eq. (106) as

$$
\begin{aligned}
Z^* [J_1^*, J_2^*, \dots] &= \int \mathcal{D}\phi \; e^{-S_0} \langle e^{-\int d^D x \; t(x)\mathcal{O}(x)} \rangle_{\psi_0} \\
&= \int \mathcal{D}\phi \; e^{-S_0} e^{-\int d^D x \langle t(x)\rangle_{\psi_0} \mathcal{O}(x) + \frac{1}{2}\int d^D x d^D y (\langle t(x)t(y)\rangle_{\psi_0} - \langle t(x)\rangle_{\psi_0}\langle t(y)\rangle_{\psi_0})\mathcal{O}(x)\mathcal{O}(y) + \dots},
\end{aligned}
\tag{107}
$$

where $\langle F[t] \rangle_{\psi_0} \equiv \int_I \mathcal{D}t \; \psi_0 [t] \, F[t]$. Identifying this as the fixed point action,

$$Z^* [J_1^*, J_2^*, \dots] = \int \mathcal{D}\phi \; e^{-S_0 - \int d^D x J_1^*(x)\mathcal{O}(x) - \int d^D x d^D y J_2^*(x-y)\mathcal{O}(x)\mathcal{O}(y) - \cdots}, \tag{108}$$

we obtain the couplings at the fixed point,

$$
\begin{aligned}
J_1^*(x) &= \langle t(x)\rangle_{\psi_0} \\
J_2^*(x-y) &= -\frac{1}{2}\Big( \langle t(x)t(y)\rangle_{\psi_0} - \langle t(x)\rangle_{\psi_0}\langle t(y)\rangle_{\psi_0} \Big).
\end{aligned}
\tag{109}
$$

Sources for higher-trace operators at the fixed point are given by the higher cumulants.

### 2. Scaling operators as local excitations of the bulk theory

Next, let us study excited states of $\mathcal{H}^0$. We start by considering states with local excitations. The $n$-th excited state that supports a local excitation at $x$ satisfies

$$e^{-\mathcal{H}^0 z}\psi_{n,x} = e^{-\mathcal{E}_n z}\psi_{n,e^{-z}x},\tag{110}$$

where $\mathcal{E}_n$ is the $n$-th eigenvalue. For general $x$, this is not a genuine eigenvalue equation because the dilatation generator in the RG Hamiltonian preserves only one point in space which is chosen to be $x = 0$ here. A local operator inserted at $x$ is transported to $e^{-z}x$ under the RG flow. Only the states that support local excitations at $x = 0$ can remain invariant under the RG evolution. $\{\psi_{n,0}\}$ forms the complete basis of states that support local excitations at $x = 0$. Generic excited states can be obtained by applying local operators, denoted as $\hat{\mathcal{A}}_{n,x,t}$, to the ground state as

$$\psi_{n,x}[t] = (\hat{\mathcal{A}}_{n,x,t}\psi_0)[t].\tag{111}$$

$\hat{\mathcal{A}}_{n,x,t}$, which consists of $t(x)$ and $\frac{\delta}{\delta t(x)}$, creates local excitations at $x$, where $\hat{\mathcal{A}}_{0,x,t} = \mathbb{1}$. From the correspondence in Eq. (96), $\hat{\mathcal{A}}_{n,x,t}$ is dual to an operator that acts on the Boltzmann weight,

$$\int_I \mathcal{D}t\,(\hat{\mathcal{A}}_{n,x,t}\psi_0)[t]\int \mathcal{D}\phi\,e^{-S_0-\int d^D x\,t(x)\mathcal{O}(x)} = \int \mathcal{D}\phi\,(\hat{\mathcal{A}}_{n,x,\phi}\mathcal{W})_\phi\,[J_1^*,J_2^*,\dots],\tag{112}$$

where $\hat{\mathcal{A}}_{n,x,\phi}$ consists of $\mathcal{O}(x)$ and $\frac{\delta}{\delta\mathcal{O}(x)}$. $\hat{\mathcal{A}}_{n,x,t}\,(\hat{\mathcal{A}}_{n,x,\phi})$ is the representation of the operator in space $V\,(W)$. Henceforth, we use $\hat{\mathcal{A}}_{n,x}$ to denote the operator itself when there is no need to specify its representation.

We now verify that excited states indeed correspond to the fixed point perturbed with local operators that have definite scaling dimensions. For this, we consider the IR fixed point theory with a small perturbation added at the origin,

$$e^{\epsilon_{n,0}\hat{\mathcal{A}}_{n,0}}|S_{J_1^*,J_2^*,\dots}\rangle = \int \mathcal{D}\phi\,(e^{\epsilon_{n,0}\hat{\mathcal{A}}_{n,0,\phi}}\mathcal{W})_\phi\,[J_1^*,J_2^*,\dots]\,|\phi\rangle,\tag{113}$$

where $\epsilon_{n,0}$ is an infinitesimal parameter. The RG Hamiltonian generates an evolution of this state as

$$e^{-H dz}\left[e^{\epsilon_{n,0}\hat{\mathcal{A}}_{n,0}}|S_{J_1^*,J_2^*,\dots}\rangle\right] = \int_I \mathcal{D}t\,e^{-\mathcal{H}^0 dz}\left(\psi_0[t]+\epsilon_{n,0}\psi_{n,0}[t]\right)\int \mathcal{D}\phi\,e^{-S_0-\int d^D x\,t(x)\mathcal{O}(x)}|\phi\rangle$$
$$= e^{-\mathcal{E}_0 dz}e^{\epsilon_{n,0}e^{-(\mathcal{E}_n-\mathcal{E}_0)dz}\hat{\mathcal{A}}_{n,0}}|S_{J_1^*,J_2^*,\dots}\rangle\tag{114}$$

to the linear order in $\epsilon_{n,0}$. This implies that the infinitesimal source evolves as

$$\frac{d\epsilon_{n,0}}{dz} = -(\mathcal{E}_n-\mathcal{E}_0)\epsilon_{n,0}\tag{115}$$

under the RG flow, and $\hat{\mathcal{A}}_{n,0,\phi}$ is a scaling operators with the scaling dimensions

$$\Delta_n = \mathcal{E}_n - \mathcal{E}_0. \qquad (116)$$

The spectrum of $\mathcal{H}^0$ encodes the information of all scaling operators and their scaling dimensions. $\hat{\mathcal{A}}_{n,x}$'s create local excitations with definite scaling dimensions, and are called scaling operators. On the other hand, $\mathbf{O}_M(x)$'s represent general multi-trace operators in terms of which the UV action is written, and are called UV operators. In general, the scaling operators can be written as linear superpositions of UV operators :

$$\hat{\mathcal{A}}_{n,x,\phi} = \sum_M g_{n,M} \mathbf{O}_M(x). \qquad (117)$$

The inverse of Eq. (117) can be written as $\mathbf{O}_M(x) = \sum_n (g^{-1})^{M,n} \hat{\mathcal{A}}_{n,x,\phi}$.

From the local scaling operators, one can generate translationally invariant eigenstates by turning on the sources uniformly in space. For example,

$$\psi_n [t] = \int d^D x \; \psi_{n,x} [t] \qquad (118)$$

is an eigenstate with energy $\Delta_n - D$. The shift in the energy by $-D$ is from the spatial dilatation included in $\mathcal{H}^0$,

$$e^{-\mathcal{H}^0 dz} \psi_n [t] = e^{-\Delta_n dz} \int d^D x \; \psi_{n,e^{-dz}x} [t] = e^{-(\Delta_n - D)dz} \int d^D \tilde{x} \; \psi_{n,\tilde{x}} [t] , \qquad (119)$$

where $x = e^{dz} \tilde{x}$ is used. If $\Delta_n > D$ for all $n$, the fixed point is stable as all deformations with spatially uniform sources are irrelevant. Local excitations with energy gap less than (equal to) $D$ correspond to relevant (marginal) operators. Throughout this paper, we assume that local excitations of the RG Hamiltonian have a non-zero gap ($\Delta_n > 0$).

### 3. Mixing matrix

According to Eq. (117), $g_{n,M}$ encodes the relation between scaling operators, $\hat{\mathcal{A}}_{n,x,\phi}$ and UV operators, $\mathbf{O}_M(x)$. By writing

$$\sum_{n,x} \epsilon_{n,x} \hat{\mathcal{A}}_{n,x,\phi} = \sum_{M,x} \epsilon_M^{UV}(x) \mathbf{O}_M(x) \qquad (120)$$

and using Eq. (117), we obtain the relation between the sources for the scaling operators and the UV operators,

$$\epsilon_M^{UV}(x) = \sum_n \epsilon_{n,x} g_{n,M}, \quad \epsilon_{n,x} = \sum_M \epsilon_M^{UV}(x)(g^{-1})^{M,n}. \qquad (121)$$

To the linear order in $\epsilon$, the $\beta$-functions of these UV couplings are

$$
\begin{aligned}
\frac{d\epsilon_M^{UV}(x)}{dz} &= -\sum_n (\mathcal{E}_n - \mathcal{E}_0)\epsilon_{n,x}g_{n,M} \\
&= -\sum_{n,M'} \epsilon_{M'}^{UV}(x)(g^{-1})^{M'n}g_{n,M}(\mathcal{E}_n - \mathcal{E}_0).
\end{aligned}
\tag{122}
$$

This gives the mixing matrix

$$
\mathcal{M}^{M'}{}_M = -\sum_n (g^{-1})^{M'n}g_{n,M}(\mathcal{E}_n - \mathcal{E}_0).
\tag{123}
$$

It is noted that what appears on the right hand side of Eq. (123) is determined from the spectrum of $\mathcal{H}^0$ which is fixed by the beta functions defined in the subspace of the single-trace couplings. Eq. (123) shows that this small set of data completely fixes the full mixing matrix that involves all multi-trace operators. This shows that the $\beta$-functions for general multi-trace operators are fixed by those for the single-trace operator to the linear order in the deformation. In the following two subsections, we show that this holds beyond the linear order.

### 4. Operator product expansion

The operator product expansion (OPE) between general multi-trace operators is also fully encoded in the spectrum of $\mathcal{H}^0$. Suppose we insert two local scaling operators $\hat{\mathcal{A}}_{n,x/2}$ and $\hat{\mathcal{A}}_{n',-x/2}$ at $x/2$ and $-x/2$, respectively. The wavefunction for the resulting theory is $\psi_{n,n',(x/2,-x/2)}[t] = (\hat{\mathcal{A}}_{n,x/2,t}\hat{\mathcal{A}}_{n',-x/2,t}\psi_0)[t]$. Under the evolution generated by $\mathcal{H}^0$, the separation between the two operators decreases exponentially in $z$. For $z \ll \ln\frac{|x|}{a}$, where $a = \Lambda^{-1}$ is the short distance cutoff length scale, the two local excitations remain well-separated in space, and evolve independently,

$$
e^{-\mathcal{H}^0 z}(\hat{\mathcal{A}}_{n,x/2,t}\hat{\mathcal{A}}_{n',-x/2,t}\psi_0)[t] \approx e^{-(\Delta_n+\Delta_{n'}+\mathcal{E}_0)z}(\hat{\mathcal{A}}_{n,xe^{-z}/2,t}\hat{\mathcal{A}}_{n',-xe^{-z}/2,t}\psi_0)[t].
\tag{124}
$$

This follows from the facts that 1) $\hat{\mathcal{A}}_{n',-x/2}$ and $\hat{\mathcal{A}}_{n,x/2}$ create local excitations with energies $\Delta_n$ and $\Delta_{n'}$ above the ground state; 2) $\mathcal{H}^0$ is local at length scales larger than $a$, and two operators evolve independently with the total energy given by $\mathcal{E}_{n,n'} = \mathcal{E}_0 + (\Delta_n + \Delta_{n'})$ in the limit that their separation is large. As two excitations approach, they interact, and the state evolves into a more complicated state. Nonetheless, the state obtained after evolving the initial state for $z_x = \ln\frac{|x|}{a}$ can be written as a linear superposition of eigenstates of the RG Hamiltonian with local excitations located at the origin,

$$
e^{-\mathcal{H}^0 z_x}(\hat{\mathcal{A}}_{n,x/2,t}\hat{\mathcal{A}}_{n',-x/2,t}\psi_0)[t] = e^{-(\Delta_n+\Delta_{n'}+\mathcal{E}_0)z_x}\sum_l F_{n,n',l}(x)(\hat{\mathcal{A}}_{l,0,t}\psi_0)[t],
\tag{125}
$$

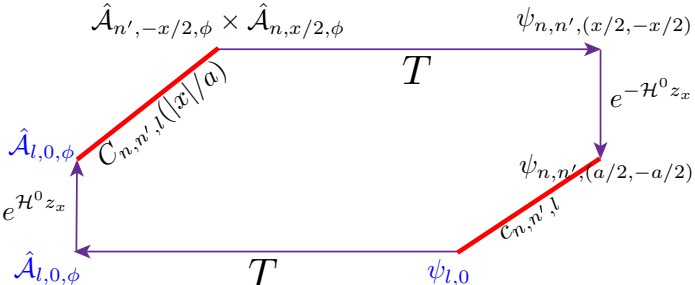

FIG. 3. The procedure of extracting OPE between two local operators. Two local operators $\hat{\mathcal{A}}_{n',-x/2,\phi}$ and $\hat{\mathcal{A}}_{n',x/2,\phi}$ shown in the upper left corner undergo a series of transformations following the arrows in the clockwise direction : 1) the operators inserted to the IR fixed point corresponds to the ground state with two local excitations through the action-state correspondence (right arrow $T$); 2) the state is evolved with the RG Hamiltonian for $z_x = \ln \frac{|x|}{a}$ (down arrow $e^{-\mathcal{H}^0 z}$ ); 3) the resulting state supports local excitation near the origin which is then expressed as a linear superposition of the eigenstates $\psi_{l,0}$ with local excitations at the origin weighted with $c_{n,n',l}$; 4) the resulting state corresponds to the fixed point with local operator insertions at the origin (left arrow $T$); 5) the theory is evolved backward in RG time by $-z_x$ (up arrow $e^{\mathcal{H}^0 z}$ ). The identification of the final operator with the initial product of two operators gives the desired operator product expansion.

where $F_{n,n',l}(x)$ is a function that captures the effect of interaction. It is a regularization dependent function which can be computed from the RG Hamiltonian. There is no interaction between two operators when the separation is much larger than $a$. This follows from the fact that one only integrates out modes whose wavelengths are order of $a$ in each coarse graining step. As a result, $F_{n,n',l}(x)$ exponentially approaches a constant in the large $|x|$ limit, $\lim_{|x|\to\infty} |F_{n,n',l}(x) - c_{n,n',l}| \sim e^{-|x|/a}$, where $c_{n,n',l} \equiv F_{n,n',l}(\infty)$. Now the state in Eq. (125) is evolved backward in RG time $z_x$, which results in

$$(\hat{\mathcal{A}}_{n,x/2,t}\hat{\mathcal{A}}_{n',-x/2,t}\psi_0)\,[t] = \sum_l F_{n,n',l}(x)\ e^{(\Delta_l - \Delta_n - \Delta_{n'})z_x}(\hat{\mathcal{A}}_{l,0,t}\psi_0)\,[t]\,. \qquad (126)$$

From this, we obtain the OPE of scaling operators :

$$\hat{\mathcal{A}}_{n,x/2,\phi} \times \hat{\mathcal{A}}_{n',-x/2,\phi} = \sum_l C_{n,n',l}\,(x)\,\hat{\mathcal{A}}_{l,0,\phi}, \qquad (127)$$

where

$$C_{n,n',l}(x) = \frac{F_{n,n',l}(x)}{|x/a|^{\Delta_n+\Delta_{n'}-\Delta_l}}\,. \qquad (128)$$

In the large $|x|$ limit, this reduces to the standard form, $C_{n,n',l}(x) \sim \frac{c_{n,n',l}}{|x|^{\Delta_n+\Delta_{n'}-\Delta_l}}$. The procedure used to extract the OPE is summarized in the Fig. 3.

### 5. β-functionals

Now we are ready to derive the full beta functions. For generality, we consider the case in which the sources are position dependent, and derive the beta functionals of the Wilsonian RG from the quantum RG. We consider a UV theory with general deformations added to the fixed point theory $S^*$ parametrized by $J^*_{n,x}$,

$$S = S^* + \int d^D x \; \epsilon_{n,x} \hat{\mathcal{A}}_{n,x,\phi}, \tag{129}$$

where $\epsilon_{n,x} = J_{n,x} - J^*_{n,x}$, and repeated indices are summed over.

The state that corresponds to this deformed theory is given by

$$|\Psi\rangle = |\psi_0\rangle - \int d^D x \left( \epsilon_{n,x} - \frac{1}{2} \int d^D x' \epsilon_{l,x+\frac{x'}{2}} \epsilon_{m,x-\frac{x'}{2}} C_{lmn}(x') \right) |\psi_{n,x}\rangle + O(\epsilon^3) \tag{130}$$

to the second order in $\epsilon_n$, where we use $\hat{\mathcal{A}}_{l,x+x'/2,\phi} \times \hat{\mathcal{A}}_{m,x-x'/2,\phi} = \sum_n C_{lmn}(x') \hat{\mathcal{A}}_{n,x,\phi}$ obtained in Eq. (127). Under the evolution generated by the RG Hamiltonian, the state evolves to

$$|\Psi(z)\rangle = e^{-\mathcal{E}_0 dz} \left[ |\psi_0\rangle - e^{-\Delta_n dz} \int d^D x \left( \epsilon_{n,x} - \frac{1}{2} \int d^D x' \epsilon_{l,x+\frac{x'}{2}} \epsilon_{m,x-\frac{x'}{2}} C_{lmn}(x') \right) |\psi_{n,xe^{-dz}}\rangle \right] + O(\epsilon^3)$$

$$= e^{-\mathcal{E}_0 dz} \left[ |\psi_0\rangle - e^{-\Delta_n dz} \int d^D \tilde{x} e^{Ddz} \left( \epsilon_{n,e^{dz}\tilde{x}} - \frac{1}{2} \int d^D \tilde{x}' e^{Ddz} \epsilon_{l,e^{dz}(\tilde{x}+\frac{\tilde{x}'}{2})} \epsilon_{m,e^{dz}(\tilde{x}-\frac{\tilde{x}'}{2})} C_{lmn}(e^{dz}\tilde{x}') \right) |\psi_{n,\tilde{x}}\rangle \right]$$

$$+ O(\epsilon^3), \tag{131}$$

where the renormalized spatial coordinate is defined as $x = \tilde{x} e^{dz}$ in the second line. The final state can be written in the form of Eq. (130) provided $\epsilon_{n,x}$ is replaced with renormalized source,

$$\epsilon'_{n,\tilde{x}} = e^{-\tilde{\Delta}_n dz} \epsilon_{n,e^{dz}\tilde{x}}$$

$$+ \frac{1}{2} \int d^D \tilde{x}' \; \epsilon_{l,e^{dz}(\tilde{x}+\frac{\tilde{x}'}{2})} \epsilon_{m,e^{dz}(\tilde{x}-\frac{\tilde{x}'}{2})} \left[ e^{-(\tilde{\Delta}_l+\tilde{\Delta}_m)dz} C_{lmn}(\tilde{x}') - e^{(D-\tilde{\Delta}_n)dz} C_{lmn}(e^{dz}\tilde{x}') \right] + O(\epsilon^3), \tag{132}$$

where $\tilde{\Delta}_n = \Delta_n - D$. $|\Psi(z)\rangle$ corresponds to the action,

$$S(z) = S^* + \int d^D x \; \epsilon'_{n,x} \hat{\mathcal{A}}_{n,x,\phi}, \tag{133}$$

where the renormalized source is given by

$$\epsilon'_{n,x} = \epsilon_{n,x} + dz \left\{ -\tilde{\Delta}_n \epsilon_{n,x} + x \frac{\partial}{\partial x} \epsilon_{n,x} + \frac{1}{2} \int d^D x' \; \epsilon_{l,x+\frac{x'}{2}} \epsilon_{m,x-\frac{x'}{2}} G_{lmn}(x') \right\} + O(\epsilon^3) \tag{134}$$

to the linear order in $dz$ with

$$G_{lmn}(y) = -C_{lmn}(y)y\frac{\partial}{\partial y}\ln F_{lmn}(y). \tag{135}$$

The term quadratic in $\epsilon$ in Eq. (134) describes the fusion of two operators into one. Since $G_{lmn}(y)$ decays exponentially in the large $y$ limit, operators whose separation is smaller than $a$ mainly contribute to the fusion process. This shows that the $\beta$-functionals of $J_{n,x}$ are local,

$$\frac{d}{dz}J_{n,x} = -(\Delta_n - D)(J_{n,x} - J_{n,x}^*) + x\frac{\partial}{\partial x}(J_{n,x} - J_{n,x}^*)$$
$$+ \frac{1}{2}\int d^D x' \,(J_{l,x+\frac{x'}{2}} - J_{l,x+\frac{x'}{2}}^*)(J_{m,x-\frac{x'}{2}} - J_{m,x+\frac{x'}{2}}^*)G_{lmn}(x') + O\Big((J - J^*)^3\Big). \tag{136}$$

This also confirms that couplings are irrelevant (relevant) if $\Delta_n > D$ ($\Delta_n < D$). It is noted that these are $\beta$ functions for sources of the scaling operators. The $\beta$ functions for the sources of the UV operators are obtained from Eq. (136) through a linear transformation in Eq. (121).

## V. TOY MODELS

In the previous two sections, it is shown that there exist general constraints that $\beta$-functions satisfy due to the fact that the full Wilsonian RG can be replaced with a quantum RG defined in the subspace of single-trace couplings. While the quantum theory is well-defined, it is in general no easier than the original problem. Only in the large $N$ limit, the quantum problem can be solved with a semi-classical approximation. See Refs. [34] for recent development. In this section, we consider toy models in which the bulk theory is non-interacting and can be solved explicitly.

### A. 0-dimensional solvable Example

In this section, we consider a toy model of a 0-dimensional theory. For simplicity, we assume that there is only one single-trace operator, and that only single-trace and double-trace operators are generated under the coarse graining when the reference theory $S_0$ is deformed by the single-trace operator. We assume that the reference theory is invariant under a $\mathbb{Z}_2$ symmetry, and the single-trace operator is odd under the symmetry. The symmetry constrains the form of the $\beta$-functions. We assume that the $\beta$-functions in the

subspace of the single-trace deformation take the following form

$$\beta_0(t, 0, ..) = f - wt^2,$$
$$\beta_1(t, 0, ..) = at,$$
$$\beta_2(t, 0, ..) = -b,$$
$$\beta_{n>2}(t, 0, ..) = 0. \tag{137}$$

Here $t$ stands for the source for the single-trace operator. $\beta_0$ describes the flow of the identity operator. $\beta_1$ ($\beta_2$) is the beta function for the single (double)-trace coupling. $a$, $b$, $w$ and $f$ are non-zero real parameters. Under the $\mathbb{Z}_2$ transformation, the single-trace coupling transforms as $t \to -t$. This guarantees that $\beta_0$ and $\beta_2$ are even in $t$, and $\beta_1$ is odd in $t$. Eq. (137) describes how couplings are renormalized when $S_0$ is deformed by the single-trace operator. In particular, $b \neq 0$ implies that the double-trace operator is generated and the RG flow leaves the subspace of single-trace deformation even if the UV theory has only single-trace deformation. From Eq. (137), it is unclear where the double-trace coupling $t_2$ eventually flows under the RG flow. Depending on the sign of $\beta_2(t, t_2)$ at large $t_2$, the theory may or may not flow to a scale invariant fixed point in the IR. Remarkably, $\beta_2(t, t_2)$ at general values of $t_2$ is already encoded in Eq. (137), which determines the fate of the RG flow in the space of all couplings.

Following the formalism in Sec. IV B, we obtain the bulk RG Hamiltonian in Eq. (104). It is convenient to shift the RG Hamiltonian to remove a constant piece,

$$\tilde{\mathcal{H}} = \mathcal{H} + \left(f - \frac{a}{2}\right) = b\left[\tilde{p} + \frac{ia}{2b}\tilde{t}\right]^2 + \left[\frac{a^2 - 4bw}{4b}\right]\tilde{t}^2, \tag{138}$$

where $\tilde{t} = it$ and $\tilde{p} = -ip$ fluctuate along the real axis [13]. They satisfy commutation relation $\left[\tilde{p}, \tilde{t}\right] = i$. This is a TP-symmetric non-Hermitian quadratic Hamiltonian[41]. As is shown in Appendix B, the spectrum of this RG Hamiltonian is given by

$$E_n = (n + \frac{1}{2})\sqrt{\eta}, \tag{139}$$

where $\eta = a^2 - 4bw$. Unlike the Hermitian cases, the left and right eigenstates take different forms,

$$\psi_n^R(\tilde{t}) = \frac{1}{\sqrt{2^n n!}}\left(\frac{\varepsilon}{2\pi}\right)^{1/4} e^{-\xi_R \tilde{t}^2} H_n\left[\sqrt{\frac{\varepsilon}{2}}\tilde{t}\right],$$
$$\psi_n^L(\tilde{t}) = \frac{1}{\sqrt{2^n n!}}\left(\frac{\varepsilon}{2\pi}\right)^{1/4} e^{-\xi_L \tilde{t}^2} H_n\left[\sqrt{\frac{\varepsilon}{2}}\tilde{t}\right], \tag{140}$$

---

[13] The additional shift $a/2$ is generated from the normal ordering.

where $\xi_{R,L} = \frac{1}{4b}(\sqrt{\eta} \pm a)$, $\varepsilon = \frac{1}{b}\sqrt{\eta}$ and $H_n(x)$ is the Hermite polynomial : $H_0(x) = 1$, $H_1(x) = 2x$, $H_2(x) = 4x^2 - 2$, ....

The spectrum of the Hamiltonian is determined by the parameters in the $\beta$-functions, $a$, $b$ and $w$. First, the eigenvalues of the Hamiltonian are real for $\eta > 0$. Second, the eigenstates are square-integrable for $\xi_{R,L} > 0$. These conditions are satisfied for $b > 0$ and $w < 0$ for any real $a$. In the following, we first focus on this parameter region that supports a real spectrum with normalizable eigenstates. At the end of this section, we will see that violation of these conditions is associated with a loss of stable fixed point in the IR.

### 1. Fixed point

A generic initial state evolves to the right ground state of $\tilde{\mathcal{H}}$ in the large $z$ limit,

$$\psi_0^R(\tilde{t}) = (\frac{\varepsilon}{2\pi})^{1/4} e^{-\xi_R \tilde{t}^2}. \tag{141}$$

As discussed in Sec. IV C, the right ground state corresponds to the stable fixed point of the theory. To extract the fixed point action, we write the right ground state as

$$\begin{aligned}|0\rangle &= \int \mathcal{D}\phi \int_R \mathcal{D}\tilde{t} \ \psi_0^R(\tilde{t}) e^{-S_0 + i\tilde{t}\mathcal{O}} |\phi\rangle \\ &= (\frac{\varepsilon}{2\pi})^{1/4} \sqrt{\frac{\pi}{\xi_R}} \int \mathcal{D}\phi \ e^{-S_0 - \frac{1}{4\xi_R}\mathcal{O}^2} |\phi\rangle.\end{aligned} \tag{142}$$

The logarithm of its wavefunction in the $\phi$ basis gives the fixed point action,

$$S^* = S_0 + J_1^* \mathcal{O} + J_2^* \mathcal{O}^2 \tag{143}$$

with $J_1^* = 0$ and $J_2^* = \frac{1}{4\xi_R}$. We emphasize that the stable fixed point exists away from the subspace of the single-trace coupling, yet the position of the fixed point is fully determined from the beta functions defined in the subspace.

### 2. Scaling operators and their OPEs

Now we turn our attention to excited states of $\tilde{\mathcal{H}}$. Each right eigenstate corresponds to the fixed point theory with an operator insertion. The $n$-th excited state is

$$\begin{aligned}|n\rangle &= \int \mathcal{D}\phi \int_R \mathcal{D}\tilde{t} \ \psi_n^R(\tilde{t}) e^{-S_0 + i\tilde{t}\mathcal{O}} |\phi\rangle \\ &= \int \mathcal{D}\phi \frac{1}{\sqrt{2^n n!}} (\frac{\varepsilon}{2\pi})^{1/4} \sqrt{\frac{\pi}{\xi_R}} e^{-S_0} H_n\left[-i\sqrt{\frac{\varepsilon}{2}} \frac{\partial}{\partial \mathcal{O}}\right] e^{-\frac{1}{4\xi_R}\mathcal{O}^2} |\phi\rangle.\end{aligned} \tag{144}$$

The excited states can be reached by applying 'raising' operators to the ground state,

$$|n\rangle = \hat{\mathcal{A}}_n|0\rangle, \tag{145}$$

where $\hat{\mathcal{A}}_n$ is the operator that maps the ground state to the $n$-th excited states that has scaling dimension $\sqrt{\bar{\eta}}n$. In general, the $n$-th scaling operator is given by a linear superposition of all $k$-trace operators with $k = n, n-2, n-4, ...$ For $n = 0, 1, 2, 3, 4$, we obtain

$$
\begin{aligned}
\hat{\mathcal{A}}_0 &= \mathbb{1}, \\
\hat{\mathcal{A}}_1 &= i\frac{\sqrt{\varepsilon}}{2\xi_R}\mathcal{O}, \\
\hat{\mathcal{A}}_2 &= \frac{1}{\sqrt{2}}(\frac{\varepsilon}{2\xi_R} - 1)\mathbb{1} - \frac{1}{\sqrt{2}}\frac{\varepsilon}{4\xi_R^2}\mathcal{O}^2, \\
\hat{\mathcal{A}}_3 &= -\frac{i\sqrt{\varepsilon^3}}{8\sqrt{6}\xi_R^3}\mathcal{O}^3 + \frac{i\sqrt{3\varepsilon}}{2\sqrt{2}\xi_R}\Big(\frac{\varepsilon}{2\xi_R} - 1\Big)\mathcal{O}, \\
\hat{\mathcal{A}}_4 &= \frac{\varepsilon^2}{32\sqrt{6}\xi_R^4}\mathcal{O}^4 + \frac{\sqrt{3}\varepsilon}{4\sqrt{2}\xi_R^2}\Big(1 - \frac{\varepsilon}{2\xi_R}\Big)\mathcal{O}^2 + \frac{\sqrt{3}}{2\sqrt{2}}(1 - \frac{\varepsilon}{2\xi_R})^2.
\end{aligned} \tag{146}
$$

It is straightforward to identify all scaling operators in this way.

The OPE coefficient can be computed accordingly. For instance, two $\hat{\mathcal{A}}_1$ fuse to

$$\hat{\mathcal{A}}_1 \times \hat{\mathcal{A}}_1 = (1 - \frac{\varepsilon}{2\xi_R})\hat{\mathcal{A}}_0 + \sqrt{2}\hat{\mathcal{A}}_2, \tag{147}$$

and the associated OPE coefficients are given by $C_{110} = (1 - \frac{\varepsilon}{2\xi_R})$, $C_{112} = \sqrt{2}$. Similarly, all OPE coefficients can be extracted from the eigenstates of the RG Hamiltonian. In Tab. II, we list all OPE for $\hat{\mathcal{A}}_n \times \hat{\mathcal{A}}_m$ up to $n, m = 2$.

TABLE II. Operator product expansion of $\hat{\mathcal{A}}_n \times \hat{\mathcal{A}}_m$ for $n, m = 0, 1, 2$.

| $m$ \ $n$ | 0 | 1 | 2 |
|---|---|---|---|
| 0 | $\hat{\mathcal{A}}_0$ | $\hat{\mathcal{A}}_1$ | $\hat{\mathcal{A}}_2$ |
| 1 | $\hat{\mathcal{A}}_1$ | $(1 - \frac{\varepsilon}{2\xi_R})\hat{\mathcal{A}}_0 + \sqrt{2}\hat{\mathcal{A}}_2$ | $\sqrt{2}(1 - \frac{\varepsilon}{2\xi_R})\hat{\mathcal{A}}_1 + \sqrt{3}\hat{\mathcal{A}}_3$ |
| 2 | $\hat{\mathcal{A}}_2$ | $\sqrt{2}(1 - \frac{\varepsilon}{2\xi_R})\hat{\mathcal{A}}_1 + \sqrt{3}\hat{\mathcal{A}}_3$ | $(1 - \frac{\varepsilon}{2\xi_R})^2\hat{\mathcal{A}}_0 + 2\sqrt{2}(1 - \frac{\varepsilon}{2\xi_R})\hat{\mathcal{A}}_2 + \sqrt{6}\hat{\mathcal{A}}_4$ |

### 3. Full $\beta$-functions

Based on Sec. IV C 5, we can immediately write down the full $\beta$-functions of the theory. In 0 dimension, there is no $x$ dependence of the couplings and OPE coefficients. By setting

$D = 0$ and $C_{lmn}(x) = c_{lmn}$ in Eq. (132), we readily obtain the $\beta$-function,

$$\frac{d}{dz} J_n = -\Delta_n(J_n - J_n^*)$$
$$+ \frac{1}{2}(\Delta_n - \Delta_l - \Delta_m)c_{lmn}(J_l - J_l^*)(J_m - J_m^*) + O\Big((J - J^*)^3\Big), \quad (148)$$

From Eq. (139) that implies $\Delta_n = \sqrt{\eta}\, n$ and Table II, we obtain the beta functions for the couplings of $\hat{\mathcal{A}}_1$ and $\hat{\mathcal{A}}_2$,

$$\frac{d}{dz} J_1 = -\sqrt{\eta}\left[1 + \frac{\varepsilon}{2\sqrt{2}\xi_R^2} - \frac{1}{\sqrt{2}\xi_R}\right] J_1 - 2\sqrt{2\eta}(1 - \frac{\varepsilon}{2\xi_R})J_1 J_2 + O\Big((J - J^*)^3\Big), \quad (149)$$

$$\frac{d}{dz} J_2 = \frac{\sqrt{\eta}}{2\xi_R} + \frac{\varepsilon\sqrt{\eta}}{8\sqrt{2}\xi_R^3} - \frac{\sqrt{\eta}}{4\sqrt{2}\xi_R^2}$$
$$- 2\sqrt{\eta}\left[1 + \frac{\varepsilon}{2\sqrt{2}\xi_R^2} - \frac{1}{\sqrt{2}\xi_R}\right] J_2 - 2\sqrt{2\eta}(1 - \frac{\varepsilon}{2\xi_R})J_2^2 + O\Big((J - J^*)^3\Big), \quad (150)$$

where $\frac{d}{dz} J_n = -\beta_n$. Although $\beta_{n>2} = 0$ in the subspace of single-trace coupling, they are in general non-zero away from the subspace. It is straightforward to compute $\beta_n$ for any $n$ order by order in $(J - J^*)$. This shows that the full beta functions are indeed encoded in the RG Hamiltonian that is fixed by the beta functions defined in the space of single-trace couplings.

The beta functions for multi-trace operators allow us to explore the RG flow away from the subspace of the single-trace coupling. Eqs. (149) and (150) computed to the quadratic order in $\delta J = (J - J^*)$ can be trusted near $J = J^*$. To describe the RG flow far away from the stable fixed point, one needs to take into account terms that are higher order in $\delta J$ and higher trace couplings. Here we focus on the flow in the space of $J_1$ and $J_2$ near $J = J^*$. To the quadratic order in $\delta J$, $J_n$ with $n > 2$ are not generated, and we can trust Eqs. (149) and (150) near $J = J^*$. The RG flow in the space of $J_1$ and $J_2$ is shown in Fig. 1 for $a = -0.1$, $b = 0.5$ and $w = -0.5$. We find two fixed points at

$$(J_1^*, J_2^*)_- = \left(0, \frac{1}{4\xi_R}\right), \quad (151)$$

$$(J_1^*, J_2^*)_+ = \left(0, \frac{1}{4\xi_R} + \frac{\sqrt{2}\xi_R}{(\varepsilon - 2\xi_R)}\right), \quad (152)$$

where $0 < (J_2^*)_- < (J_2^*)_+$ because $\xi_R > 0$ and $\varepsilon > 2\xi_R$. In either the small $\xi_R$ or large $\varepsilon$ limit, the second fixed point is close to the stable fixed point, and the terms that are higher order in $\delta J$ in Eqs. (149) and (150) are negligible near these two fixed points. The first fixed point in Eq. (151) is the stable fixed point identified from the ground state of the RG Hamiltonian in Eq. (143). Both $\delta J_1$ and $\delta J_2$ are irrelevant whose scaling dimensions

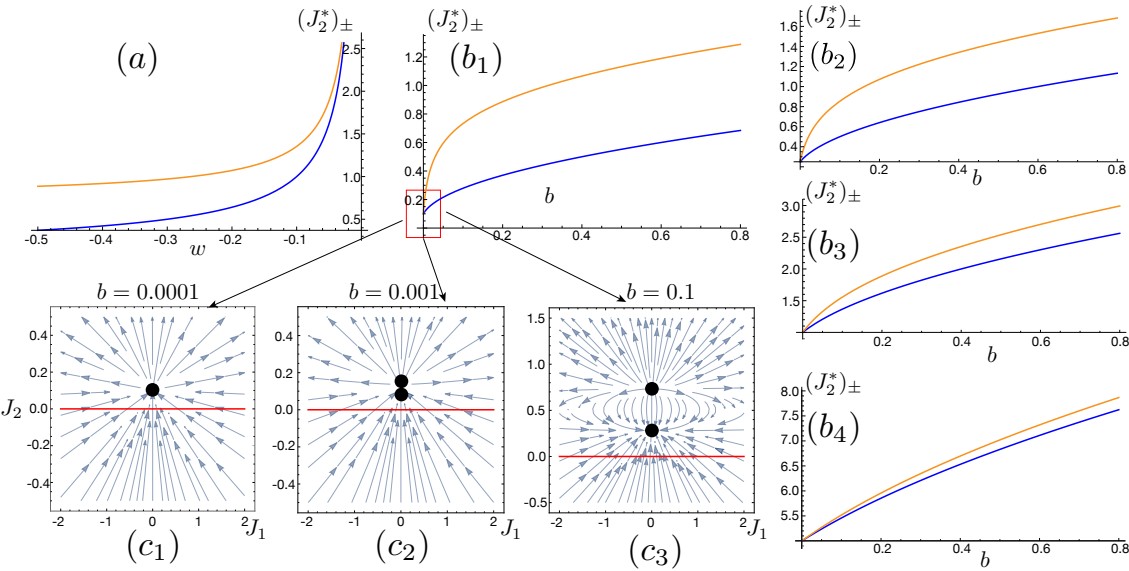

FIG. 4. The double-trace coupling $J_2$ at the stable (blue line) and unstable (orange line) fixed point as a function of $w$ at $b = 0.2$ (a); as a function of $b$ at $w = -0.5$ ($b_1$), $w = -0.2$ ($b_2$), $w = -0.05$ ($b_3$), $w = -0.01$ ($b_4$). Three RG flow diagrams at $w = -0.5$ and $b = 0.0001$ ($c_1$), $b = 0.001$ ($c_2$), $b = 0.1$ ($c_3$), $b = 0.5$ (Fig. 1) are also presented. $a = -0.1$ is used for all plots.

are $-\sqrt{\eta}$ and $-2\sqrt{\eta}$, respectively. The second fixed point in Eq. (152) is an unstable fixed point. Both $\delta J_1$ and $\delta J_2$ are relevant with scaling dimensions, $\sqrt{\eta}$ and $2\sqrt{\eta}$, respectively. In Fig. 4(a) and Fig. 4($b_{1\sim4}$), we plot the value of $J_2$ at the two fixed points as $b$ and $w$ are varied, respectively. For $b > 0$ and $w < 0$, the spectrum of the RG Hamiltonian is real and the eigenstates are normalizable. In this case, $\xi_R$ and $\varepsilon - 2\xi_R$ are both positive and finite such that the two fixed points remain separated, as shown in Fig. 4($c_{2\sim3}$). The RG flow changes qualitatively if $b$ or $w$ approaches 0. In Appendix C, we examine the RG flow in the $w \to 0^-$ and $b \to 0^+$ limits in more details.

## B. $D$-dimensional solvable example

In this section, we extend the discussion in the previous section to a $D$-dimensional field theory. For simplicity, we continue to assume that there is only one single-trace operator, and that multi-trace operators higher than double-trace operator are not generated when the reference action is deformed only by the single-trace operator. We also assume that the reference theory is invariant under the spatial translation, the rotation and the inversion symmetry, and has an internal $\mathbb{Z}_2$ symmetry under which the single-trace operator is odd. The symmetry largely fixes the form of the $\beta$-functionals in the subspace of the single-

trace couplings order by order in the coupling. To be concrete, we consider the following $\beta$-functionals in the subspace of single-trace couplings,

$$\beta_0\left[t, 0, ..; x\right] = f - g\left[\partial_x t(x)\right]^2 - wt^2(x),$$
$$\beta_1\left[t, 0, ..; x\right] = at(x) - x\partial_x t(x),$$
$$\beta_2\left[t, 0, ..; x\right] = -b,$$
$$\beta_{k \geq 3}\left[t, 0, ..; x\right] = 0. \tag{153}$$

Here $\beta_k[t^{(1)}, t^{(2)}, t^{(3)}, ..; x]$ represents the $\beta$-functional for the $k$-trace operator at $\left(t^{(1)}, t^{(2)}, t^{(3)}, ..\right)$, where $t^{(m)}$ is the $m$-trace coupling. $(\partial_x t)^2 \equiv \sum_{\mu=1}^{D} (\partial_\mu t)^2$, and $x\partial_x t \equiv \sum_{\mu=1}^{D} x^\mu \partial_\mu t$. $f, g, w, a, b$ are constants that represent the contributions to the beta functions generated from integrating out short distance modes and rescaling the fundamental fields at every RG step. The last term in $\beta_1$ dilates the space because the coordinate in the $(l+1)$-th RG step is related to that in the previous step through $x^{(l+1)} = x^{(l)}e^{-dz}$. The rescaling makes sure that the UV cutoff remains invariant under the RG flow, and the same coarse graining can be applied at all steps. On the other hand, the rescaling of space reduces the size of the system in real space by $e^{-dz}$ at every RG step.

Eq. (153) fixes the bulk theory in Eq. (103), which in turn determines the fate of the field theory in the low-energy limit. The wavefunction that fully determines the renormalized action at scale $z$ is given by the path integration of the single-trace source and its conjugate variable,

$$\Psi[t, z] = \int_I \mathcal{D}t(x, z')\mathcal{D}p(x, z') \ \Psi_{J_1, J_2, ...}[t(0)]e^{-\int_0^z dz' L[t, p, z']}\Big|_{t(z)=t}. \tag{154}$$

While the bulk Lagrangian is quadratic in the present case, it depends on $x$ explicitly because of the dilatation term in $\beta_1$ of Eq. (153). This gives rise to a mixing between different Fourier modes in the momentum space[14]. The mixing makes it hard to compute the path integration directly. To bypass this problem, we follow the three steps described below.

1. We introduce new variables in the bulk,

$$t(x, z) = -i\tilde{t}(xe^z, z)e^{\frac{D}{2}z}, \quad p(x, z) = i\tilde{p}(xe^z, z)e^{\frac{D}{2}z}. \tag{155}$$

Besides the rescaling of spatial coordinate that undoes the dilatation, the fields are also multiplied with a factor $e^{\frac{D}{2}z}$ to compensate the $z$-dependent volume of the space. $\pm i$ is multiplied so that $\tilde{t}$ and $\tilde{p}$ fluctuate along the real axis. In the new variables, the dilatation effect disappears and Fourier modes with different momenta do not mix as will be shown later.

---

[14] The mixing arises because the momentum in the $(l+1)$-th RG step is related to the momentum in the previous step as $k^{(l+1)} = k^{(l)}e^{dz}$.

2. The path integration in Eq. (154) is performed in $\tilde{t}$ and $\tilde{p}$. This is done in the Hamiltonian picture.

3. The scale transformation is reinstated by expressing the $z$-dependent state in terms of

$$t'(x,z) = \tilde{t}(xe^z, z)e^{\frac{D}{2}z}, \quad p'(x,z) = \tilde{p}(xe^z, z)e^{\frac{D}{2}z}. \tag{156}$$

In the following sections, we implement these steps to identify the IR fixed point and the spectrum of scaling operators at the fixed point.

### 1. The RG Hamiltonian

In terms of the variables introduced in Eq. (155), the bulk Lagrangian is written as

$$L[t,p,z] = \int^{1/\lambda} d^D X \left( i\tilde{p}\partial_z\tilde{t} + i\frac{D}{2}\tilde{p}\tilde{t} - \sum_{n\geq 0} \tilde{\beta}_n[\tilde{t}; X]\tilde{p}^n e^{\frac{(n-2)D}{2}z} \right), \tag{157}$$

where

$$\tilde{\beta}_1[\tilde{t}; X] = \beta_1[t; x] - ie^{\frac{D}{2}z}X\partial_X\tilde{t},$$
$$\tilde{\beta}_n[\tilde{t}; X] = \beta_n[t; x] \quad \text{for } n \neq 1. \tag{158}$$

with $X = xe^z$. $\tilde{t}(X)$ and $\tilde{p}(X)$ obey the canonical commutation relation,

$$[\tilde{t}(X), \tilde{p}(X')] = -i\delta(X - X'). \tag{159}$$

The RG Hamiltonian density is given by

$$\mathcal{H}[\tilde{t}, \tilde{p}, z] = i\frac{D}{2}\tilde{p}\tilde{t} - \sum_{n\geq 0} \tilde{\beta}_n[\tilde{t}; X]\tilde{p}^n e^{\frac{(n-2)D}{2}z}. \tag{160}$$

By shifting the Hamiltonian by a constant, we write the Hamiltonian density as

$$\tilde{\mathcal{H}}[\tilde{t}, \tilde{p}, z] = \mathcal{H}[\tilde{t}, \tilde{p}, z] + e^{-Dz}f - \frac{1}{2}(a - \frac{D}{2})\delta(0) \tag{161}$$

$$= -ge^{2z}[\partial_X\tilde{t}(X)]^2 - w\tilde{t}^2(X) + i\frac{1}{2}(a + \frac{D}{2})[\tilde{t}(X)\tilde{p}(X) + \tilde{p}(X)\tilde{t}(X)] + b\tilde{p}^2(X).$$

As expected, the dilatation in Eq. (153) cancels with that in Eq. (158). Instead, the RG Hamiltonian acquires explicit $z$ dependence.

In the Fourier basis,

$$\tilde{t}(X) = \frac{1}{\sqrt{V}} \sum_K e^{iK \cdot X} \tilde{t}_K, \quad \tilde{p}(X) = \frac{1}{\sqrt{V}} \sum_K e^{iK \cdot X} \tilde{p}_K, \tag{162}$$

where $V = \lambda^{-D}$ is the volume of the system, the RG Hamiltonian can be written as

$$\tilde{\mathcal{H}}(z) = \sum_K \tilde{h}_K, \tag{163}$$

where

$$\tilde{h}_K = b\left\{ \left[\tilde{p}_K + i\zeta\tilde{t}_K\right]\left[\tilde{p}_{-K} + i\zeta\tilde{t}_{-K}\right] + \Omega_{K,z}^2 \tilde{t}_K \tilde{t}_{-K} \right\} \tag{164}$$

with

$$\Omega_{K,z}^2 = \sigma + \alpha e^{2z} K^2. \tag{165}$$

Here, $\zeta = \frac{1}{2b}(a + \frac{D}{2})$, $\sigma = \frac{1}{4b^2}(a + \frac{D}{2})^2 - \frac{w}{b}$ and $\alpha = -g/b$. Henceforth, we set $b = 1/2$, resulting in $\zeta = a + \frac{D}{2}$, $\sigma = \zeta^2 - 2w$ and $\alpha = -2g$. $\tilde{t}_K$ and $\tilde{p}_{-K}$ are canonical conjugate variables that satisfy $\left[\tilde{t}_K, \tilde{p}_{K'}\right] = -i\delta_{K,-K'}$. While $\tilde{t}_{K=0}$ and $\tilde{p}_{K=0}$ are real, $\tilde{t}_{K \neq 0}$ and $\tilde{p}_{K \neq 0}$ are complex with $\tilde{t}_K = \tilde{t}_{-K}^*$ and $\tilde{p}_K = \tilde{p}_{-K}^*$. The Hamiltonian can be decomposed into a sum of time-dependent harmonic oscillators,

$$\tilde{\mathcal{H}}(z) = \tilde{h}_0 + \sum_{K>0}' (\tilde{h}_{(R;K)} + \tilde{h}_{(I;K)}), \tag{166}$$

where $\sum_{K>0}'$ runs over the half of non-zero momenta with $K$ identified with $-K$, and

$$\tilde{h}_0 = b\left\{ \left[\tilde{p}_0 + i\zeta\tilde{t}_0\right]^2 + \Omega_{0,z}^2 \tilde{t}_0^2 \right\},$$
$$\tilde{h}_{(R;K>0)} = b\left\{ \left[\tilde{p}_{(R;K)} + i\zeta\tilde{t}_{(R;K)}\right]^2 + \Omega_{K,z}^2 (\tilde{t}_{(R;K)})^2 \right\}, \tag{167}$$
$$\tilde{h}_{(I;K>0)} = b\left\{ \left[\tilde{p}_{(I;K)} + i\zeta\tilde{t}_{(I;K)}\right]^2 + \Omega_{K,z}^2 (\tilde{t}_{(I;K)})^2 \right\}$$

with $\tilde{t}_{(R(I);K)} = \sqrt{2}\text{Re(Im) } \tilde{t}_K$ and $\tilde{p}_{(R(I);K)} = \sqrt{2}\text{Re(Im) } \tilde{p}_K$ that satisfy the commutation relation $\left[\tilde{t}_{(S;K)}, \tilde{p}_{(S';K')}\right] = -i\delta_{K,-K'}\delta_{SS'}$ with $S, S' = I, R$.

The RG flow is described by the imaginary time Schrodinger equation,

$$\tilde{\mathcal{H}}(z)\Psi\left[\tilde{t}, z\right] = -\frac{\partial}{\partial z}\Psi\left[\tilde{t}, z\right]. \tag{168}$$

The three parameters $\zeta$, $\sigma$ and $\alpha$ fully determine the solution $\Psi\left[\tilde{t}, z\right]$. The problem of the harmonic oscillator with time-dependent frequency has been studied extensively in Refs. [42–44], which is reviewed in Appendix D. We consider a UV theory obtained by adding

the single-trace and double-trace couplings to the reference theory $S_0$ in a translationally invariant way. In this case, the initial wavefunction is a Gaussian product state in the $K$-space. Because the Hamiltonian is non-interacting, $\Psi\left[\tilde{t}, z\right]$ remains Gaussian at all $z$. The solution is written as

$$\Psi\left[\tilde{t}, z\right] = \Psi_0\left[\tilde{t}_0, z\right] \prod_{K>0}' \left\{ \Psi_{(R;K)}\left[\tilde{t}_{(R;K)}, z\right] \Psi_{(I;K)}\left[\tilde{t}_{(I;K)}, z\right] \right\}, \tag{169}$$

where $\prod_{K>0}'$ runs over the half of the non-zero momenta. The wavefunction for each mode satisfies $\tilde{h}_s \Psi_s\left[\tilde{t}_s, z\right] = -\frac{\partial}{\partial z}\Psi_s\left[\tilde{t}_s, z\right]$, where the subscript $s$ stands for $0$, $(R;K)$ or $(I;K)$. The initial state can be written as

$$\Psi_s\left[\tilde{t}_s, 0\right] = \sum_m c_m \Psi_{m,s}\left[\tilde{t}_s, 0\right], \tag{170}$$

where $\left\{\Psi_{m,s}\left[\tilde{t}_s, 0\right]\right\}$ represents the eigenstates of the Hamiltonian $\tilde{h}_s$ at $z = 0$ and $\{c_m\}$ is a set of $z$-independent coefficients. Under the RG flow, the state evolves to

$$\Psi_s\left[\tilde{t}_s, z\right] = \sum_m c_m \Psi_{m,s}\left[\tilde{t}_s, z\right], \tag{171}$$

where

$$\Psi_{m,s}\left[\tilde{t}_s, z\right] = \frac{1}{\pi^{1/4}\sqrt{2^m m!}} e^{-\frac{1}{2}\Delta_{s,z}} \exp\left[-\frac{1}{2\Lambda_{s,z}}\tilde{t}_s^2\right] \times$$
$$\exp\left[\frac{\omega_{s,z}}{2A_{s,z}^2}\tilde{t}_s^2\right] H_m\left[-\frac{A_{s,z}}{\sqrt{\Omega_{s,0}}}\frac{\delta}{\delta \tilde{t}_s}\right] \exp\left[-\frac{\omega_{s,z}}{2A_{s,z}^2}\tilde{t}_s^2\right]. \tag{172}$$

Here $\omega_{s,z} = \left[\int_0^z \frac{dz'}{A_{s,z'}^2} + \frac{1}{\Omega_{s,0}}\right]^{-1}$. $A_{s,z}$ is a function that satisfies $\ddot{A}_{s,z} - A_{s,z}\Omega_{s,z}^2 = 0$ with $A_{s,0} = 1$ and $\dot{A}_{s,0} = 0$ ( $\dot{A} \equiv \partial_z A$). $e^{-\Delta_{s,z}} = \frac{\omega_{s,z}}{A_{s,z}\sqrt{\Omega_{s,0}}}$. $\frac{1}{\Lambda_{s,z}} = \zeta + \frac{\dot{A}_{s,z}}{A_{s,z}} + \frac{\omega_{s,z}}{A_{s,z}^2}$. At $z = 0$, $\frac{1}{\Lambda_{s,z}}$ is reduced to $\Omega_{s,0} + \zeta$, and $e^{-\Delta_{s,z}}$ becomes $\sqrt{\Omega_{s,0}}$.

Finally, the $z$-dependent state is written in terms of the variables in Eq. (156),

$$\Psi'[t', z] = \Psi_0\left[t_0', z\right] \prod_{K>0}' \Psi_{(R;K)}\left[t'_{(R;Ke^z)}, z\right] \Psi_{(I;K)}\left[t'_{(I;Ke^z)}, z\right]$$
$$= \Psi_0\left[t_0', z\right] \prod_{k>0}' \Psi_{(R;ke^{-z})}\left[t'_{(R;k)}, z\right] \Psi_{(I;ke^{-z})}\left[t'_{(I;k)}, z\right]. \tag{173}$$

Here we use $\tilde{t}_K = t'_k$ for the Fourier modes, where $x = e^{-z}X$ and $k = e^z K$ [15]. For a finite system size, $k$ and $K$ are discrete,

$$K = \frac{2\pi}{L}(n_1, n_2, .., n_D), \quad k = \frac{2\pi e^z}{L}(n_1, n_2, .., n_D), \tag{174}$$

where $L = V^{1/D}$ is the linear system size and $n_i$'s are integers.

Our next goal is to extract the fixed point of the full Wilsonian RG and local scaling operators with their scaling dimensions from the scale dependent state obtained from the quantum RG. As discussed in the previous sections, the asymptotic ground state that emerges in the large $z$ limit corresponds to the stable fixed point, and eigenstates with local excitations and eigenvalues give scaling operators and scaling dimensions, respectively. However, it is not easy to extract the asymptotic state in the large $z$ limit because the RG Hamiltonian is $z$-dependent. Even if one prepares an initial state to be an eigenstate of the instantaneous RG Hamiltonian at $z = 0$, the state does not remain the same under the RG evolution as is shown in Eq. (172). Therefore, we use the following strategy. Given that the RG Hamiltonian is invariant under the $\mathbb{Z}_2$ symmetry, we consider a generic initial state in each of the $\mathbb{Z}_2$ even sector and the $\mathbb{Z}_2$ odd sector. Under the quantum RG flow, those initial states evolve within each sector as

$$\lim_{z\to\infty} |\Psi^+(z)\rangle = \sum_n e^{-\mathcal{E}_n^+ z}|n; +\rangle,$$

$$\lim_{z\to\infty} |\Psi^-(z)\rangle = \sum_n e^{-\mathcal{E}_n^- z}|n; -\rangle, \tag{175}$$

where $|n, \pm\rangle$ corresponds to the eigenstates of the RG Hamiltonian that emerges in the large $z$ limit in each parity sector, and $\mathcal{E}_n^\pm$ is the corresponding eigenvalue. From this, we identify the eigenstate with the lowest eigenvalue in the even sector as the ground state that represents the stable IR fixed point. The excited states in each parity sector correspond to the states obtained by deforming the ground state with scaling operators with the corresponding $\mathbb{Z}_2$ parity and scaling dimension, $\mathcal{E}_n^\pm - \mathcal{E}_0^+$.

### 2. Fixed point

In this section, we identify the IR fixed point of the theory from quantum RG. As an initial state, we choose the ground state of the instantaneous RG Hamiltonian at $z = 0$,

---

[15] This follows from

$$\tilde{t}_K = \int^{1/\lambda} \frac{d^D X}{\sqrt{V}} e^{-iKX} \tilde{t}(X, z) = \int^{1/\lambda} \frac{d^D X}{\sqrt{V}} e^{-iKX} e^{-\frac{D}{2}z} t'(Xe^{-z}, z)$$

$$= \int^{e^{-z}/\lambda} \frac{d^D x}{\sqrt{Ve^{-Dz}}} e^{-ikx} t'(x, z) = t'_k.$$

which has the translational invariance and even $\mathbb{Z}_2$ parity,

$$\Psi\left[\tilde{t},0\right] = \Psi_{0,K=0}\left[\tilde{t}_0,0\right] \prod_{K>0}' \Psi_{0,(R;K)}[\tilde{t}_{(R;K)},0]\Psi_{0,(I;K)}[\tilde{t}_{(I;K)},0]. \tag{176}$$

In the large $z$ limit, the $z$-dependent wavefunction for each $s$-mode becomes

$$\Psi_{0,s}\left[\tilde{t}_s,z\right] = \pi^{-1/4}e^{-\frac{1}{2}\Delta_{s,z}}\exp\left[-\frac{1}{2\Lambda_{s,z}}\tilde{t}_s^2\right]. \tag{177}$$

The asymptotic many-body wavefunction is written as

$$\Psi\left[\tilde{t},z\right] = \mathcal{N}'(z)\exp\left[-\sum_K \frac{\tilde{t}_K\tilde{t}_{-K}}{2\Lambda_{K,z}}\right], \tag{178}$$

where $\mathcal{N}'(z) = \left[\prod_K \pi^{-1/4}e^{-\frac{1}{2}\Delta_{K,z}}\right]$ and $\Lambda_{K,z}$ are expressed in terms of $\alpha$, $\zeta$ and $\sigma$ as

$$\Lambda_{K,z} = \left[\mathbb{G}_{Ke^z}(\alpha,\sigma) + \zeta\right]^{-1}, \tag{179}$$

where

$$\mathbb{G}_k(\alpha,\sigma) = \frac{1}{2}\sqrt{\alpha}\frac{|k|}{I_\sigma\left[\sqrt{\alpha}|k|\right]}(I_{-1+\sqrt{\sigma}}\left[\sqrt{\alpha}|k|\right] + I_{1+\sqrt{\sigma}}\left[\sqrt{\alpha}|k|\right]) \tag{180}$$

in the large $z$ limit with fixed $k = Ke^z$ (see Appendix E for the details). This shows that $\Lambda_{K,z}$ converges to a $z$-independent function when viewed as a function of $k$. Physically, this is due to the fact that the scale invariance becomes manifest if one zooms in toward the $K = 0$ point progressively as $z$ increases. The overall normalization of the wavefunction decreases with increasing $z$ due to the damping associated with the imaginary time evolution [16].

To show that the wavefunction approaches a scale invariant asymptotic in the large $z$ limit, we need to go back to the scaled variable in Eq. (156). The wavefunction for $t'_k, p'_k$ is written as

$$\Psi'[t',z] = \mathcal{N}'(z)\exp\left(-\sum_k \frac{1}{2\tilde{\Lambda}_k}t'_k t'_{-k}\right), \tag{181}$$

---

[16] The normalization factor $\mathcal{N}'(z)$ is determined by

$$e^{-\Delta_{K,z}} \approx \frac{\sigma^{1/4}}{2\mathbb{A}_{Ke^z}(\alpha,\sigma)}e^{-\sqrt{\sigma}z}$$

in the large $z$ limit, where $\mathbb{A}_k$ is a function of $k = Ke^z$(Eq. (E3) in Appendix E),

$$\mathbb{A}_k(\alpha,\sigma) = -\frac{2^{-1+\sqrt{\sigma}}\pi\alpha^{-\frac{1}{2}\sqrt{\sigma}}}{\sin(\pi\sqrt{\sigma})}\frac{I_{\sqrt{\sigma}}\left[\sqrt{\alpha}|k|\right]}{\Gamma(-\sqrt{\sigma})}|k|^{-\sqrt{\sigma}}$$

where $\tilde{\Lambda}_k = \Lambda_{ke^{-z},z}$. In the large $z$ limit for a fixed $k$, $\tilde{\Lambda}_k$ takes the following forms (see Appendix E),

$$\tilde{\Lambda}_k = \begin{cases} \frac{1}{(\sqrt{\sigma}+\zeta)} & \text{for } |k| \ll 1, \\ \frac{1}{[\sqrt{\alpha}|k|+\zeta]} & \text{for } |k| \gg 1 \end{cases}. \tag{182}$$

This confirms that in the large $z$ limit $\Psi'[t',z]$ evolves to a $z$-independent state up to the $z$-dependent normalization factor.

Similar to what we studied in Sec. V A 1, the state in the large $z$ limit encodes the information on the IR fixed point. Defining $J_{2,k}^* \equiv \tilde{\Lambda}_k$, we rewrite the asymptotic state in the large $z$ limit as

$$\lim_{z \to \infty} |\Psi(z)\rangle = \mathcal{N}'(z) \int \mathcal{D}\phi \; e^{-S_0} \int_R \mathcal{D}t' e^{-\left(\frac{1}{2}\sum_k (J_{2,k}^*)^{-1} t'_k t'_{-k} - i \sum_k t'_k \mathcal{O}_{-k}\right)} |\phi\rangle$$

$$= \mathcal{N}(z) \int \mathcal{D}\phi \; e^{-S^*} |\phi\rangle, \tag{183}$$

where $\mathcal{N}(z) = \mathcal{N}'(z) \det [2\pi J_2^*]^{1/2}$ and the fixed point action $S^*$ is given by

$$S^* = S_0 + \frac{1}{2} \sum_k J_{2,k}^* \mathcal{O}_k \mathcal{O}_{-k} = S_0 + \frac{1}{2} \int d^D x d^D x' J_{2,x-x'}^* \mathcal{O}_x \mathcal{O}_{x'}, \tag{184}$$

where

$$J_{2,x-x'}^* = \frac{1}{Ve^{-Dz}} \sum_k J_{2,k}^* e^{-ik(x-x')} = \int \frac{d^D k}{(2\pi)^D} \tilde{\Lambda}_k e^{-ik(x'-x)}. \tag{185}$$

Here we use $\mathcal{O}_k = \frac{1}{\sqrt{Ve^{-Dz}}} \int d^D x \mathcal{O}_x e^{-ikx}$.

As is shown in Fig. 5 (see Appendix F for the details), $\lim_{z \to \infty} J_{2,x-x'}$ converges to a universal profile in the thermodynamic limit. $J_{2,x-x'}^*$ is peaked at $x - x' = 0$ with a finite width that is order of the short distance cutoff. It decays exponentially at large $|x' - x|$. We emphasize that the IR fixed point that exists away from the subspace of the single-trace couplings has been extracted solely from the $\beta$-functions that are defined in the subspace.

### 3. Scaling operators

In this subsection, we extract scaling operators from excited states of the RG Hamiltonian.

We first consider the $\mathbb{Z}_2$ odd sector. In the $\mathbb{Z}_2$ odd sector, we consider an initial state in which one of the Fourier modes is excited. Suppose that the mode with momentum $P$ is in the first excited state with respect to the RG Hamiltonian at $z = 0$, where the momentum

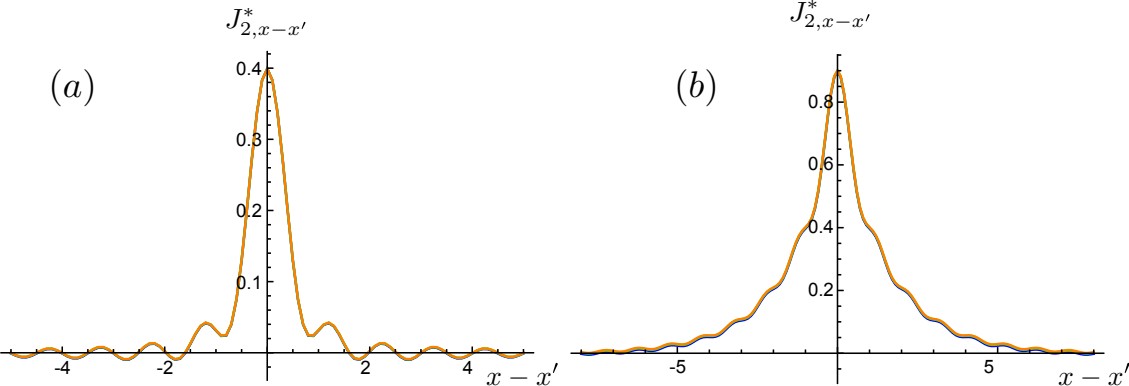

FIG. 5. (a) $J^*_{2,x-x'}$ as a function of $x - x'$ at $\sigma = 2.01$ and $\zeta = -0.1$ for $D = 1$ at $z = 22$ (orange), $z = 23$ (green), $z = 24$ (blue). For the computation, the lattice regularization is used with the total number of sites $L/a = e^{30}$. (b) $J^*_{2,x-x'}$ at $\sigma = 0.01$ and $\zeta = 0.1$ with the corresponding value of $z$ for each color as in (a). $\alpha = 1$.

is measured in the coordinate system defined at $z = 0$. In the large $z$ limit, the state evolves to (see Appendix G for derivation)

$$|\Psi_{1,P}(z)\rangle = (i\sqrt{2})\mathcal{N}(z) \int \mathcal{D}\phi \left( \frac{\sigma^{1/4}}{2\mathbb{A}_{Pe^z}} \tilde{\Lambda}_{Pe^z} e^{-\sqrt{\sigma}z} \mathcal{O}_{Pe^z} \right) e^{-S^*} |\phi\rangle. \tag{186}$$

$\mathcal{N}(z)$ is the normalization of the ground state defined in Eq. (183). Compared to the ground state, the weight of the first excited state with a definite momentum decays as $e^{-\sqrt{\sigma}z}$ in the large $z$ limit. The state that supports an excitation at $P \neq 0$ at $z = 0$ can not be invariant under the RG flow because a non-zero $P$ is pushed toward larger momenta in the large $z$ limit due to the rescaling. Namely, a source that is added periodically in space at UV flows to a periodic source with a shorter wavelength at larger $z$ when measured in the rescaled coordinate system.

The excited state with $P = 0$ is an exception. In the presence of a uniform source, the excited state flows to a scale invariant state in the large $z$ limit. Using $\mathcal{O}_p = \frac{1}{\sqrt{Ve^{-Dz}}} \int d^D x \, e^{-ipx} \mathcal{O}_x$ for the Fourier transformation at $z$, where $p = Pe^z$ and $x = Xe^{-z}$, we rewrite Eq. (186) for $P = 0$ as

$$|\Psi_{1,0}(z)\rangle = e^{-(\sqrt{\sigma}-D/2)z} \frac{i\sqrt{2}\mathcal{N}(z)\sigma^{1/4}\tilde{\Lambda}_0}{2\sqrt{V}\mathbb{A}_0} \int \mathcal{D}\phi \left( \int d^D x \, \mathcal{O}_x \right) e^{-S^*} |\phi\rangle. \tag{187}$$

The first excited state with the uniform source flows to $\int \mathcal{D}\phi \left( \int d^D x \, \mathcal{O}_x \right) e^{-S^*} |\phi\rangle$ in the large $z$ limit with the $z$-dependent amplitude, $e^{-(\sqrt{\sigma}-D/2)z}$ relative to the ground state. This implies that the spatially uniform deformation of the $\mathbb{Z}_2$ odd single-trace operator is relevant (irrelevant) if $\sqrt{\sigma} < D/2$ ($\sqrt{\sigma} > D/2$).

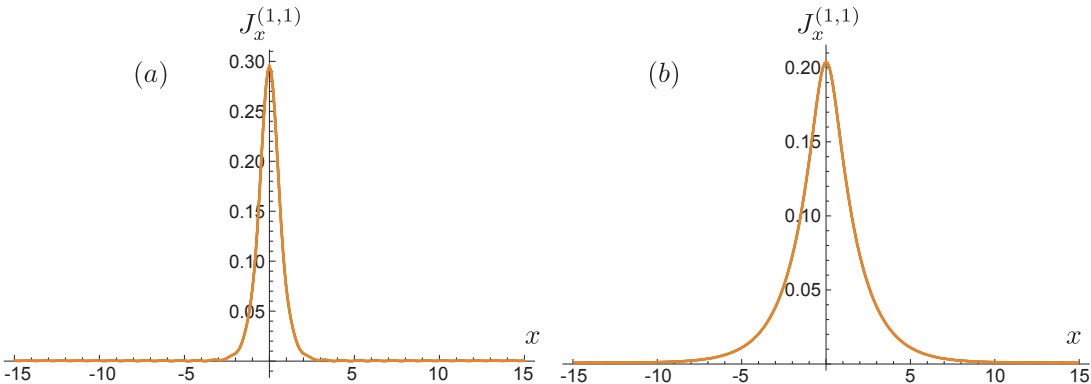

FIG. 6. $J_x^{(1,1)}$ at $z = 22$ (orange), $z = 23$ (green), $z = 24$ (blue) with $L/a = e^{30}$ in $D = 1$. We set parameters to be $\alpha = 1$ and (a) $\sigma = 2.01$ and $\zeta = -0.1$; (b) $\sigma = 0.01$ and $\zeta = 0.1$.

The other type of eigenstates that are invariant under the RG evolution is the ones that support excitations localized in space. In order to find local scaling operators associated with states with local excitations, we consider an initial state in which the single-trace operator is inserted at $X$. For $z > 0$, the state becomes

$$|\Psi_{1,X}(z)\rangle = \frac{1}{\sqrt{V}} \sum_P e^{iPX} |\Psi_{1,P}(z)\rangle. \tag{188}$$

In the large $z$ limit, the state evolves to [17]

$$\lim_{z\to\infty} |\Psi_{1,X}(z)\rangle = (i\sqrt{2})\mathcal{N}(z)e^{-(\sqrt{\sigma}+D/2)z} \int \mathcal{D}\phi \; \hat{\mathcal{A}}_1(Xe^{-z})e^{-S^*}|\phi\rangle, \tag{189}$$

where

$$\hat{\mathcal{A}}_1(x) = \int d^D x' J_{x-x'}^{(1,1)} \mathcal{O}_{x'} \tag{190}$$

with

$$J_x^{(1,1)} = \int \frac{d^D p}{(2\pi)^D} \frac{\sigma^{1/4}}{2\mathbb{A}_p} \tilde{\Lambda}_p e^{ipx}. \tag{191}$$

---

[17]
$$\lim_{z\to\infty} |\Psi_{1,X}(z)\rangle = \frac{(i\sqrt{2})\mathcal{N}(z)}{\sqrt{V}} \int \mathcal{D}\phi \left( \sum_P \frac{\sigma^{1/4}}{2\mathbb{A}_{Pe^z}} \tilde{\Lambda}_{Pe^z} e^{-\sqrt{\sigma}z} \mathcal{O}_{Pe^z} e^{iPX} \right) e^{-S^*}|\phi\rangle$$

$$= \frac{(i\sqrt{2})\mathcal{N}(z)e^{-\sqrt{\sigma}z}}{\sqrt{V}} \int \mathcal{D}\phi \left( \sum_p \frac{\sigma^{1/4}}{2\mathbb{A}_p} \tilde{\Lambda}_p \mathcal{O}_p e^{ip(Xe^{-z})} \right) e^{-S^*}|\phi\rangle$$

$$= \frac{(i\sqrt{2})\mathcal{N}(z)e^{-\sqrt{\sigma}z}}{V e^{-Dz/2}} \int \mathcal{D}\phi \left( \int d^D x' \sum_p \frac{\sigma^{1/4}}{2\mathbb{A}_p} \tilde{\Lambda}_p e^{ip(e^{-z}X-x')} \mathcal{O}_{x'} \right) e^{-S^*}|\phi\rangle$$

$$= (i\sqrt{2})\mathcal{N}(z)e^{-(\sqrt{\sigma}+D/2)z} \int \mathcal{D}\phi \; \hat{\mathcal{A}}_1(Xe^{-z})e^{-S^*}|\phi\rangle,$$

where we used $\frac{1}{Ve^{-Dz}} \sum_p = \int \frac{d^D p}{(2\pi)^D}$ at $z$.

$\hat{\mathcal{A}}_1(x)$ inserts a single-trace operator around $x$ with distribution given by $J^{(1,1)}_{x-x'}$. Henceforth, we use $J^{(n,m)}$ to denote the contribution of the $m$-trace operator to the $n$-th scaling operator. The local operator inserted at $X$ at the UV boundary evolves to a distribution of local operators centered at $Xe^{-z}$ at $z > 0$. The shift of the central position is due to the rescaling of the space. In the large $z$ limit, the local operator evolves to $\hat{\mathcal{A}}_1(0)$, which we identify as the local scaling operator inserted at the origin. The broadening of the distribution in $J^{(1,1)}_x$ is due to the correlation in the fluctuations of the single-trace coupling at the fixed point. Irrespective of the initial profile of the local operator at the UV, it converges to the universal profile $J^{(1,1)}_x$ at large $z$ in the thermodynamic limit. In Fig. 6, we numerically plot $J^{(1,1)}_x(z)$ which converges to a $z$-independent profile in the large $z$ limit. Compared to the ground state, the overall weight of the first excited state with the local excitation decays as $e^{-(\sqrt{\sigma}+D/2)z}$ in the large $z$ limit. This implies that the operator $\mathcal{O}_x$ has scaling dimension $\Delta_1 = \sqrt{\sigma} + D/2$. This is consistent with the fact that the operator is relevant (irrelevant) if $\sqrt{\sigma} < D/2$ ($\sqrt{\sigma} > D/2$).

If we impose the $\mathbb{Z}_2$ symmetry, the $\mathbb{Z}_2$ odd operator is not allowed. To see if the low-energy fixed point is stable in the presence of the $\mathbb{Z}_2$ symmetry, we need to consider local scaling operators in the even parity sector. The operator with the smallest scaling dimension in the even sector is the identity operator. In the following section, we obtain the next lowest scaling operator in the even sector.

Now, let us consider the $\mathbb{Z}_2$ even sector. Excited states in the $\mathbb{Z}_2$ even sector should include even number of excited modes. Let us consider an initial state with two excited modes labelled by momenta $P$ and $P'$. Under the RG evolution, the state in general evolves into a linear superposition of the ground state (for $P + P' = 0$) and excited states. Since we are interested in the excited state above the ground state, we discard the slowest decaying state (the ground state). The state with the next slowest decaying amplitude in the large $z$ limit is given by (see Appendix H)

$$
\begin{aligned}
|\Psi_{2,P,P'}(z)\rangle = -\sqrt{2}\mathcal{N}(z)e^{-2\sqrt{\sigma}z} \int \mathcal{D}\phi \, \Big( &\frac{\sigma^{1/2}}{4\mathbb{A}_{Pe^z}\mathbb{A}_{P'e^z}} \tilde{\Lambda}_{Pe^z}\tilde{\Lambda}_{P'e^z}\mathcal{O}_{Pe^z}\mathcal{O}_{P'e^z} \\
&- \delta_{P+P',0}\left[\frac{\sigma^{1/2}}{4\mathbb{A}^2_{Pe^z}}\tilde{\Lambda}_{Pe^z} + \frac{1}{2}\frac{\sqrt{\sigma}}{\sqrt{\alpha P^2 + \sigma}}\mathbb{W}_{Pe^z}\right] \Big)e^{-S^*}|\phi\rangle,
\end{aligned}
\tag{192}
$$

where $\mathcal{N}(z)$ encodes the rate at which the ground state decays and $e^{-2\sqrt{\sigma}z}$ is the additional decay for the next slowest decaying state. Again, the state with non-zero momenta can not be invariant under the RG evolution due to the rescaling that shifts momenta to larger values with increasing $z$. To find a local scaling operator, we consider the state that evolves from an initial state that supports local excitations at positions $X$ and $X'$,

$$
|\Psi_{2,X,X'}(z)\rangle = \frac{1}{V}\sum_{P,P'} e^{iPX+iP'X'}|\Psi_{2,P,P'}(z)\rangle.
\tag{193}
$$

In the large $z$ limit, the state evolves to [18]

$$|\Psi_{2,X,X'}(z)\rangle = -\sqrt{2}\mathcal{N}(z)e^{-(2\sqrt{\sigma}+D)z}\int \mathcal{D}\phi \; \hat{\mathcal{A}}_2(Xe^{-z}, X'e^{-z})e^{-S^*}|\phi\rangle, \tag{194}$$

where

$$\hat{\mathcal{A}}_2(x,x') = \int d^D y d^D y' J^{(2,2)}_{x-y,x'-y'}\mathcal{O}_y\mathcal{O}_{y'} - J^{(2,0)}_{x-x'}\mathbb{1} \tag{195}$$

with

$$J^{(2,2)}_{x,x'} = \int \frac{d^D p d^D p'}{(2\pi)^{2D}} \frac{\sigma^{1/2}}{4\mathbb{A}_p\mathbb{A}_{p'}}\tilde{\Lambda}_p\tilde{\Lambda}_{p'}e^{ipx+ip'x'},$$

$$J^{(2,0)}_x = \int \frac{d^D p}{(2\pi)^D} e^{ipx}\left[\frac{\sigma^{1/2}}{4\mathbb{A}_p^2}\tilde{\Lambda}_p + \frac{1}{2}\mathbb{W}_p\right]. \tag{196}$$

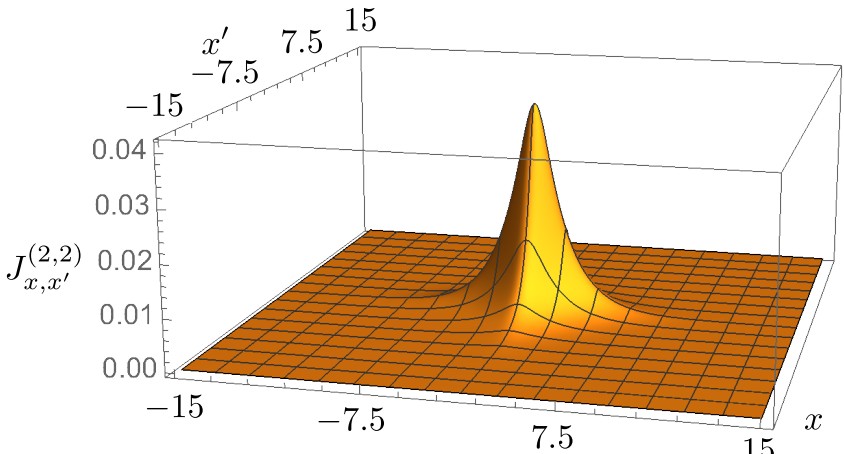

FIG. 7. The profile of $J^{(2,2)}_{x,x'}$ as a function of $x$ and $x'$ at $z = 25$ for a system with $L/a = e^{30}$ in $D = 1$. We use $\alpha = 1$, $\zeta = 0.1$ and $\sigma = 0.01$ for the plot.

---

[18]

$$|\Psi_{2,X,X'}(z)\rangle = \frac{1}{V}\sum_{P,P'} e^{iPX+iP'X'}|\Psi_{2,P,P'}(z)\rangle$$

$$= -\sqrt{2}\mathcal{N}(z)e^{-2\sqrt{\sigma}z}\int \mathcal{D}\phi\left(\frac{1}{V}\sum_{p,p'} e^{ip(Xe^{-z})+ip'(X'e^{-z})}J^{(2,2)}_{p,p'}\mathcal{O}_p\mathcal{O}_{p'} - \frac{1}{V}\sum_p e^{ip(X-X')e^{-z}}J^{(2,0)}_p\mathbb{1}\right)e^{-S^*}|\phi\rangle$$

$$= -\sqrt{2}\mathcal{N}(z)e^{-(2\sqrt{\sigma}+D)z}\int \mathcal{D}\phi\left(\int d^D y d^D y'\left[\frac{1}{V^2 e^{-2Dz}}\sum_{p,p'} e^{ip(Xe^{-z}-y)+ip'(X'e^{-z}-y')}J^{(2,2)}_{p,p'}\right]\mathcal{O}_y\mathcal{O}_{y'}\right)e^{-S^*}|\phi\rangle$$

$$+ \sqrt{2}\mathcal{N}(z)e^{-(2\sqrt{\sigma}+D)z}\int \mathcal{D}\phi\left(\frac{1}{V e^{-Dz}}\sum_p e^{ip(X-X')e^{-z}}J^{(2,0)}_p\mathbb{1}\right)e^{-S^*}|\phi\rangle,$$

where $p = Pe^z$ and

$$J^{(2,2)}_{p,p'} = \frac{\sigma^{1/2}}{4\mathbb{A}_p\mathbb{A}_{p'}}\tilde{\Lambda}_p\tilde{\Lambda}_{p'}, \quad J^{(2,0)}_p = \frac{\sigma^{1/2}}{4\mathbb{A}_p^2}\tilde{\Lambda}_p + \frac{\sqrt{\sigma}}{2\sqrt{\alpha p^2 e^{-2z}+\sigma}}\mathbb{W}_p \approx \frac{\sigma^{1/2}}{4\mathbb{A}_p^2}\tilde{\Lambda}_p + \frac{1}{2}\mathbb{W}_p.$$

in the large $z$ limit.

$\hat{\mathcal{A}}_2(x, y)$ is a composite operator that supports two single-trace operators centered at position $x$ and $y$ respectively. In the large $z$ limit, the initial state flows to the state obtained by applying $\hat{\mathcal{A}}_2(0, 0)$ to the ground state. Therefore, we identify $\hat{\mathcal{A}}_2(0, 0)$ as the lowest scaling operator above the identity operator in the even sector. It is noted that $\hat{\mathcal{A}}_2(0, 0)$ is a linear superposition of a double-trace operator and an identity operator. This is because the double-trace operator and the identity operator mix under the RG flow. $J_{x,x'}^{(2,2)}$, that describes the distribution of the two single-trace operators, can be written as $J_{x,x'}^{(2,2)} = J_x^{(1,1)} J_{x'}^{(1,1)}$, and its profile in the real space is determined by that of $J_x^{(1,1)}$. In Fig. 7, $J_{x,x'}^{(2,2)}$ is shown as a function of $x$ and $x'$ at a fixed $z$. It has a peak at the origin and decays exponentially away from the peak with the width that is comparable to the short distance cutoff scale. According to Eq. (194), the local deformation induced by this scaling operator decays with rate $2\sqrt{\sigma} + D$ relative to the ground state. Thus, its scaling dimension of the local operator is $2\sqrt{\sigma} + D$ which is twice of the single-trace operator. It is an irrelevant operator due to $\sqrt{\sigma} > 0$.

It can be shown that general scaling operators have scaling dimensions given by $n\left(\sqrt{\sigma} + \frac{D}{2}\right)$ for $n = 1, 2, 3, \ldots$ See Appendix I for the details. All operators in the even sector are irrelevant for $\sqrt{\sigma} > 0$. This shows that the fixed point in Eq. (183) is stable in the presence of the $\mathbb{Z}_2$ symmetry. The fact that the scaling dimensions are additive is a feature of the generalized free theory for which the bulk RG Hamiltonian is quadratic. For general theories whose RG Hamiltonian is not quadratic, this is no longer the case. It will be of great interest to consider large $N$ theories whose RG Hamiltonian includes interactions that are suppressed by $1/N$, and compute $1/N$ corrections to the scaling dimensions from the quantum RG.

### 4. Operator Product Expansion

Now we consider the OPE between two parity-odd operators with the lowest scaling dimension. For this, we insert $\hat{\mathcal{A}}_1$ at $X$ and $-X$ in the stable fixed point theory. The deformed theory corresponds to the initial state with two local excitations,

$$|\Psi_{1\times 1; X, -X}(0)\rangle = -\sqrt{2}\mathcal{N}(0) \int \mathcal{D}\phi \hat{\mathcal{A}}_1(X)\hat{\mathcal{A}}_1(-X)e^{-S^*}|\phi\rangle. \tag{197}$$

Then, following the steps explained in Sec. IV C 4, we evolve the state with the RG Hamiltonian for $z = \ln\left|\frac{X}{a}\right|$ to obtain

$$|\Psi_{1\times 1; X, -X}(z)\rangle = -\sqrt{2}\mathcal{N}(z)e^{-(2\sqrt{\sigma}+D)z} \int \mathcal{D}\phi \hat{\mathcal{A}}_1(a)\hat{\mathcal{A}}_1(-a)e^{-S^*}|\phi\rangle$$

$$= -\sqrt{2}\mathcal{N}(z)e^{-(2\sqrt{\sigma}+D)z} \int \mathcal{D}\phi \int d^D y d^D y' J_{a-y}^{(1,1)} J_{-a-y'}^{(1,1)} \mathcal{O}_y \mathcal{O}_{y'} e^{-S^*}|\phi\rangle, \tag{198}$$

where $a$ is the short distance cutoff length scale. In the second equality, we use the expression for $\hat{\mathcal{A}}_1$ in Eq. (190). By using Eq. (195) for $\hat{\mathcal{A}}_2$, we rewrite Eq. (198) as [19]

$$|\Psi_{1\times 1;X,-X}(z)\rangle = -\sqrt{2}\mathcal{N}(z)e^{-(2\sqrt{\sigma}+D)z}\int \mathcal{D}\phi\left[J_{2a}^{(2,0)}\mathbb{1} + \hat{\mathcal{A}}_2(a,-a)\right]e^{-S^*}|\phi\rangle. \quad (199)$$

We expand $\hat{\mathcal{A}}_2(a,-a)$ in $a$ to obtain

$$|\Psi_{1\times 1;X,-X}(z)\rangle = -\sqrt{2}\mathcal{N}(z)e^{-(2\sqrt{\sigma}+D)z}\int \mathcal{D}\phi\left[J_{2a}^{(2,0)}\mathbb{1} + \hat{\mathcal{A}}_2(0,0) + ...\right]e^{-S^*}|\phi\rangle, \quad (200)$$

where the ellipsis includes derivative terms such as $a\partial_x\hat{\mathcal{A}}_2(x,y)|_{x=y=0}$ and $-a\partial_y\hat{\mathcal{A}}_2(x,y)|_{x=y=0}$. Finally, the backward evolution for RG time $-z$ restores the initial state,

$$|\Psi_{1\times 1;X,-X}(0)\rangle = -\sqrt{2}\mathcal{N}(0)\int \mathcal{D}\phi\left[e^{-(2\sqrt{\sigma}+D)z}J_{2a}^{(2,0)}\mathbb{1} + \hat{\mathcal{A}}_2(0,0) + ...\right]e^{-S^*}|\phi\rangle. \quad (201)$$

Comparing this with Eq. (197), we obtain the OPE for two $\hat{\mathcal{A}}_1$ operators inserting at $X$ and $-X$ as

$$\hat{\mathcal{A}}_1(X)\times \hat{\mathcal{A}}_1(-X) = J_{2a}^{(2,0)}\left|\frac{a}{X}\right|^{2\Delta_1}\hat{\mathcal{A}}_0 + \hat{\mathcal{A}}_2(0,0) + ..., \quad (202)$$

where $\hat{\mathcal{A}}_0 = \mathbb{1}$ and $\Delta_1 = \sqrt{\sigma} + D/2$. Eq. (202) shows the channels in which two single-trace operators fuse into a local double-trace operator with spin 0 and the identity operator. The ellipsis includes double-trace operators with larger spins and descendants. Following the procedure explained in Sec. IV C 5, one can compute the $\beta$-functions for general multi-trace couplings.

## VI.   CONCLUSION AND DISCUSSION

In this paper, we show that the full $\beta$-function of the exact Wilsonian RG is completely fixed by the $\beta$-function defined in the subspace of single-trace couplings. We establish this general constraints on $\beta$-functions using the quantum RG, which is an exact reformulation of the Wilsonian RG. In quantum RG, the conventional RG flow in the space of couplings is replaced with a quantum evolution of a wavefunction defined in the subspace of single-trace couplings, where fluctuations of the dynamical single-trace couplings encode the information about all multi-trace couplings. Since the quantum evolution of the RG flow is completely determined from the $\beta$-functions defined in the subspace of single-trace couplings, the full Wilsonian $\beta$-functions can be extracted from the $\beta$-functions defined in the subspace. This

---

[19] $F_{n,n',l}(x)$ introduced in Eq. (125) is independent of $x$ because the RG Hamiltonian is quadratic in this case.

is used to compute the full $\beta$-functions of two concrete models : the $O(N)$ vector model and the $O_L(N) \times O_R(N)$ matrix model.

We also provide the general algorithm for extracting other field theory data such as scaling operators and OPE. The general procedure consists of two steps. First, we construct the RG Hamiltonian that generates the quantum RG flow from the $\beta$-functions defined in the subspace of the single-trace couplings. Second, we establish the correspondence between the ground state of the RG Hamiltonian with the stable IR fixed point. Similarly, excited states with local excitations are mapped to the IR fixed point deformed with corresponding local operators. The energies of the excited states determine the scaling dimensions of the local operators. From the completeness of the eigenstates of the RG Hamiltonian, one can also extract the OPE coefficients among general operators and reconstruct the full $\beta$-functions.

We conclude with open questions and future directions. First, QRG can be used to compute the exact quantum effective action. The scale dependence of the quantum effective action obeys the exact RG equation[6, 8]. In general solving the exact RG equation is challenging because the exact effective action includes operators made of arbitrarily many fields and derivatives. As a result, exact effective actions remain unknown even for relatively simple theories. In QRG, the exact RG equation is mapped to a quantum evolution of a wavefunction for single-trace couplings. Since the set of single-trace operators is far smaller than the set of all possible operators, QRG can be potentially more tractable. For general quantum field theories, it is still difficult to solve the corresponding quantum evolution problem in QRG. However, in the large $N$ limit, quantum fluctuations of the single-trace couplings become weak, and the theory that describes QRG evolution becomes classical. In the large $N$ limit, the solution to the exact RG equation can be obtained from the saddle-point solution. Recently, the exact effective action for the $O(N)$ vector model has been computed from QRG in the large $N$ limit[34]. It would be of great interest to compute exact effective actions for matrix models in the large $N$ limit. Second, QRG provides a concrete prescription for constructing the holographic duals for general quantum field theories[16]. The construction gives a well-defined bulk theory that includes dynamical gravity as far as the boundary theory is regularized[15, 30, 39]. However, the continuum limit of the bulk theory obtained from regularized boundary theories such as lattice models is not fully understood. It is of great interest to understand how the regularized bulk theory obtained from QRG is related to continuum theories conjectured as holographic duals of known field theories in the semi-classical limit. Third, the $\beta$-functions in the subspace of single-trace couplings include the information on all fixed points that exist away from the subspace as is discussed in Sec. V A. As multiple fixed points collide with a parameter of the theory tuned, the stable fixed point can disappear in the space of real couplings. It will be of interest to understand how the loss of conformality or an appearance of non-unitary fixed

points [45–48] manifests itself in the bulk. Finally, it would be interesting to consider cases in which the bulk RG Hamiltonian supports multiple ground states. One can consider a few scenarios in which degenerate ground states for the RG Hamiltonian arise. Degenerate ground states can be related to each other through symmetry, in which case the emergence of degenerate ground state is a sign of a spontaneous symmetry breaking. A degeneracy can also arise due to a suppression of tunneling between topologically distinct RG paths[39]. An exactly marginal deformation can also give rise to degenerate ground states.

## ACKNOWLEDGEMENT

Research at Perimeter Institute is supported in part by the Government of Canada through the Department of Innovation, Science and Economic Development Canada and by the Province of Ontario through the Ministry of Colleges and Universities. SL acknowledges the support of the Natural Sciences and Engineering Research Council of Canada.

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

## Appendix A: Wilsonian RG with Gaussian action as reference action

In this appendix, we show the form of an RG Hamiltonian that generates the exact RG flow for the $\phi^4$- theory starting from the deformed Gaussian action[6, 30]. Let us consider

a $D$-dimensional scalar field theory whose Euclidean action is written as

$$S = S_0 + S_1. \tag{A1}$$

Here $S_0$ is a quadratic reference action,

$$S_0 = \frac{1}{2} \int d^D k \, G_\Lambda^{-1}(k) \phi_k \phi_{-k}, \tag{A2}$$

where $k$ is momentum, and $G_\Lambda^{-1}(k) = e^{\frac{k^2}{\Lambda^2}} k^2$ is a regularized kinetic term that suppresses fluctuations at momenta larger than UV cutoff $\Lambda$. $S_1$ is a deformation that includes interactions and higher derivative terms. The standard exact RG flow equation can be obtained by lowering the UV cut-off as $\Lambda \to \Lambda e^{-dz}$ followed by a rescaling of field and momentum, $\phi_k \to e^{\frac{D+2}{2} dz} \phi_{e^{dz}k}$[6]. The correction to the effective action generated from this coarse graining is obtained by applying an RG Hamiltonian to the wavefunction $e^{-S}$ as

$$e^{-Hdz} e^{-S} = e^{-S - \delta S}, \tag{A3}$$

where $\delta S$ is the correction to the effective action, and

$$\hat{H} = e^{-S_0} \int d^D k \left[ \frac{\tilde{G}(k)}{2} \pi_k \pi_{-k} - i \left( \frac{D+2}{2} \phi_k + k \partial_k \phi_k \right) \pi_{-k} + C \right] e^{S_0} \tag{A4}$$

is the RG Hamiltonian. Here $\pi_k = -i \frac{\delta}{\delta \phi_{-k}}$ is the conjugate momentum of $\phi_k$. $\tilde{G}(k) = \frac{\partial G_\Lambda(k)}{\partial \ln \Lambda}$ is the propagator of the high-energy modes that are integrated out in the coarse graining scheme. $C = -\int d^D k \delta^D(0) \left[ \frac{\tilde{G}}{2} G_\Lambda^{-1} + 1 \right]$ is a constant. One can check that the RG Hamiltonian leaves the trivial state invariant, $\langle \mathbb{1} | H = 0$, and the partition function is invariant under the RG evolution,

$$\langle \mathbb{1} | S \rangle = \langle \mathbb{1} | e^{-dzH} | S \rangle. \tag{A5}$$

**Appendix B: Solving Non-Hermitian Harmonic oscillator**

The RG Hamiltonian considered in the paper takes the following form,

$$H = \frac{1}{2m} \boldsymbol{\pi}_x^2 + \frac{1}{2} m \omega^2 (\boldsymbol{x} + i\gamma \boldsymbol{\pi}_x)^2, \tag{B1}$$

where $\boldsymbol{x}$ ($\boldsymbol{\pi}_x$) corresponds to $p$ ($j$) in the RG Hamiltonian in the main text. The conjugate variables satisfy the commutation relation $[\boldsymbol{x}, \boldsymbol{\pi}_x] = i$. The RG Hamiltonian is invariant under the $\mathcal{P}$ and $\mathcal{T}$ symmetries,

$$\begin{aligned}
\mathcal{P} \boldsymbol{x} \mathcal{P} = -\boldsymbol{x}, \quad \mathcal{P} \boldsymbol{\pi}_x \mathcal{P} = -\boldsymbol{\pi}_x, \quad \mathcal{P} i \mathcal{P} = i, \\
\mathcal{T} \boldsymbol{x} \mathcal{T} = \boldsymbol{x}, \quad \mathcal{T} \boldsymbol{\pi}_x \mathcal{T} = -\boldsymbol{\pi}_x, \quad \mathcal{T} i \mathcal{T} = -i.
\end{aligned} \tag{B2}$$

Under the similarity transformation $S = e^{\frac{\gamma}{2}\boldsymbol{\pi}_x^2}$, the non-Hermitian RG Hamiltonian is transformed to an Hermitian RG Hamiltonian as

$$H_0 \equiv SHS^{-1} = \left[\frac{1}{2m}\boldsymbol{\pi}_x^2 + \frac{1}{2}m\omega^2\boldsymbol{x}^2\right]. \tag{B3}$$

In terms of the $n$-th eigenstate $|\psi_n\rangle$ of $H_0$, the right and left eigenstates of $H$ are given by

$$|\psi_n^R\rangle = S^{-1}|\psi_n\rangle, \qquad \langle\psi_n^L| = \langle\psi_n|S \tag{B4}$$

with eigenvalue $(n + \frac{1}{2})\omega$. Their wavefunctions in the $\boldsymbol{\pi}_x$ basis are given by

$$\begin{aligned}
\psi_n^R(\boldsymbol{\pi}_x) &= \langle\boldsymbol{\pi}_x|\psi_n^R\rangle = \mathcal{N}^R e^{-\frac{1}{2}(\frac{1}{m\omega}+\gamma)\boldsymbol{\pi}_x^2} H_n\left(\frac{\boldsymbol{\pi}_x}{\sqrt{m\omega}}\right), \\
\psi_n^L(\boldsymbol{\pi}_x) &= \langle\boldsymbol{\pi}_x|\psi_n^L\rangle = \mathcal{N}^L e^{-\frac{1}{2}(\frac{1}{m\omega}-\gamma)\boldsymbol{\pi}_x^2} H_n\left(\frac{\boldsymbol{\pi}_x}{\sqrt{m\omega}}\right),
\end{aligned} \tag{B5}$$

where $\mathcal{N}^{R(L)}$ are the normalization constants. The right (left) eigenstates are normalizable if $\gamma > -\frac{1}{m\omega}$ ($\gamma < \frac{1}{m\omega}$).

From the standard ladder operators of the Hermitian Hamiltonian,

$$a^\dagger = \sqrt{\frac{m\omega}{2}}\left(\boldsymbol{x} - \frac{i}{m\omega}\boldsymbol{\pi}_x\right), \quad a = \sqrt{\frac{m\omega}{2}}\left(\boldsymbol{x} + \frac{i}{m\omega}\boldsymbol{\pi}_x\right), \tag{B6}$$

the raising and lowering operators for the non-Hermitian Hamiltonian can be obtained via the similarity transformation,

$$\begin{aligned}
a_1 &= S^{-1}a^\dagger S = \sqrt{\frac{m\omega}{2}}\left[\boldsymbol{x} + i(\gamma - \frac{1}{m\omega})\boldsymbol{\pi}_x\right], \\
a_2 &= S^{-1}aS = \sqrt{\frac{m\omega}{2}}\left[\boldsymbol{x} + i(\gamma + \frac{1}{m\omega})\boldsymbol{\pi}_x\right].
\end{aligned} \tag{B7}$$

The non-Hermitian RG Hamiltonian can be written in terms of $a_1$ and $a_2$ as

$$H = \omega a_1 a_2 + \frac{\omega}{2}. \tag{B8}$$

**Appendix C: $\beta$-functions for the $0$-dimensional model in limiting cases**

In this appendix, we analyze the RG flow of the 0-dimensional example in various limits.

- $w \to 0^-$ limit with $a < 0$ and $b > 0$ :

  In the $w \to 0^-$ limit, $\xi_R$ approaches $0^+$. In this limit, the two fixed points are pushed to the region with large $J_2$ as is shown in Fig. 4(a). As $\xi_R$ vanishes in the small $w$ limit, the eigenstates of the RG Hamiltonian become non-normalizable.

- $w \to 0^-$ limit with $a > 0$ and $b > 0$ :

  In this case, $\xi_R = \frac{a}{2b}$ and $\varepsilon = \frac{a}{b}$ are finite and positive, and the right wave function is normalizable. Because $\varepsilon - 2\xi_R = 0$, the $\beta$-functions become

  $$\frac{d}{dz}J_1 = -\sqrt{\eta}J_1 + O\left((J - J^*)^3\right),$$
  $$\frac{d}{dz}J_2 = \frac{\sqrt{\eta}}{2\xi_R} - 2\sqrt{\eta}J_2 + O\left((J - J^*)^3\right).$$

  The unstable fixed point is pushed to positive infinity, and only the stable fixed point located at $(J_1^*, J_2^*)_- = \left(0, \frac{1}{4\xi_R}\right)$ survives.

- $b \to 0^+$ limit with $w < 0$ and $a < 0$ :

  In the small $b$ limit, $\xi_R = \frac{1}{4b}(|a| + a - \frac{2bw}{|a|})$ and $\varepsilon - 2\xi_R = \frac{1}{4b}(|a| - a - \frac{2bw}{|a|})$. For $a < 0$, $\xi_R = \frac{w}{2a}$ and $\varepsilon - 2\xi_R \approx -\frac{a}{2b} \to \infty^+$. The $\beta$-functions in Eq. (149) and Eq. (150) become

  $$\frac{d}{dz}J_1 = -\sqrt{\eta}\frac{\sqrt{2}(\varepsilon - 2\xi_R)}{\xi_R}\left[\frac{2\sqrt{2}\xi_R^2 + (\varepsilon - 2\xi_R)}{4(\varepsilon - 2\xi_R)\xi_R} - J_2\right]J_1 + O\left((J - J^*)^3\right),$$
  $$\frac{d}{dz}J_2 = \sqrt{2\eta}\frac{\varepsilon - 2\xi_R}{\xi_R}\left(J_2 - \frac{1}{4\xi_R}\right)^2 - 2\sqrt{\eta}\left(J_2 - \frac{1}{4\xi_R}\right) + O\left((J - J^*)^3\right).$$

  As $\varepsilon - 2\xi_R$ approaches infinity in the small $b$ limit, the second term in $\frac{d}{dz}J_2$ is negligible. Thus, the two fixed points are getting closer to each other until they collide at $(J_1^*, J_2^*)_\pm = \left(0, \frac{1}{4\xi_R}\right)$. This is shown in Fig. 4(c$_{1,2}$). Before collision, the stable (unstable) fixed point has two irrelevant (relevant) directions. As the fixed points collide, both directions become marginal because the negative and positive scaling dimensions can meet only at 0. It turns out that the perturbations are marginally irrelevant from one side and marginally relevant from the other side. If $b$ becomes negative, the normalizable ground state disappears, which suggests a loss of stable fixed point in the real space of the couplings [45–48]. It will be of interest to understand constraints on the range of conformality from unitarity.

- $b \to 0^+$ limit with $w < 0$ and $a > 0$ :

  If $a > 0$, $\xi_R = \frac{a}{2b} \to \infty^+$ and $\varepsilon - 2\xi_R = -\frac{w}{2a}$ is finite. In the $b \to 0^+$ limit, the unstable fixed point moves towards the positive infinity, while the stable fixed point moves to the origin. The $\beta$-functions in this limit are given by

  $$\frac{d}{dz}J_1 \approx -\sqrt{\eta}J_1 + O\left((J - J^*)^3\right),$$
  $$\frac{d}{dz}J_2 \approx -2\sqrt{\eta}J_2 + O\left((J - J^*)^3\right),$$

where $\eta \approx a^2$.

## Appendix D: Solution to the Schrodinger equation for the time-dependent harmonic oscillator

We first review the evolution of a time-dependent harmonic oscillator with Hamiltonian

$$H = \frac{1}{2}\boldsymbol{x}^2 + \frac{1}{2}\Omega_z^2 \boldsymbol{\pi}_x^2, \tag{D1}$$

where $\Omega_z$ is a function of imaginary time $z$. An initial state can be written as a superposition of the eigenstates of the instantaneous Hamiltonian at $z = 0$ as $\Psi[\boldsymbol{\pi}_x, 0] = \sum_m c_m \Psi_m[\boldsymbol{\pi}_x, 0]$, where

$$\Psi_m[\boldsymbol{\pi}_x, 0] = \frac{1}{\sqrt{2^n n!}} \left(\frac{\Omega_0}{\pi}\right)^{1/4} H_n\left[\sqrt{\Omega_0}\boldsymbol{\pi}_x\right] \exp\left[-\frac{1}{2}\Omega_0 \boldsymbol{\pi}_x^2\right]. \tag{D2}$$

The time dependent state satisfies the Schrodinger equation,

$$-\frac{1}{2}\frac{\partial^2}{\partial \boldsymbol{\pi}_x^2}\Psi[\boldsymbol{\pi}_x, z] + \frac{1}{2}\Omega_z^2 \boldsymbol{\pi}_x^2 \Psi[\boldsymbol{\pi}_x, z] = -\frac{\partial \Psi[\boldsymbol{\pi}_x, z]}{\partial z}. \tag{D3}$$

We introduce a new variable, $\xi = \frac{\boldsymbol{\pi}_x}{A_z}$ and write the time-dependent solution as

$$\Psi[\boldsymbol{\pi}_x, z] = C_z \exp\left[-A_z^2 K_z \xi^2\right] \tilde{\Psi}[\xi, \theta], \tag{D4}$$

where $K_z$, $C_z$ and $\theta_z$ are function of $z$ which are related to $A_z$ through $K_z = \frac{\dot{A}_z}{2A_z}$, $\dot{C}_z = -C_z K_z$ and $\dot{\theta}(z) = 1/A_z^2$ with $\dot{A}_z = dA_z/dz$. $\tilde{\Psi}[\xi, \theta]$ satisfies

$$\frac{1}{2}A_z^3\left[\ddot{A}_z - A_z\Omega_z^2\right]\xi^2\tilde{\Psi}[\xi, \theta] - \frac{\delta\tilde{\Psi}[\xi, \theta]}{\delta\theta} + \frac{1}{2}\frac{\delta^2}{\delta\xi^2}\tilde{\Psi}[\xi, \theta] = 0 \tag{D5}$$

We choose $A_z$ so that $\ddot{A}_z - A_z\Omega_z^2 = 0$, and $\tilde{\Psi}$ satisfies

$$-\frac{\delta\tilde{\Psi}[\xi, \theta]}{\delta\theta} + \frac{1}{2}\frac{\delta^2}{\delta\xi^2}\tilde{\Psi}[\xi, \theta] = 0. \tag{D6}$$

This is the Schrodinger equation of a particle in the free space, which has a one-parameter family of solutions,

$$\tilde{\Psi}_\ell[\xi, \theta] = \exp\left[-\frac{\ell^2}{2}\theta(z)\right]\exp\left[-i\ell\xi(z)\right], \tag{D7}$$

where $\ell$ is a real parameter that corresponds to the momentum conjugae to $\xi$ in the particle analogy. The general solution is given by a linear superposition of $\Psi_\ell$ as

$$\Psi[\boldsymbol{\pi}_x, z] = \frac{1}{\sqrt{A_z}}\exp\left(-\frac{\dot{A}_z}{2A_z}\boldsymbol{\pi}_x^2\right)\int_{-\infty}^{\infty}d\ell\left\{\phi[\ell]\exp\left[-\frac{\ell^2}{2}\int_0^z\frac{dz'}{A_{z'}^2}\right]\exp\left[-i\frac{\ell\boldsymbol{\pi}_x}{A_z}\right]\right\}, \tag{D8}$$

where $\phi[\ell]$ is the weight for the mode labelled by $\ell$, and $\theta(z) = \int_0^z \frac{dz'}{A_{z'}^2}$ is used. At $z = 0$, the initial conditions $\dot{A}_0 = 0$ and $A_0 = 1$ lead to

$$\Psi[\boldsymbol{\pi}_x, 0] = \int d\ell \Big\{ \phi[\ell] \exp(-i\ell\boldsymbol{\pi}_x) \Big\}. \tag{D9}$$

Thus, $\phi[\ell]$ is the Fourier transformation of $\Psi[\boldsymbol{\pi}_x, 0]$,

$$\phi[\ell] = \frac{1}{2\pi} \int \Psi[\boldsymbol{\pi}_x', 0] \exp(i\ell\boldsymbol{\pi}_x') d\boldsymbol{\pi}_x' \tag{D10}$$

The $n$-th eigenstate of the $H$ at $z = 0$ has the Fourier components given by

$$\begin{aligned}
\phi_n[\ell] &= \frac{1}{2\pi\sqrt{2^n n!}} \Big(\frac{\Omega_0}{\pi}\Big)^{1/4} \int H_n\Big[\sqrt{\Omega_0}\boldsymbol{\pi}_x\Big] \exp\Big(-\frac{1}{2}\Omega_0\boldsymbol{\pi}_x^2\Big) \exp[i\ell\boldsymbol{\pi}_x] \, d\boldsymbol{\pi}_x \\
&= \frac{\sqrt{2}}{2\pi\sqrt{2^n}} \Big(\frac{\pi}{\Omega_0}\Big)^{1/4} H_n\Big(-i\sqrt{\Omega_0}\frac{\delta}{\delta\ell}\Big) \exp\Big[-\frac{\ell^2}{2\Omega_0}\Big].
\end{aligned} \tag{D11}$$

Because the Hermite polynomial is complete, $\phi_n[\ell]$ can be decomposed as a linear superposition of

$$\tilde{\phi}_{n'}[\ell] = \frac{\sqrt{2}}{2\pi\sqrt{2^{n'} n'!}} \Big(\frac{\pi}{\Omega_0}\Big)^{1/4} H_{n'}\Big[i\frac{\ell}{\sqrt{\Omega_0}}\Big] \exp\Big[-\frac{\ell^2}{2\Omega_0}\Big]. \tag{D12}$$

Inserting Eq. (D12) into Eq. (D8), we obtain a solution,

$$\Psi_{n'}[\boldsymbol{\pi}_x, z] = \frac{\sqrt{4\pi}}{2\pi\sqrt{2^{n'} n'!}} \Big(\frac{\pi}{\Omega_0}\Big)^{1/4} \sqrt{\frac{\omega_z}{A_z}} \exp\Big(-\frac{\dot{A}_z}{2A_z}\boldsymbol{\pi}_x^2\Big) H_{n'}\Big[-\frac{A_z}{\sqrt{\Omega_0}}\frac{\delta}{\delta\boldsymbol{\pi}_x}\Big] \exp\Big[-\frac{\omega_z}{2A_z^2}\boldsymbol{\pi}_x^2\Big], \tag{D13}$$

where we use $\omega_z = \Big[\int_0^z \frac{dz'}{A_{z'}^2} + \frac{1}{\Omega_0}\Big]^{-1}$ as in the main context. First three states are given by

$$\Psi_0[\boldsymbol{\pi}_x, z] = \frac{1}{\sqrt{\pi}} \Big(\frac{\pi}{\Omega_0}\Big)^{1/4} \sqrt{\frac{\omega_z}{A_z}} \exp\Big[-\frac{1}{2}\Big(\frac{\dot{A}_z}{A_z} + \frac{\omega_z}{A_z^2}\Big)\boldsymbol{\pi}_x^2\Big],$$

$$\Psi_1[\boldsymbol{\pi}_x, z] = \frac{1}{\sqrt{2\pi}} \Big(\frac{\pi}{\Omega_0}\Big)^{1/4} \sqrt{\frac{\omega_z}{A_z}} \exp\Big[-\frac{1}{2}\Big(\frac{\dot{A}_z}{A_z} + \frac{\omega_z}{A_z^2}\Big)\boldsymbol{\pi}_x^2\Big] \Big[2\frac{\omega_z}{A_z\sqrt{\Omega_0}}\boldsymbol{\pi}_x\Big],$$

$$\Psi_2[\boldsymbol{\pi}_x, z] = \frac{1}{2\sqrt{2\pi}} \Big(\frac{\pi}{\Omega_0}\Big)^{1/4} \sqrt{\frac{\omega_z}{A_z}} \exp\Big[-\frac{1}{2}\Big(\frac{\dot{A}_z}{A_z} + \frac{\omega_z}{A_z^2}\Big)\boldsymbol{\pi}_x^2\Big] \Big[4\Big(\frac{\omega_z}{A_z\sqrt{\Omega_0}}\boldsymbol{\pi}_x\Big)^2 - 2 - \frac{4\omega_z}{\Omega_0}\Big]. \tag{D14}$$

Now we consider the non-Hermitian RG Hamiltonian which is of our interest :

$$H' = \frac{1}{2}(\boldsymbol{x} + i\gamma\boldsymbol{\pi}_x)^2 + \frac{1}{2}\Omega_z^2\boldsymbol{\pi}_x^2. \tag{D15}$$

This is related to Eq. (D1) through the similarity transformation, $H' = e^{-\frac{\gamma}{2}\boldsymbol{\pi}_x^2} H e^{\frac{\gamma}{2}\boldsymbol{\pi}_x^2}$. Accordingly, its solution is related to Eq. (D13) through $\Psi'_n = e^{-\frac{\gamma}{2}\boldsymbol{\pi}_x^2}\Psi_n$,

$$\Psi'_n [\boldsymbol{\pi}_x, z] = \frac{\sqrt{4\pi}}{2\pi\sqrt{2^n n!}} \left(\frac{\pi}{\Omega_0}\right)^{1/4} \sqrt{\frac{\omega_z}{A_z}} \exp\left[-\frac{1}{2}\left(\gamma + \frac{\dot{A}_z}{A_z}\right)\boldsymbol{\pi}_x^2\right] H_n\left[-\frac{A_z}{\sqrt{\Omega_0}}\frac{\delta}{\delta\boldsymbol{\pi}_x}\right]\exp\left[-\frac{\omega_z}{2A_z^2}\boldsymbol{\pi}_x^2\right],$$
(D16)

where we use $\omega_z = \left[\int_0^z \frac{dz'}{A_{z'}^2} + \frac{1}{\Omega_0}\right]^{-1}$ as in the main context.

### Appendix E: Computation of $A_{s,z}$, $\omega_{s,z}$, $\Lambda_{s,z}$ and $\Delta_{s,z}$

In this appendix, we provide the expressions for $A_{s,z}$, $\omega_{s,z}$, $\Lambda_{s,z}$ and $\Delta_{s,z}$ that appear in the solution for the RG Hamiltonian in Sec. V B. Since the expressions of $A_{s,z}$, $\omega_{s,z}$, $\Lambda_{s,z}$ and $\Delta_{s,z}$ are the same for $s = (R; K)$ and $s = (I; K)$, we will just denote them as $A_{K,z}$, $\omega_{K,z}$ $\Lambda_{K,z}$ and $\Delta_{K,z}$ in this appendix. For $b = 1/2$, we express the solution for the Schrodinger equation Eq. (168) in terms of $\alpha = -2g$, $\zeta = a + \frac{D}{2}$ and $\sigma = \zeta^2 - 2w$.

#### 1. $A_{s,z}$

We start with $A_{s,z}$ that satisfies $\ddot{A}_{s,z} - A_{s,z}\Omega_{s,z}^2 = 0$ with the initial conditions $\dot{A}_{0,0} = 0$ and $A_{0,0} = 1$. For $K = 0$, $A_{0,z} = \cosh(\sqrt{\sigma}z)$ is the solution. For general $K$, $A_{K,z}$ is given by

$$A_{K,z} = \frac{\pi\sqrt{\alpha}}{4\sin(\pi\sqrt{\sigma})}|K|\left\{I_{-\sqrt{\sigma}}\left[\sqrt{\alpha}|K|e^z\right]\left(I_{-1+\sqrt{\sigma}}\left[\sqrt{\alpha}|K|\right] + I_{1+\sqrt{\sigma}}\left[\sqrt{\alpha}|K|\right]\right)\right.$$
$$\left. - I_{\sqrt{\sigma}}\left[\sqrt{\alpha}|K|e^z\right]\left(I_{-1-\sqrt{\sigma}}\left[\sqrt{\alpha}|K|\right] + I_{1-\sqrt{\sigma}}\left[\sqrt{\alpha}|K|\right]\right)\right\}.$$
(E1)

In the large $z$ limit with fixed $k \equiv Ke^z$, $A_{K,z}$ can be written as

$$\tilde{A}_{k,z} \equiv \lim_{z\to\infty} A_{K,z}\Big|_k = \mathbb{A}_k(\alpha, \sigma)e^{\sqrt{\sigma}z},$$
(E2)

where

$$\mathbb{A}_k(\alpha, \sigma) \approx -\frac{2^{-1+\sqrt{\sigma}}\pi\alpha^{-\frac{1}{2}\sqrt{\sigma}}}{\sin(\pi\sqrt{\sigma})}\frac{I_{\sqrt{\sigma}}\left[\sqrt{\alpha}|k|\right]}{\Gamma(-\sqrt{\sigma})}|k|^{-\sqrt{\sigma}}.$$
(E3)

This has the following limiting behaviour in $k = Ke^z$.

*a.* $|K|e^z \ll 1$ : For $\sqrt{\alpha}|K|e^z \ll \sqrt{1 - \sqrt{\sigma}} < \sqrt{1 + \sqrt{\sigma}}$, one can approximate $A_{K,z}$ to

$$A_{K,z} \sim \mathcal{A}_{K,1} e^{-\sqrt{\sigma}z} + \mathcal{A}_{K,2} e^{\sqrt{\sigma}z}, \tag{E4}$$

where

$$\mathcal{A}_{K,1} = \frac{\pi \sqrt{\alpha}^{1-\sqrt{\sigma}}}{2^{2-\sqrt{\sigma}} \sin(\pi\sqrt{\sigma})} \frac{1}{\Gamma(1 - \sqrt{\sigma})} |K|^{1-\sqrt{\sigma}} \left( I_{-1+\sqrt{\sigma}} \left[ \sqrt{\alpha}|K| \right] + I_{1+\sqrt{\sigma}} \left[ \sqrt{\alpha}|K| \right] \right),$$
$$\mathcal{A}_{K,2} = -\frac{\pi \sqrt{\alpha}^{1+\sqrt{\sigma}}}{2^{2+\sqrt{\sigma}} \sin(\pi\sqrt{\sigma})} \frac{1}{\Gamma(1 + \sqrt{\sigma})} |K|^{1+\sqrt{\sigma}} \left( I_{-1-\sqrt{\sigma}} \left[ \sqrt{\alpha}|K| \right] + I_{1-\sqrt{\sigma}} \left[ \sqrt{\alpha}|K| \right] \right). \tag{E5}$$

In the large $z$ limit with fixed $k$, Eq. (E5) becomes

$$\tilde{\mathcal{A}}_{k,1} \equiv \lim_{z\to\infty} \mathcal{A}_{K,1} \Big|_k = \frac{1}{2} + \frac{\alpha k^2 e^{-2z}}{8} \frac{1}{\sqrt{\sigma}(1 + \sqrt{\sigma})} + O\left( (ke^{-z})^4 \right),$$
$$\tilde{\mathcal{A}}_{k,2} \equiv \lim_{z\to\infty} \mathcal{A}_{K,2} \Big|_k = \frac{1}{2} + \frac{\alpha k^2 e^{-2z}}{8} \frac{1}{\sqrt{\sigma}(\sqrt{\sigma} - 1)} + O\left( (ke^{-z})^4 \right). \tag{E6}$$

*b.* $|K|e^z \gg 1$ : For $|K|e^z \gg 1$, $A_{K,z}$ can be expanded in powers of $1/|K|e^z$,

$$A_{K,z} = \mathcal{A}_{K,3} \frac{e^{\sqrt{\alpha}|K|e^z}}{e^{z/2}} \left[ 1 - \frac{1}{8} \frac{(4\sigma - 1)}{\sqrt{\alpha}|K|e^z} + O\left( \frac{1}{(Ke^z)^2} \right) \right], \tag{E7}$$

where

$$\mathcal{A}_{K,3} = \frac{\pi^{1/2} \alpha^{1/4} g_K}{4\sqrt{2} \sin(\pi\sqrt{\sigma})} |K|^{1/2} \tag{E8}$$

and

$$g_K = I_{-1+\sqrt{\sigma}} \left[ \sqrt{\alpha}|K| \right] + I_{1+\sqrt{\sigma}} \left[ \sqrt{\alpha}|K| \right] - I_{-1-\sqrt{\sigma}} \left[ \sqrt{\alpha}|K| \right] - I_{1-\sqrt{\sigma}} \left[ \sqrt{\alpha}|K| \right]. \tag{E9}$$

In the large $z$ limit with fixed $k$, we obtain

$$\tilde{\mathcal{A}}_{k,3} \equiv \lim_{z\to\infty} \mathcal{A}_{K,3} \Big|_k = V_3 k^{-\frac{1}{2} - \sqrt{\sigma}} e^{(\frac{1}{2} + \sqrt{\sigma})z} + O\left( (ke^{-z})^{\frac{3}{2} - \sqrt{\sigma}} \right), \tag{E10}$$

where

$$V_3 = -\frac{\pi^{1/2} \alpha^{-1/4 - \frac{\sqrt{\sigma}}{2}}}{2^{-\sqrt{\sigma} + \frac{3}{2}} \sin(\pi\sqrt{\sigma})} \frac{1}{\Gamma(-\sqrt{\sigma})}. \tag{E11}$$

**2.** $\omega_{s,z}$

Here, we show that

$$\tilde{\omega}_{k,z} \equiv \lim_{z \to \infty} \omega_{K,z}\Big|_k \approx \frac{1}{2}\sqrt{\sigma}\left[1 - \mathbb{W}_k e^{-2\sqrt{\sigma}z}\right]^{-1}, \tag{E12}$$

where

$$\mathbb{W}_k \approx \begin{cases} 1 & k \ll 1 \\ k^{2\sqrt{\sigma}}\left[1 - \frac{\sqrt{\sigma}}{4V_3^2\sqrt{\alpha}}\left(e^{-2\sqrt{\alpha}} - e^{-2\sqrt{\alpha}k}\right)\right] & k \gg 1 \end{cases} \tag{E13}$$

in the large $z$ limit with fixed $k = Ke^z$.

    *a.* $|K|e^z \ll 1$ :    Using Eq. (E4), we obtain

$$\omega_{K,z} = \left[\frac{1}{\Omega_{K,0}} + \int_0^z \frac{dz'}{A_{K,z'}^2}\right]^{-1} \approx \left[\frac{1}{\Omega_{K,0}} + \frac{e^{2\sqrt{\sigma}z} - 1}{2\sqrt{\sigma}(\mathcal{A}_{K,1} + \mathcal{A}_{K,2})(\mathcal{A}_{K,1} + \mathcal{A}_{K,2}e^{2\sqrt{\sigma}z})}\right]^{-1}. \tag{E14}$$

According to Eq. (E6), we find

$$\tilde{\omega}_{k,z} \equiv \lim_{z \to \infty} \omega_{K,z}\Big|_k \approx \frac{1}{2}\sqrt{\sigma}\left[1 - e^{-2\sqrt{\sigma}z}\right]^{-1}, \tag{E15}$$

where we used $\lim_{z \to \infty} \Omega_{K,0}|_k = \sqrt{\sigma}$

    *b.* $|K|e^z \gg 1$ :    According to Eq. (E7), we have

$$\omega_{K,z} \approx \left[\frac{1}{\Omega_{K,0}} + \frac{K^{-2\sqrt{\sigma}} - 1}{2\sqrt{\sigma}(\mathcal{A}_{K,1} + \mathcal{A}_{K,2})(\mathcal{A}_{K,1} + \mathcal{A}_{K,2}K^{-2\sqrt{\sigma}})} + \frac{1}{\mathcal{A}_{K,3}^2}\left(-\frac{e^{-2\sqrt{\alpha}|K|e^z}}{2\sqrt{\alpha}|K|} + \frac{e^{-2\sqrt{\alpha}}}{2\sqrt{\alpha}|K|}\right)\right]^{-1}. \tag{E16}$$

In the large $z$ limit, based on Eq. (E10) we find

$$\tilde{\omega}_{k,z} \equiv \lim_{z \to \infty} \omega_{K,z}\Big|_k \approx \frac{1}{2}\sqrt{\sigma}\left[1 - k^{2\sqrt{\sigma}}\left(1 - \frac{\sqrt{\sigma}}{4V_3^2\sqrt{\alpha}}\left(e^{-2\sqrt{\alpha}} - e^{-2\sqrt{\alpha}k}\right)\right)e^{-2\sqrt{\sigma}z}\right]^{-1} \tag{E17}$$

for $k \gg 1$.

**3.** $\Lambda_{s,z}$

In this section, we compute $\Lambda_{s,z}$ defined by

$$\frac{1}{\Lambda_{s,z}} = \left[\zeta + \frac{\dot{A}_{s,z}}{A_{s,z}} + \frac{\omega_{s,z}}{A_{s,z}^2}\right], \tag{E18}$$

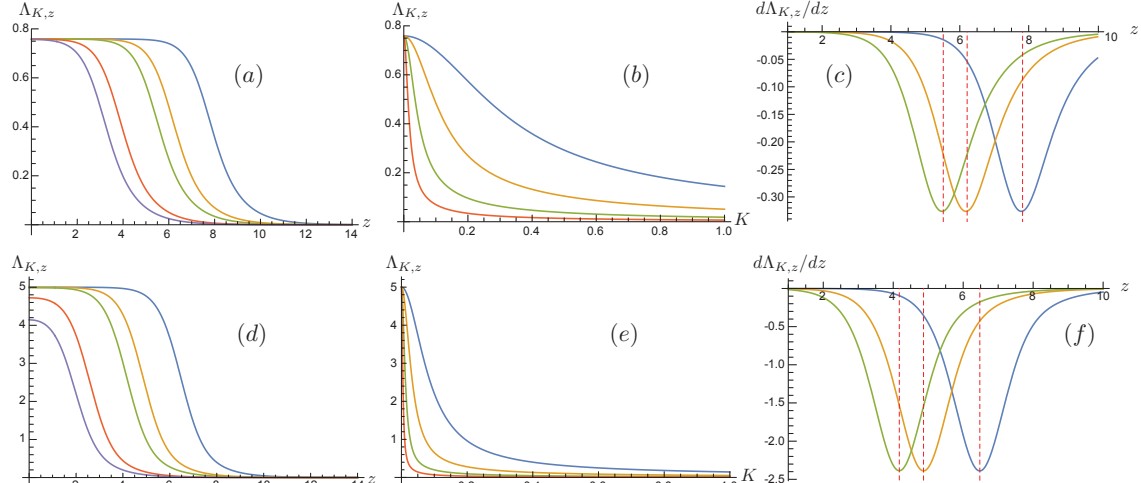

FIG. 8. $\Lambda_{K,z}$ plotted as a functions of $K$ or $z$ with $\alpha = 1$ in $D = 1$. For (a), (b), (c), we choose $\zeta = -0.1$ and $\sigma = 2.01$. For (d), (e), (f), we choose $\zeta = 0.1$ and $\sigma = 0.01$. (a) $\Lambda_{K,z}$ vs $z$ at $K = 0.001$ (blue), 0.005 (orange), 0.01 (green), 0.05 (red) and 0.1 (purple). (b) $\Lambda_{K,z}$ vs $K$ at $z = 2$ (blue), 3 (orange), 4 (green), 5 (red). (c) $\frac{d}{dz}\Lambda_{K,z}$ vs $z$ for $K = 0.001$, $K = 0.005$ and $K = 0.01$. For each value of $K$, the minimum occurs at $z = 7.82184$, $z = 6.2124$ and $z = 5.51925$, respectively. At all minima, $Ke^z$ takes the same value, 2.49448. (d) $\Lambda_{K,z}$ vs $z$ at $K = 0.001$ (blue), 0.005 (orange), 0.01 (green), 0.05 (red) and 0.1 (purple). (e) $\Lambda_{K,z}$ vs $K$ at $z = 2$ (blue), 3 (orange), 4 (green), 5 (red). (f) $\frac{d}{dz}\Lambda_{K,z}$ vs $z$ for $K = 0.001$, $K = 0.005$ and $K = 0.01$. For each value of $K$, the minimum is located at $z = 6.47645$, $z = 4.86718$ and $z = 4.17465$. At all minima, $Ke^z$ takes the same value, 0.649.

where $\omega_{s,z} = \left[\int_0^z \frac{dz'}{A_{s,z'}^2} + \frac{1}{\Omega_{s,0}}\right]^{-1}$. In Fig. 8, we show the profile of $\Lambda_{K,z}$ for two sets of parameters. $\Lambda_{K,z}$ smoothly interpolates the two limiting behaviours of the $Ke^z \gg 1$ and $Ke^z \ll 1$ limits. If one takes the large $z$ limit with fixed $k = Ke^z$, $\Lambda_{K,z}$ approaches a universal function as is shown in Fig. 9. Now, let us find the analytic expression for

$$\tilde{\Lambda}_k = \lim_{z \to \infty} \Lambda_{K,z}\Big|_k, \tag{E19}$$

where the limit is taken with $k = Ke^z$ fixed.

In the large $z$ limit, as we shown in Eq. (E2) and Eq. (E12), $\tilde{\omega}_{k,z}$ approaches $\frac{1}{2}\sqrt{\sigma}$, and $\tilde{\omega}_{k,z} \ll \tilde{A}_{k,z}$. So the dominant contribution to $\Lambda_{K,z}$ in Eq. (E18) is from $\left[\frac{\dot{A}_{K,z}}{A_{K,z}} + \zeta\right]^{-1}$. Therefore, $\Lambda_{K,z}$ approaches a universal form as a function of $k = |K|e^z$,

$$\tilde{\Lambda}_k = [\mathbb{G}_k(\alpha, \sigma) + \zeta]^{-1}, \tag{E20}$$

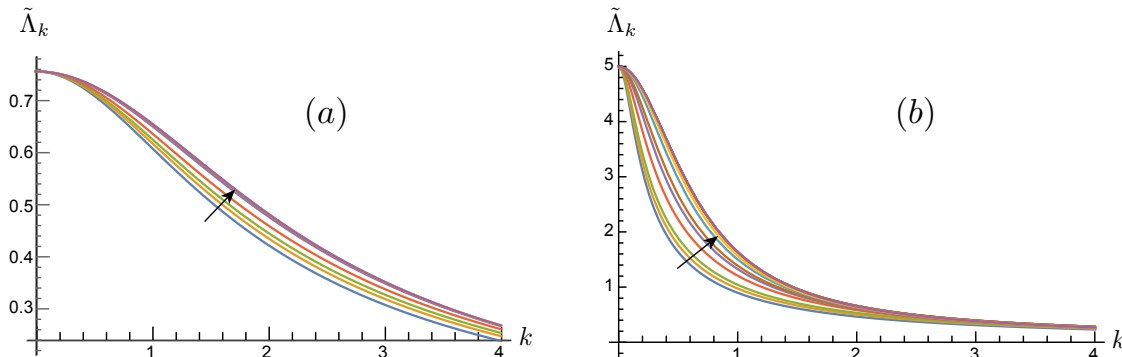

FIG. 9. $\Lambda_{K,z} = \tilde{\Lambda}_k$ vs $k$ for various values of $z$ between 0 and 20 in $D = 1$. The arrows point towards the direction of increasing $z$. (a) $\alpha = 1$, $\zeta = -0.1$ and $\sigma = 2.01$ (b) $\alpha = 1$, $\zeta = 0.1$ and $\sigma = 0.01$. The curves converge to a universal one in the large $z$ limit.

as the large $z$ limit is taken with fixed $k$, where

$$\mathbb{G}_k(\alpha, \sigma) = \frac{\dot{A}_{K,z}}{A_{K,z}} = \frac{1}{2}\sqrt{\alpha}\frac{|k|}{I_\sigma\left[\sqrt{\alpha}|k|\right]}(I_{-1+\sqrt{\sigma}}\left[\sqrt{\alpha}|k|\right] + I_{1+\sqrt{\sigma}}\left[\sqrt{\alpha}|k|\right]). \tag{E21}$$

$\mathbb{G}_k(\alpha, \sigma)$ becomes

$$\mathbb{G}_k(\alpha, \sigma) \approx \sqrt{\sigma} \tag{E22}$$

for $k \ll 1$, and

$$\mathbb{G}_k(\alpha, \sigma) \approx \sqrt{\alpha}|k| \tag{E23}$$

for $k \gg 1$. In order for the wavefunction to be normalizable, the width of the Gaussian wavefunction in Eq. (172) should be finite. This requires $\Lambda_{K,z} > 0$ for all $K$ and $z$. This, in turn, implies that $\Omega_{K,0} > -\zeta$ for all $K$, equivalently $\sqrt{\sigma} > -\zeta$.

### 4. $\Delta_{s,z}$

According to Eq. (E2) and Eq. (E12), we have expressed $\Omega_{K,0}$, $A_{K,z}$ and $\omega_{K,z}$ in terms of $k$ and $z$. In the large $z$ limit, we have

$$e^{-\Delta_{K,z}} = \frac{\omega_{K,z}}{A_{K,z}\sqrt{\Omega_{K,0}}} \approx \frac{1}{2}\frac{\sigma^{1/4}}{\mathbb{A}_{Ke^z}(\alpha, \sigma)}e^{-\sqrt{\sigma}z}. \tag{E24}$$

This analytical expression is consistent with the numerical plot of $\Delta_{K,z}$ shown in Fig. 10. As a function of $k$, $\Delta_{K,z}$ at different $z$ behave in the same way except for a vertical shift, as is shown in Fig. 11. This agrees with our analytical expression, $\Delta_{K,z} = -\log 2 - \frac{1}{4}\log \sigma + \log \mathbb{A}_k + \sqrt{\sigma}z$. Under the RG transformation from length scale $z$ to $z + dz$ followed by the rescaling of $K$ to $Ke^{-z}$, $\Delta_{K,z}$ transforms to $\Delta_{K,z} + \sqrt{\sigma}dz$.

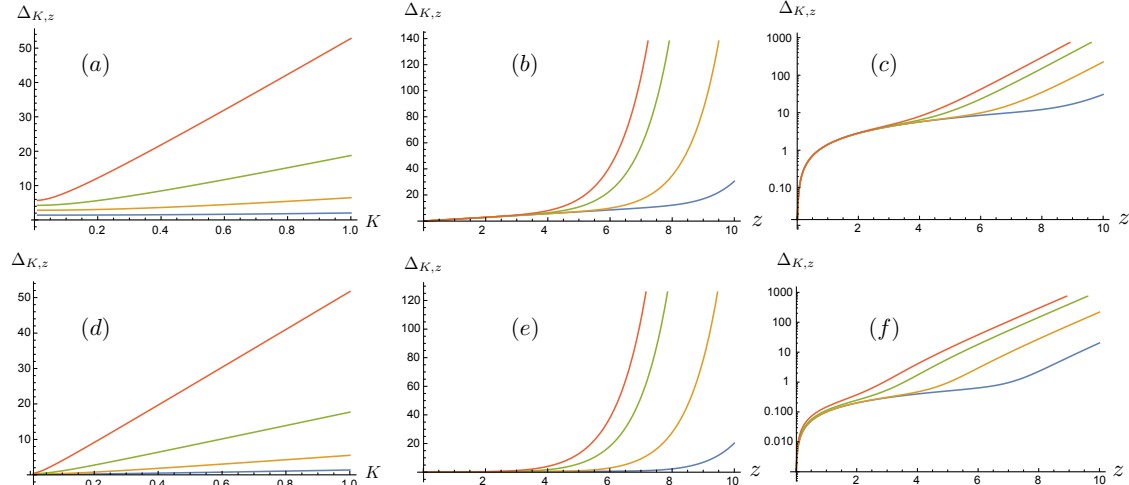

FIG. 10. (a) and (d): $\Delta_{K,z}$ plotted as a function of $K$ at $z = 1$ (blue), $z = 2$ (orange), $z = 3$ (green) and $z = 4$ (red) for (a) $\zeta = 0.1$, $\sigma = 0.01$, and (d) $\zeta = -0.1$, $\sigma = 2.01$. (b) and (e): $\Delta_{K,z}$ plotted as a function of $z$ at $K = 0.001$ (blue), $K = 0.01$ (orange), $K = 0.05$ (green) and $K = 0.1$ (red) for (b) $\zeta = 0.1$, $\sigma = 0.01$, and (e) $\zeta = -0.1$, $\sigma = 2.01$. (c) and (f) are (b) and (e) shown in the logarithmic scale. In all plots, we set $D = 1$ and $\alpha = 1$.

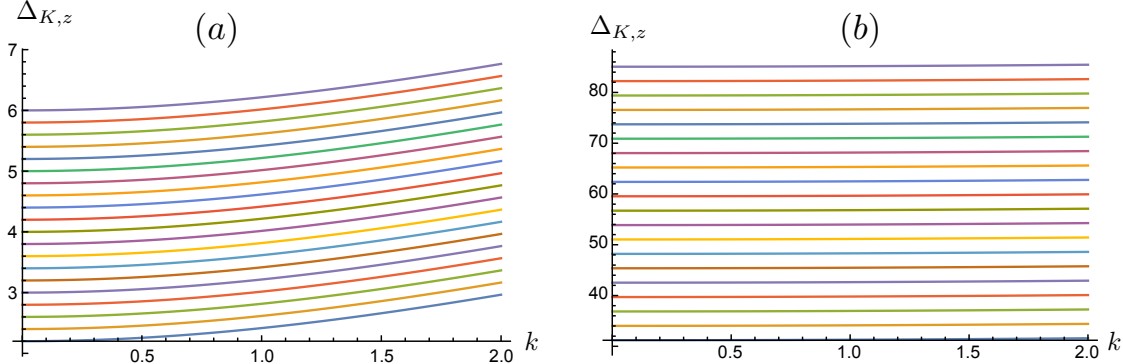

FIG. 11. $\Delta_{K,z}$ as a function of $k = Ke^z$ at different values of $z$ ranging from $z = 22$ (bottom) to $z = 60$ (top) for (a) $\zeta = 0.1$, $\sigma = 0.01$, and (b) $\zeta = -0.1$, $\sigma = 2.01$. $D = 1$ and $\alpha = 1$ are used for both plots.

## Appendix F: Numerical calculation of $J^*_{2,x-x'}$

It is hard to obtain a full expression for $J^*_{2,x-x'}$ in a closed form. In this appendix, we compute it numerically for $D = 1$. This requires UV and IR regularizations. Consider a one-dimensional lattice with $N$ sites. Before doing the scale transformation in Eq. (156), if

the lattice spacing is $a$, a function $f_{X=xe^z,z}$ at $z = 0$ can be expressed as

$$f_{X,0} = \frac{1}{N} \sum_{m=1}^{N} f_{\frac{2\pi}{Na}m} e^{i\frac{2\pi}{Na}mX} = a \int_0^{\Lambda_0 \sim \frac{2\pi}{a}} \frac{dK}{2\pi} f_{K,0} e^{iKX}. \tag{F1}$$

For $z \neq 0$, the lattice spacing increases to $ae^z$ and the number of sites decreases to $N(z) = Ne^{-z}$. Then Eq. (F1) becomes

$$f_{xe^z,z} = \frac{1}{N(z)} \sum_{m=1}^{N(z)} f_{\frac{2\pi}{N(z)ae^z}m} e^{i\frac{2\pi}{N(z)a}mx} = a \int_0^{\Lambda_0 e^{-z}} \frac{e^z dK}{2\pi} f_{K,z} e^{iKX} = a \int_0^{\Lambda_0} \frac{dk}{2\pi} f_{ke^{-z},z} e^{ikx}, \tag{F2}$$

where $k = Ke^z$ and $x = e^{-z}X$. If $f_{ke^{-z},z} = \tilde{f}_k$ is independent of $z$ for a fixed $k$, $\tilde{f}_x = f_{xe^z,z}$ is scale invariant, i.e. $z$-independent. Since $\Lambda_{K,z} = \tilde{\Lambda}_k$, the profile of $J_{2,x-x'}$ would be invariant under RG transformation.

Now, let us numerically compute $J^*_{2,x-x'}$, which is expressed as

$$J^*_{2,x-x'} = \frac{1}{N(z)} \sum_{m=1}^{N(z)} \tilde{\Lambda}_{\frac{2\pi m}{N(z)}} \cos\left[\frac{2\pi m}{N(z)}(x - x')\right] \tag{F3}$$

with $a = 1$. The profile is shown in Fig. 5. For large enough system size $N$, the coupling in the real space reaches a $z$-independent profile at large $z$ provided $Ne^{-z} \gg 1$. This profile is universal because it does not depend on $\Omega_{K,0}$ at UV. We note that there are regions of negative coupling at large $|x - x'|$. We attribute this phenomenon as a finite size effect. In Fig. 12, as the system size $N$ increases, the coupling becomes more positive.

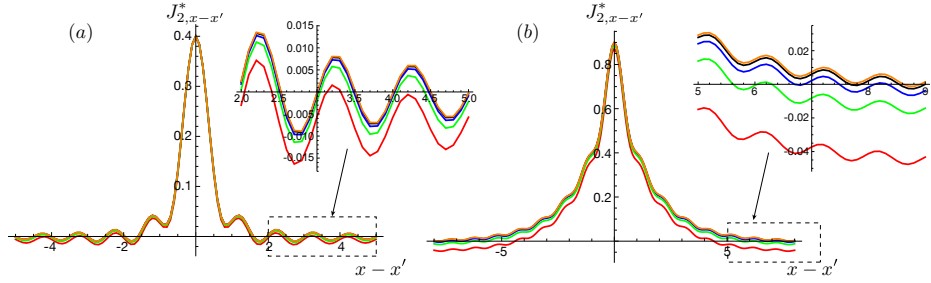

FIG. 12. (a) $J^*_{2,x-x'}$ in $D = 1$ plotted as a function of $x - x'$ for $\sigma = 2.01$ and $\zeta = -0.1$ for $z = 26$ at $N = e^{30}$ (red), $N = e^{31}$ (green), $N = e^{32}$ (blue), $N = e^{33}$ (black), $N = e^{34}$ (orange). (b) The profile at $\sigma = 0.01$ and $\zeta = 0.1$. Curves with a same color are the ones with a same $N$. $\alpha$ is set to 1.

**Appendix G: Wavefunction with one excited mode in the D-dimensional example**

Suppose that the mode $s$ is in its first excited state, where $s$ can be either $P = 0$, $(R; P)$ or $(I; P)$. The wavefunction for mode $s$ is given by

$$\Psi_{1,s}\left[\tilde{t}_s, z\right] = \pi^{-1/4} e^{-\frac{1}{2}\Delta_{s,z}} \left(\sqrt{2} e^{-\Delta_{s,z}} \tilde{t}_s\right) \exp\left[-\frac{1}{2\Lambda_{s,z}} \tilde{t}_s^2\right], \tag{G1}$$

where $H_1(x) = 2x$ is used. The excited state corresponds to the following state in the rescaled variables in the large $z$ limit,

$$|\Psi_{1,0}(z)\rangle = \mathcal{N}(z) \int \mathcal{D}\phi \; e^{-S_0}\left(i\sqrt{2}\frac{\sigma^{1/4}}{2\mathbb{A}_0} e^{-\sqrt{\sigma}z}\tilde{\Lambda}_0 \mathcal{O}_0\right) e^{-S^*}|\phi\rangle,$$

$$|\Psi_{1,(R;P)}(z)\rangle = \mathcal{N}(z) \int \mathcal{D}\phi \; \left(i\frac{\sigma^{1/4}}{2\mathbb{A}_{Pe^z}} e^{-\sqrt{\sigma}z}\tilde{\Lambda}_{Pe^z}(\mathcal{O}_{Pe^z} + \mathcal{O}_{-Pe^z})\right) e^{-S^*}|\phi\rangle, \tag{G2}$$

$$|\Psi_{1,(I;P)}(z)\rangle = \mathcal{N}(z) \int \mathcal{D}\phi \; \left(\frac{\sigma^{1/4}}{2\mathbb{A}_{Pe^z}} e^{-\sqrt{\sigma}z}\tilde{\Lambda}_{Pe^z}(\mathcal{O}_{Pe^z} - \mathcal{O}_{-Pe^z})\right) e^{-S^*}|\phi\rangle,$$

where we used $e^{-\Delta_{K,z}} \approx \frac{\sigma^{1/4}}{2\mathbb{A}_{Ke^z}(\alpha,\sigma)} e^{-\sqrt{\sigma}z}$ and $k = e^z K$. $\mathcal{N}(z)$ is the $z$-dependent normalization of the ground state in Eq. (183). $P$ labels the initial momentum of the excited mode. It is scaled to be $Pe^z$ as $z$ increases. For $P \neq 0$, we can construct the excited state with a momentum $\pm P$ by making linear superpositions of $|\Psi_{1,(R;P)}(z)\rangle$ and $|\Psi_{1,(I;P)}(z)\rangle$ : $|\Psi_{1,\pm P}(z)\rangle = \frac{1}{\sqrt{2}}\left(|\Psi_{1,(R;P)}(z)\rangle \pm i|\Psi_{1,(I;P)}(z)\rangle\right)$. This leads to Eq. (186).

**Appendix H: Possible wavefunctions with two excited modes in the D-dimensional example**

In order to derive Eq. (192), we first list the wave functions for two excited modes as

$$\Psi_{2,0,0}\left[\tilde{t}_0, z\right] = \pi^{-1/4} e^{-\frac{1}{2}\Delta_{0,z}} \frac{1}{\sqrt{2}}\left[2(e^{-\Delta_{0,z}}\tilde{t}_0)^2 - 1 - \frac{2\omega_{0,z}}{\Omega_{0,0}}\right] \exp\left[-\frac{1}{2\Lambda_{0,z}}\tilde{t}_0^2\right] \tag{H1}$$

$$= \frac{1}{\sqrt{2}}\Psi_{1,0}\left[\tilde{t}_0, z\right] \times \Psi_{1,0}\left[\tilde{t}_0, z\right] - \frac{1}{\sqrt{2}}(1 + \frac{2\omega_{0,z}}{\Omega_{0,0}})\Psi_{0,0}\left[\tilde{t}_0, z\right],$$

$$\Psi_{2,(S;P),(S;P)}\left[\tilde{t}_{S;P}, z\right] = \pi^{-1/4} e^{-\frac{1}{2}\Delta_{P,z}} \frac{1}{\sqrt{2}}\left[2(e^{-\Delta_{P,z}}\tilde{t}_{S;P})^2 - 1 - \frac{2\omega_{P,z}}{\Omega_{P,0}}\right] \exp\left[-\frac{1}{2\Lambda_{P,z}}\tilde{t}_{S;P}^2\right]$$

$$= \frac{1}{\sqrt{2}}\Psi_{1,(S;P)}\left[\tilde{t}_{S;P}, z\right] \times \Psi_{1,(S;P)}\left[\tilde{t}_{S;P}, z\right] - \frac{1}{\sqrt{2}}(1 + \frac{2\omega_{P,z}}{\Omega_{P,0}})\Psi_{0,(S;P)}\left[\tilde{t}_{S;P}, z\right],$$

$$\Psi_{2,(S;P),(S';P')\neq(S;P)} = \Psi_{1,S;P}\left[\tilde{t}_{S;P}, z\right] \times \Psi_{1,S';P'}\left[\tilde{t}_{S';P'}, z\right].$$

Here $S$ and $S'$ can be $R$ or $I$. Using Eq. (E12) and Eq. (E24), we rewrite the excited states in terms of rescaled variables $p$ in the large $z$ limit as

$$
\begin{aligned}
|\bar{\Psi}_{2,0,0}(z)\rangle &= \frac{\mathcal{N}_2(z)}{\sqrt{2}} \int \mathcal{D}\phi \left( -2\frac{\sigma^{1/2}}{4\mathbb{A}_0^2} \left[ \tilde{\Lambda}_0^2 \mathcal{O}_0^2 \right] \right) e^{-S^*} |\phi\rangle \\
&\quad + \frac{1}{\sqrt{2}} \left( 2\frac{\sigma^{1/2}}{4\mathbb{A}_0^2} e^{-2\sqrt{\sigma}z} \tilde{\Lambda}_0 - 1 - \left[ 1 - \mathbb{W}_0 e^{-2\sqrt{\sigma}z} \right] \right) |\Psi_0(z)\rangle, \\
|\Psi_{2,(R;P),(R;P)}(z)\rangle &= \frac{\mathcal{N}_2(z)}{\sqrt{2}} \int \mathcal{D}\phi \left( -\frac{\sigma^{1/2}}{4\mathbb{A}_{Pe^z}^2} \tilde{\Lambda}_{Pe^z}^2 (\mathcal{O}_{Pe^z} + \mathcal{O}_{-Pe^z})^2 \right) e^{-S^*} |\phi\rangle \\
&\quad + \frac{1}{\sqrt{2}} \left( 2\frac{\sigma^{1/2}}{4\mathbb{A}_{Pe^z}^2} e^{-2\sqrt{\sigma}z} \tilde{\Lambda}_{Pe^z} - 1 - \frac{\sqrt{\sigma}}{\sqrt{\alpha P^2 + \sigma}} \left[ 1 - \mathbb{W}_{Pe^z} e^{-2\sqrt{\sigma}z} \right] \right) |\Psi_0(z)\rangle, \\
|\Psi_{2,(I;P),(I;P)}(z)\rangle &= \frac{\mathcal{N}_2(z)}{\sqrt{2}} \int \mathcal{D}\phi \left( \frac{\sigma^{1/2}}{4\mathbb{A}_{Pe^z}^2} \tilde{\Lambda}_{Pe^z}^2 (\mathcal{O}_{Pe^z} - \mathcal{O}_{-Pe^z})^2 \right) e^{-S^*} |\phi\rangle \\
&\quad + \frac{1}{\sqrt{2}} \left( 2\frac{\sigma^{1/2}}{4\mathbb{A}_{Pe^z}^2} e^{-2\sqrt{\sigma}z} \tilde{\Lambda}_{Pe^z} - 1 - \frac{\sqrt{\sigma}}{\sqrt{\alpha P^2 + \sigma}} \left[ 1 - \mathbb{W}_{Pe^z} e^{-2\sqrt{\sigma}z} \right] \right) |\Psi_0(z)\rangle, \\
|\Psi_{2,(R;P),(I;P)}(z)\rangle &= \mathcal{N}_2(z) \int \mathcal{D}\phi \left( i\frac{\sigma^{1/2}}{4\mathbb{A}_{Pe^z}^2} \tilde{\Lambda}_{Pe^z}^2 \left[ \mathcal{O}_{Pe^z}^2 - \mathcal{O}_{-Pe^z}^2 \right] \right) e^{-S^*} |\phi\rangle,
\end{aligned}
$$

$$
\tag{H2}
$$

$$
\begin{aligned}
|\Psi_{2,0,(R;P)}(z)\rangle &= \mathcal{N}_2(z) \int \mathcal{D}\phi \left( -\sqrt{2}\frac{\sigma^{1/2}}{4\mathbb{A}_{Pe^z}\mathbb{A}_0} \tilde{\Lambda}_0 \tilde{\Lambda}_{Pe^z} \mathcal{O}_0 \left[ \mathcal{O}_{-Pe^z} + \mathcal{O}_{Pe^z} \right] \right) e^{-S^*} |\phi\rangle, \\
|\Psi_{2,0,(I;P)}(z)\rangle &= \mathcal{N}_2(z) \int \mathcal{D}\phi \left( i\sqrt{2}\frac{\sigma^{1/2}}{4\mathbb{A}_{Pe^z}\mathbb{A}_0} \tilde{\Lambda}_0 \tilde{\Lambda}_{Pe^z} \mathcal{O}_0 \left[ \mathcal{O}_{Pe^z} - \mathcal{O}_{-Pe^z} \right] \right) e^{-S^*} |\phi\rangle, \\
|\Psi_{2,(R;P),(R;P')}(z)\rangle &= \mathcal{N}_2(z) \int \mathcal{D}\phi \left( \frac{-\sigma^{1/2}}{4\mathbb{A}_{Pe^z}\mathbb{A}_{P'e^z}} \tilde{\Lambda}_{P'e^z} \tilde{\Lambda}_{Pe^z} \left[ \mathcal{O}_{Pe^z} + \mathcal{O}_{-Pe^z} \right] \left[ \mathcal{O}_{P'e^z} + \mathcal{O}_{-P'e^z} \right] \right) e^{-S^*} |\phi\rangle, \\
|\Psi_{2,(R;P),(I;P')}(z)\rangle &= \mathcal{N}_2(z) \int \mathcal{D}\phi \left( \frac{i\sigma^{1/2}}{4\mathbb{A}_{Pe^z}\mathbb{A}_{P'e^z}} \tilde{\Lambda}_{P'e^z} \tilde{\Lambda}_{Pe^z} \left[ \mathcal{O}_{Pe^z} + \mathcal{O}_{-Pe^z} \right] \left[ \mathcal{O}_{P'e^z} - \mathcal{O}_{-P'e^z} \right] \right) e^{-S^*} |\phi\rangle, \\
|\Psi_{2,(I;P),(R;P')}(z)\rangle &= \mathcal{N}_2(z) \int \mathcal{D}\phi \left( \frac{i\sigma^{1/2}}{4\mathbb{A}_{Pe^z}\mathbb{A}_{P'e^z}} \tilde{\Lambda}_{P'e^z} \tilde{\Lambda}_{Pe^z} \left[ \mathcal{O}_{Pe^z} - \mathcal{O}_{-Pe^z} \right] \left[ \mathcal{O}_{P'e^z} + \mathcal{O}_{-P'e^z} \right] \right) e^{-S^*} |\phi\rangle, \\
|\Psi_{2,(I;P),(I;P')}(z)\rangle &= \mathcal{N}_2(z) \int \mathcal{D}\phi \left( \frac{\sigma^{1/2}}{4\mathbb{A}_{Pe^z}\mathbb{A}_{P'e^z}} \tilde{\Lambda}_{P'e^z} \tilde{\Lambda}_{Pe^z} \left[ \mathcal{O}_{Pe^z} - \mathcal{O}_{-Pe^z} \right] \left[ \mathcal{O}_{P'e^z} - \mathcal{O}_{-P'e^z} \right] \right) e^{-S^*} |\phi\rangle,
\end{aligned}
$$

$$
\tag{H3}
$$

where $\mathcal{N}_2(z) = \mathcal{N}(z)e^{-2\sqrt{\sigma}z}$. For $P \neq P'$, we can superpose the wavefunctions above to obtain

$$|\Psi_{2,P,P'}(z)\rangle = \frac{\sqrt{2}}{4}\Big(|\Psi_{2,(R;P),(R;P')}(z)\rangle + i|\Psi_{2,(R;P),(I;P')}(z)\rangle - |\Psi_{2,(I;P),(I;P')}(z)\rangle + i|\Psi_{2,(I;P),(R;P')}(z)\rangle\Big),$$

$$|\Psi_{2,P,-P'}(z)\rangle = \frac{\sqrt{2}}{4}\Big(|\Psi_{2,(R;P),(R;P')}(z)\rangle - i|\Psi_{2,(R;P),(I;P')}(z)\rangle + |\Psi_{2,(I;P),(I;P')}(z)\rangle + i|\Psi_{2,(I;P),(R;P')}(z)\rangle\Big),$$

$$|\Psi_{2,-P,P'}(z)\rangle = \frac{\sqrt{2}}{4}\Big(|\Psi_{2,(R;P),(R;P')}(z)\rangle + i|\Psi_{2,(R;P),(I;P')}(z)\rangle + |\Psi_{2,(I;P),(I;P')}(z)\rangle - i|\Psi_{2,(I;P),(R;P')}(z)\rangle\Big),$$

$$|\Psi_{2,-P,-P'}(z)\rangle = \frac{\sqrt{2}}{4}\Big(|\Psi_{2,(R;P),(R;P')}(z)\rangle - i|\Psi_{2,(R;P),(I;P')}(z)\rangle - |\Psi_{2,(I;P),(I;P')}(z)\rangle - i|\Psi_{2,(I;P),(R;P')}(z)\rangle\Big),$$

$$|\Psi_{2,0,P}(z)\rangle = \frac{1}{2}(|\Psi_{2,0,(R;P)}(z)\rangle + i|\Psi_{2,0,(I;P)}(z)\rangle),$$

$$|\Psi_{2,0,-P}(z)\rangle = \frac{1}{2}(|\Psi_{2,0,(R;P)}(z)\rangle - i|\Psi_{2,0,(I;P)}(z)\rangle).$$

$$\text{(H4)}$$

For non-zero $P$, we have

$$|\Psi_{2,P,P}(z)\rangle = \frac{1}{2}\Big(|\Psi_{2,(R;P),(R;P)}(z)\rangle - |\Psi_{2,(I;P),(I;P)}(z)\rangle + \sqrt{2}i|\Psi_{2,(R;P),(I;P)}(z)\rangle\Big),$$

$$|\Psi_{2,-P,-P}(z)\rangle = \frac{1}{2}\Big(|\Psi_{2,(R;P),(R;P)}(z)\rangle - |\Psi_{2,(I;P),(I;P)}(z)\rangle - \sqrt{2}i|\Psi_{2,(R;P),(I;P)}(z)\rangle\Big),$$

$$|\Psi_{2,P,-P}(z)\rangle = \frac{1}{2}\Big(|\Psi_{2,(R;P),(R;P)}(z)\rangle + |\Psi_{2,(I;P),(I;P)}(z)\rangle\Big) + \frac{1}{\sqrt{2}}\left[1 + \frac{\sqrt{\sigma}}{\sqrt{\alpha P^2 + \sigma}}\right]|\Psi_0(z)\rangle.$$

$$\text{(H5)}$$

Finally, together with

$$|\Psi_{2,0,0}(z)\rangle = |\bar{\Psi}_{2,0,0}(z)\rangle + \sqrt{2}|\Psi_0(z)\rangle, \tag{H6}$$

$|\Psi_{2,P,P'}(z)\rangle$ for any $P$ and $P'$ can be written in the general form given in Eq. (192).

### Appendix I: Other scaling operators

In this section, we consider general excited states. The wavefunction for $n$ excited modes is

$$|\Psi_{n,\{P\}}(z)\rangle = \left[(-i)^n\frac{\sqrt{2^n}}{\sqrt{n!}}\right]\mathcal{N}(z)\int \mathcal{D}\phi \left(\sigma^{n/4}\frac{\prod_i^n \tilde{\Lambda}_{P_ie^z}}{2^n \prod_i^n \mathbb{A}_{P_ie^z}}e^{-n\sqrt{\sigma}z}\left[\prod_i^n \mathcal{O}_{P_ie^z}\right] + \dots\right)e^{S^*}|\phi\rangle, \tag{I1}$$

where ... includes terms with less number of $\mathcal{O}_{Pe^z}$ operators. This wave function leads to state of the system as

$$
\begin{aligned}
|\Psi_{n,\{X\}}(z)\rangle &= \frac{1}{V^{n/2}} \sum_{\{P\}} e^{i\sum_i^n P_i X_i} |\Psi_{n,\{P\}}(z)\rangle \\
&= \left[ (-i)^n \frac{\sqrt{2^n}}{\sqrt{n!}} \right] \mathcal{N}(z) e^{-(n\sqrt{\sigma}+\frac{nD}{2})z} \int \mathcal{D}\phi \ \hat{\mathcal{A}}_n(\{x \equiv Xe^{-z}\}) e^{-S^*} |\phi\rangle + \dots,
\end{aligned}
$$
(I2)

where the scaling operator is defined as

$$
\hat{\mathcal{A}}_n(\{x\}) = \sum_{m=0}^{[n/2]} \int \left[ \prod_i^{n-2m} d^D y_i \right] J_{\{x-y\}}^{(n,n-2m)} \left[ \prod_i^{n-2m} \mathcal{O}_{y_i} \right]
$$
(I3)

with

$$
J_{\{x-y\}}^{(n,n-2m)} = \int \left[ \prod_i^{n-2m} \frac{d^D p_i}{(2\pi)^D} \right] J_{\{p\}}^{(n,n-2m)} e^{i\sum_i^{n-2m} p_i(x_i-y_i)} e^{i\sum_{i=n-2m+1}^n p_i x_i} \delta\left( \sum_{i=n-2m+1}^n p_i \right).
$$
(I4)

Here $J^{(n,n-2m)}$ represents the weight for $(n-2m)$-trace operators to the $n$-th scaling operator. For example, the contribution from the $n$-trace operator is given by

$$
J_{\{p\}}^{(n,n)} = \sigma^{n/4} \frac{\prod_i^n \tilde{\Lambda}_{p_i}}{2^n \prod_i^n \mathbb{A}_{p_i}}.
$$
(I5)

The local operator $\hat{\mathcal{A}}_n$ has scaling dimension $n\left(\sqrt{\sigma} + \frac{D}{2}\right)$.