# Peer review of "Constraints on beta functions in field theories"

_SciPost Physics_

## Round 3 · Referee Report · Anonymous (Referee 1) · 2021-5-11

Strengths

  1. This explains reasonably well Lee's ideas about a quantum renormalisation group and attempts some advance on these ideas.

Weaknesses

  1. The ideas rest on formal manipulations. It is far from clear that these manipulations are well defined outside the simple models considered.

  2. It is not clear to me that this paper adds anything substantive to previous papers the senior author has already released on the subject.

Report

The central claim of the paper (that the beta-functions for all symmetric operators in a general quantum field theory can be reconstructed from beta-functions in a measure zero subspace of such conjugate couplings) is not surprising, since the continuum limit of such quantum field theories are parametrised by typically a finite dimensional space of marginally relevant couplings. At least within perturbation theory, there are well established procedures for determining the form of the beta-functions for all other operators in terms of these couplings -- crucially after appropriate care is taken to define the regularisation and renormalisation of those operators. What the authors actually do is both weaker (they are typically left to deal with a subspace still containing an infinite number of couplings) and nowhere near as well-defined. To achieve their result they cast the Polchinski version of the exact renormalisation group flow equation into a kind of functional Schrodinger equation where the dependence on sources for all operators outside the subspace, is finessed into a functional generalisation of a Schrodinger wave function. No algorithm is presented for solving this functional Schrodinger equation, or justification for thinking this is easier than working with the original quantum field theory (most likely it is not) and it is far from clear that this reformulation is in general well defined (again most likely it is not). Instead in examples, the authors make Gaussian-like ansatze for the beta functions inside this subspace, equations (93) and (109). Only with these simple ansatze can they proceed to `reconstruct' the full theory but in these now over-simplified models.
  • validity: low
  • significance: low
  • originality: good
  • clarity: good
  • formatting: good
  • grammar: reasonable

Author:  Han Ma  on 2021-07-18  [id 1578]

(in reply to Report 1 on 2021-05-11)
Category:
answer to question
reply to objection

We thank the referee for the comments. Here are our responses to the referee’s comments.

  1. “The ideas rest on formal manipulations. It is far from clear that these manipulations are well defined outside the simple models considered.” The bulk theory that describes the quantum RG evolution of a boundary field theory is well defined as far as the boundary field theory is regularized. The regularization of the bulk theory follows from the regularization of the boundary field theory. For example, the bulk theory is a finite quantum mechanical theory of lattice variables for a boundary theory defined on a lattice. For example, well-defined bulk theories for lattice gauge theories and lattice vector models can be found in Nucl. Phys. B 862, 781 (2012) and JHEP (2016) 2016: 44, respectively. While it is generally hard to solve those bulk theories analytically, they can be solved in a computer with enough computational power as they are well defined quantum mechanics problems. In section II of the revised manuscript, we explicitly construct well-defined bulk theories for two realistic field theories that are regularized on a lattice.

  2. “It is not clear to me that this paper adds anything substantive to previous papers the senior author has already released on the subject; at least within perturbation theory, there are well established procedures for determining the form of the beta-functions for all other operators in terms of these (a finite number of marginally relevant) couplings — what the authors actually do is weaker” What is new in this paper is the constraints that beta functions satisfy non-perturbatively in general field theories. This result is beyond the known constraint of beta functions present in perturbative field theories defined in the continuum. To understand this, let us consider a field theory defined non-perturbatively with a finite UV cutoff. Examples include field theories regularized on a lattice. In such theories, there are infinitely many couplings that can be turned on independently at UV. If the energy scale of an effective theory is much smaller than the UV cutoff, the RG flow of all couplings is indeed fixed by a small number of marginal and relevant couplings. At high energies, however, irrelevant couplings are not fixed by the marginal and relevant couplings through that constraint that emerges only in the low-energy limit, and one has to keep track of the flow of infinitely many couplings. The constraint discussed in our paper is new as it applies to the exact beta functions at all energy scales even at scales that are comparable to the UV cutoff.

  3. “No algorithm is presented for solving this functional Schrodinger equation, or justification for thinking this is easier than working with the original quantum field theory (most likely it is not)” It is true that the bulk theory obtained form quantum RG is generally hard to solve because the bulk theory becomes semi-classical only in the large N limit. However, the main purpose of the present paper is not to provide a general way of solving strongly interacting bulk theories. Rather, our goal is to uncover non-perturbative constraints that beta functions satisfy through quantum RG. These constraints hold for any N, irrespective of whether the bulk theory is in the semi-classical limit.

  4. “it is far from clear that this reformulation is in general well defined (again most likely it is not). Instead in examples, the authors make Gaussian-like ansatze for the beta functions inside this subspace, equations (93) and (109). Only with these simple ansatze can they proceed to `reconstruct' the full theory but in these now over-simplified models.” The quantum RG gives well-defined bulk theories as far as the boundary field theories are regularized. Furthermore, the constraints of beta functions uncovered in our paper are valid for general theories beyond the toy models. In order to demonstrate this, in the revised manuscript, we added a new section (Sec. II) in which well-defined bulk theories are explicitly derived for two realistic field theories regularized on a lattice. We also derive the full beta functions of these theories solely from the beta functions defined in the subspace of single-trace couplings. The salient point of our paper is not the exact beta-functions itself but the fact that the entire beta-functions are fully characterized by a small set of data defined in the subspace of single-trace couplings.

---

## Round 3 · Referee Report · Anonymous (Referee 2) · 2021-5-17

Report

I believe the paper requires a significant rewriting. Please see the report attached

Attachment

  • validity: -
  • significance: -
  • originality: -
  • clarity: -
  • formatting: -
  • grammar: -

Author:  Han Ma  on 2021-07-18  [id 1579]

(in reply to Report 2 on 2021-05-17)

We thank the referee for the comments. Here are our responses to the referee’s comments.

  1. “I do not see why for a generic strongly coupled QFT there should be a preferred set of single-trace fields” There needs not be a preferred set of single-trace operators. All we need in quantum RG is the existence of a set of single-trace operators. Its existence follows from the fact that the space of theories can be always viewed as a Hilbert space, where an action S[\phi] of fundamental field \phi defines a wavefunction exp( -S[\phi] ) in the Hilbert space. Then, it follows that there exists a set of basis states that span the Hilbert space. In general, there exist multiple ways of choosing a complete set of basis states. Moreover, the basis states do not need to be orthogonal, and an over-complete set is an acceptable choice. Once a complete set of basis is chosen, the wavefunctions of the basis states define a set of actions. The operators that are needed to construct the wavefunctions of the basis states define the set of single-trace operators in general theories. This is explained in details for the O(N) vector model and the O(N)*O(N) matrix model in section II. While it is true that one choice of basis may give a simpler bulk theory and a stronger constraint, the purpose of our paper is to demonstrate the existence of general constraints from one choice of basis.

  2. “why it (the set of single-trace fields) should be the same at different energy scales” The Hilbert space associated with the space of theories is independent of scale. What is scale dependent is the effective action and the associated state in the Hilbert space that runs along the RG flow. Therefore, one can always choose a set of basis states in a scale independent way. This is explicitly shown to be the case in the examples included in Sec. II. While it is in principle possible to choose basis states in a scale dependent way, this is not necessary.

  3. “the path integral over HS (Hubbard-Stratonovich) fields will generically get strongly coupled and I expect the book keeping of operators used throughout the paper to fail” Indeed, the dynamical single-trace couplings are strongly interacting in theories with small numbers of flavours or colours. Only in the large N limit, the interactions become weak, and one can use a semi-classical approximation. However, the book keeping does not fail because the basis states made of the single-trace operators span the full Hilbert space independent of their dynamics. Furthermore, the constraints of beta functions discussed in this paper does not require that the bulk theory is weakly interacting. In the revised manuscript, we explicitly compute the full beta functions of two realistic models valid for any N entirely from the beta functions defined in the space of single-trace operators.

  4. “there is a Hamiltonian constraint HΨ = 0, and Shroedinger equation only appears in a semiclassical expansion of this constraint” In quantum RG, general states defined at a scale are not annihilated by the RG Hamiltonian. This is because the effective action generally changes as a function of RG scale. What is invariant is an overlap between two states associated with a fixed point and a deformation, where the overlap corresponds to the generating function of the boundary theory. This is described in Sec. II and III in details.

  5. “in a real QFT this would still be infinitely many operators” It is true that even in quantum RG one needs to include infinitely many single-trace operators in the thermodynamic limit. This is because couplings are in general space dependent. In the vector model, these are position dependent bi-local operators, and in the matrix model, they are loop operators, as is discussed in Sec. II in details. Nonetheless, the exact mapping from the Wilsonian RG to quantum RG is powerful enough to reveal general constraints among beta functions.

  6. “Can any of this be achieved using the authors’ method in any interesting QFTs?” In the revised manuscript, we add a new section for two realistic models. The first is the O(N) vector model and the other is the O(N)*O(N) matrix model. It is explicitly shown that the full beta functions can be obtained solely from the beta functions defined in the subspace of single-trace operators in these models.

  7. “what prevents one from considering an RG flow with an arbitrary β2(j, j2), without modifying the beta functions at zero j2?” Naively, one might think that the beta functions away from the subspace of j2=0 can be modified without modifying the beta function on the subspace. However, this is impossible due to the constraint that we find in this paper - this is the main point of our paper. Because multi-trace operators are composites of the single-trace operators, the RG flow in the presence of general multi-trace operators are completely fixed by the beta functions defined in the subspace of single-trace couplings. This constraint holds even when multi-trace operators have large anomalous dimensions. This is explicitly demonstrated through the two realistic field theories in the revised manuscript.

---

## Round 4 · Referee Report · Anonymous (Referee 1) · 2021-8-22

Report

The addition of the new sec. II has considerably clarified and strengthened the paper. I recommend publication in its current form.

---

## Round 4 · Referee Report · Anonymous (Referee 2) · 2021-9-25

Report

See attached

Attachment

  • validity: -
  • significance: -
  • originality: -
  • clarity: -
  • formatting: -
  • grammar: -

Author:  Han Ma  on 2021-10-11  [id 1838]

(in reply to Report 2 on 2021-09-25)

We thank the referee for the comments. Below, we provide our responses to the comments.

1) ``the procedure proposed by the authors is a mere rewriting of the Wilsonian path integral which, unless the the theory has a meaningful gravitational dual, does not lead to any improvement of understanding of RG flows''

Besides being an exact reformulation of the Wilsoninan RG, quantum RG (QRG) can be of practical use. First, it can be used to compute the exact quantum effective action of interacting field theories in the large N limit. Recently, the exact quantum effective action has been computed for the O(N) vector model in the large N limit via QRG [arXiv:2107.05654]. The exact effective action was not known before even for that relatively simple model because it includes operators with arbitrarily many fields and derivatives. Second, QRG provides a prescription to derive concrete holographic duals for general quantum field theories. As far as boundary quantum field theories are regularized, the bulk theories that arise from QRG are fully regularized. It is known that the bulk theory includes dynamical gravity [JHEP 1210 (2012) 160; JHEP 01 (2014) 076]. What is not well understood is how those regularized theory of dynamical gravity are related to continuum theories.

2) ``why not choosing just the field bi-linears as a basis also in the example of section IIB (O(N) × O(N) matrix model)?''

One can not choose bi-linears as a basis in the matrix model. This is because the single-trace operators should be singlets of the symmetry, and the set of bi-linear singlet operators of the matrix model do not form a complete basis of the vector space formed by the actions that respect the symmetry.

3) ``I would ask the authors to make a very concrete list of calculable physical quantities or properties of realistic QFTs that can be computed using their method, as well as an estimate of how hard the computation is''

We are grateful to the referee for the suggestion. In the revised manuscript, we added the following paragraph that includes a list of open questions and future directions. We feel that it is natural to place it in the conclusion as these are not directly related to the main content of the present paper.

We conclude with open questions and future directions. First, QRG can be used to compute the exact quantum effective action. The scale dependence of the quantum effective action obeys the exact RG equation[6,8]. In general solving the exact RG equation is challenging because the exact effective action includes operators made of arbitrarily many fields and derivatives. As a result, exact effective actions remain unknown even for relatively simple theories. In QRG, the exact RG equation is mapped to a quantum evolution of a wavefunction for single-trace couplings. Since the set of single-trace operators is far smaller than the set of all possible operators, QRG can be potentially more tractable. For general quantum field theories, it is still difficult to solve the corresponding quantum evolution problem in QRG. However, in the large N limit, quantum fluctuations of the single-trace couplings become weak, and the theory that describes QRG evolution becomes classical. In the large N limit, the solution to the exact RG equation can be obtained from the saddle-point solution. Recently, the exact effective action for the O(N) vector model has been computed from QRG in the large N limit[34]. It would be of great interest to compute exact effective actions for matrix models in the large N limit. Second, QRG provides a concrete prescription for constructing the holographic duals for general quantum field theories[16]. The construction gives a well-defined bulk theory that includes dynamical gravity as far as the boundary theory is regularized[15,30,39]. However, the continuum limit of the bulk theory obtained from regularized boundary theories such as lattice models is not fully understood. It is of great interest to understand how the regularized bulk theory obtained from QRG is related to continuum theories conjectured as holographic duals of known field theories in the semi-classical limit.

---

## Round 4 · List of Changes

The main change of the manuscript is that we added a new section (Sec. II) in which our result is applied to two realistic field theories: the O(N) vector model and the O(N)*O(N) matrix model. In the revised manuscript, we explicitly construct the well-defined bulk theories that govern the quantum RG flow of these realistic models. We also demonstrates the main claim of our paper by constructing all beta functions entirely from the beta functions defined in the subspace of single-trace couplings in those theories.

---

## Round 5 · List of Changes

We have added a paragraph that lists open questions and future directions in the conclusion of the revised manuscript.

---

## Editorial Decision

publication_decision_taken:_accept